# Measurement report: Atmospheric Ice Nuclei at Changbai Mountain (2623 m a.s.l.) in Northeastern Asia

Yue Sun[1], Yujiao Zhu[1], Yanbin Qi[2], Lanxiadi Chen[3], Jiangshan Mu[1], Ye Shan[1], Yu Yang[1], Yanqiu Nie[1], Ping Liu[1], Can Cui[1], Ji Zhang[1], Mingxuan Liu[1], Lingli Zhang[4], Yufei Wang[2], Xinfeng Wang[1], Mingjin Tang[3], Wenxing Wang[1], Likun Xue[1]

[1]Environment Research Institute, Shandong University, Qingdao 266237, China
[2]Jilin Provincial Technology Center for Meteorological Disaster Prevention, Changchun 130062, China
[3]State Key Laboratory of Organic Geochemistry, Guangzhou Institute of Geochemistry, Chinese Academy of Sciences, Guangzhou 510640, China
[4]Changbai Mountain Meteorological Observatory, An Tu, Jilin 133613, China

*Correspondence to*: Yujiao Zhu (zhuyujiao@sdu.edu.cn), Likun Xue (xuelikun@sdu.edu.cn)

**Abstract.** Atmospheric ice nucleation plays an important role in modulating the global hydrological cycle and atmospheric radiation balance. To date, few comprehensive field observations of ice nuclei have been carried out at high-altitude sites, which is close to the height of mixed-phase cloud formation. In this study, we measured the concentration of ice-nucleating particles (INPs) in the immersion freezing mode at the summit of Changbai Mountain (2623 m above sea level), Northeast Asia, in summer 2021. The cumulative number concentration of INPs varied from $1.6 \times 10^{-3}$ $L^{-1}$ to 78.3 $L^{-1}$ over the temperature range from −5.5 ℃ to −29.0 ℃. Proteinaceous-based biological materials accounted for the majority of INPs, with the proportion of biological INPs (bio-INPs) exceeding 67% across the entire freezing temperature range, with this proportion even exceeding 90% above −13.0 ℃. At freezing temperatures ranging from −11.0 ℃ to −8.0 ℃, bio-INPs were found to significantly correlate with wind speed ($r = 0.5-0.8$, $p < 0.05$) and $Ca^{2+}$ ($r = 0.6-0.9$), and good but not significant correlation was found with isoprene ($r = 0.6-0.7$) and its oxidation products (isoprene × $O_3$) ($r = 0.7$), suggesting that biological aerosols may attach to or mix with soil dust and contribute to INPs. During the daytime, bio-INPs showed a positive correlation with the planetary boundary layer height at freezing temperatures ranging from −22.0 ℃ to −19.5 ℃ ($r > 0.7$, $p < 0.05$), with the valley breezes from southern mountainous regions also influencing the concentration of INPs. Moreover, the long-distance transport of air mass from the Japan Sea and South Korea significantly contributed to the high concentrations of bio-INPs. Our study emphasizes the important role of biological sources of INPs in the high-altitudes atmosphere of northeastern Asia, as well as the significant contribution of long-range transport to the INP concentrations in this region.

## 1 Introduction

Clouds play a crucial role in regulating the Earth's energy balance by absorbing, reflecting, and scattering solar and terrestrial radiation (Zhou et al., 2016; Bjordal et al., 2020). Global precipitation is predominantly produced by clouds

containing the ice phase, especially in continental regions and mid-latitude oceans, emphasizing the paramount significance of investigating ice formation within clouds (Mulmenstadt et al., 2015; Lau and Wu, 2003; Demott et al., 2010; Kanji et al.,

2017). Atmospheric aerosols can act as ice-nucleating particles (INPs), triggering the freezing of cloud droplets through heterogeneous nucleation processes (Rinaldi et al., 2017; Rosenfeld and Woodley, 2000; Demott et al., 2003; Murray et al., 2012). Currently, the four main mechanisms of heterogeneous ice nucleation are considered: deposition nucleation, condensation freezing, immersion freezing, and contact freezing (Demott et al., 2003; Vali et al., 2015). Recent studies have concluded that water saturation is a prerequisite for ice formation in low- and mid-level clouds, and that contact and immersion freezing are the most primary pathways for ice formation (Sassen and Khvorostyanov, 2008; Phillips et al., 2007;

Murray et al., 2012).

Various aerosol particles are potential INPs, such as biological aerosols (Pratt et al., 2009; Tobo et al., 2013), mineral dusts (Pratt et al., 2009; Atkinson et al., 2013), sea spray aerosols (Mccluskey et al., 2017; Alpert et al., 2022), carbonaceous aerosols (Grawe et al., 2018; Demott, 1990; Diehl and Mitra, 1998; Fornea et al., 2009), and volcanic ashes (Grawe et al., 2016; Umo et al., 2015). Biological aerosols from microbial and proteinaceous origin, containing ice nucleation active

protein, demonstrate significant efficiency as INPs at temperatures above −15 ℃ (Phelps et al., 1986; Petters and Wright, 2015; Murray et al., 2012; Kunert et al., 2019; Huang et al., 2021). For example, biological components in lichens can induce freezing above −10 ℃ (Kieft, 1988; Moffett et al., 2015), and certain bacterial organisms such as *Pseudomonas syringae* can facilitate droplet freezing even at extremely high temperatures (above −2 ℃) (Maki et al., 1974). The other biological aerosols with non-proteinaceous compounds act as ice nucleation catalysts, such as pollen, cellulose, and other

macromolecular organic particles, can also induce ice formation through heat-resistant polysaccharides on their surfaces, but at lower temperatures than proteinaceous biological particles (Knopf et al., 2010; Pummer et al., 2012). Mineral dust and sea spray aerosols predominantly consist of inorganic compounds and serve as effective INPs at temperatures below −15℃ (Alpert et al., 2022; Atkinson et al., 2013; Ladino et al., 2019). The ice-nucleating properties of aerosols are affected by many factors. For instance, the size of particles is a crucial factor for providing active sites for ice formation, with larger

particles containing more efficient ice nucleation sites than smaller ones (Chen et al., 2018; Demott et al., 2010; Demott et al., 2015). In addition, the chemical composition and surface properties of aerosol particles, such as their surface topology, defects, roughness, and functional groups, also influence their activity as INPs (Hiranuma et al., 2014; Marcolli, 2014; Roudsari et al., 2022; Yun et al., 2021; Zolles et al., 2015). Furthermore, the ice-nucleating properties of aerosol particles can be modified through chemical reactions with trace gases or organic/inorganic component or through physical processes

such as efflorescence or deliquescence (Cziczo et al., 2009; Hoose and Möhler, 2012; Creamean et al., 2013; Tang et al., 2016; Tang et al., 2018).

Over recent decades, numerous studies have focused on investigating heterogeneous ice nucleation in various atmosphere environments (Gong et al., 2022; Chen et al., 2021; Testa et al., 2021; Knopf et al., 2023; Harrison et al., 2022; Ren et al., 2023; Sanchez-Marroquin et al., 2023). In low-altitude atmospheric environments, the abundance of ground-based

sources and sinks results in spatial distribution heterogeneity (Larsen, 2007), which restricts the characterization of INPs properties on a regional scale. At present, there remain uncertainties about how INPs transported to the altitudes of mixed-phase cloud formation (approximately 3–7 km) (Knopf et al., 2018). High-altitude sites provide favorable conditions for in situ observations to investigate INPs characteristics, as they can represent tropospheric background conditions and reflect long-distance transport and vertical mixing processes prior to arriving at the ground sampling site (Wieder et al., 2022).

Therefore, field experiments have been conducted in several high-altitude sites. For example, in Switzerland, simultaneous measurements taken at different-altitude stations revealed a reduction of approximately 50% per kilometer in the abundance of INPs in the vertical gradient (ranging from 489 m above sea level (a.s.l.) to 3580 m a.s.l. in the Swiss Alps) in the warm season (Conen et al., 2017). This decline in INPs could exceed 60% per kilometer during the cold season (from 1631 m a.s.l. to 2693 m a.s.l.) (Wieder et al., 2022), which was attributed to the scarcity of effective INPs sources in high-altitudes. Note

that variations in sampling methods and the influence of wind directions can also exert an impact on INP concentrations. Schrod et al. (2017) reported an increase in INPs abundance of approximately 10 times over the eastern Mediterranean (2500 m a.s.l.) relative to ground level using unmanned aircraft systems, with this difference attributed to the long-distance transport of a series of elevated Saharan dust plumes at the height of a few kilometers. In some mountainous areas, biogenic aerosols can act as the most abundant type of INPs. For example, at the Jungfraujoch station (3580 m a.s.l.) in the Swiss

Alps, approximately 80% of INPs were biological aerosols at freezing temperatures above −15 °C under free-tropospheric conditions (Conen et al., 2022). Similarly, at the Puy de Dôme station (1465 m a.s.l.) in France, the average contribution of biological aerosols in cloud water could reach up to 85% at freezing temperatures above −10 °C (Joly et al., 2014). To date, fewer field observations have been carried out in high-altitude regions in China. For example, Jiang et al. (2014, 2015) performed measurements at Mt. Huangshan (1840 m a.s.l.) in Southeast China, finding that larger particles were more likely

to be effective INPs. Lu et al. (2016) collected seven rainwater samples from three high-altitude sites in eastern China, i.e., Changbai Mountain (at the peak of 2740 m a.s.l.), Wuling Mountain (900 m a.s.l.), and Dinghu Mountain (1000 m a.s.l.). The authors found that the onset freezing temperature was approximately −6 °C, but bacteria played minor roles in the overall INP activity. Because the number of rainwater samples was limited, further research is necessary to explore the impact of biological INPs on ice formation and their contribution to the formation of precipitation.

In this study, we conducted offline INP measurements at the top of Changbai Mountain (2623 m a.s.l.) in Jilin province, China, which is located in Northeast Asia. This region is particularly vulnerable to climate change because of the presence of distinct ecotones caused by land type changes, as well as the influence of the North Atlantic Oscillation and the Northern Hemisphere circulation (Sugita et al., 2007; Zhang et al., 2021). Moreover, Northeast Asia is densely populated and serves as a crucial breadbasket for the world, making rainfall an essential factor for determining crop yields (Zhang et al., 2021).

Given the high-altitude of Changbai Mountain, it is an ideal location to capture the characteristics of the regional atmospheric background and transboundary transport of air masses from nearby countries. Our main objective was to investigate the concentration levels of INPs and identify their major sources at the height of the mountain's peak.

Additionally, we evaluated the impact of the planetary boundary layer (PBL) height, valley breezes, and transport pathways on the INP population to gain a better understanding of their sources in this region. Our findings could provide valuable insights into the formation and behaviour of mixed-phase clouds over this region.

## 2 Methods

### 2.1 Site description

Changbai Mountain is the highest mountain in the border region between China and the Korean Peninsula. It is situated on the transport pathways of continental air pollutants from Asia to the North Pacific Ocean and even as far as the Arctic (Ikeda et al., 2021). The regional topography is characterized by forests and mountains, with elevations ranging from 410 m a.s.l. to 2740 m a.s.l. The southeast exhibits higher elevations compared to the northwest (Wang et al., 2014). At the top of Changbai Mountain, there is a vast dormant crater known as Tianchi Lake, which has a depth of 373 m and covers an area of 9.82 km$^2$. In this study, a field campaign was carried out at the Tianchi Meteorological Station (Tianchi Site, 42.03°N, 128.07°E, 2623 m a.s.l., Figure 1), which is approximately 410 m north of Tianchi Lake, from July 24 to August 24, 2021.

Changbai Mountain is situated within the westerly wind belt and experiences a typical temperate continental mountain climate influenced by the monsoon, characterized by long cold winters and short temperate summers. The prevailing winds in this region are the westerly and northwesterly winds in the spring, autumn, and winter season and the southeasterly and southwesterly winds in the summer season (Zhao et al., 2015). The annual average temperature is typically lower than −7.4 °C (Jin et al., 2018), with the mountain summit always covered by snow and ice for approximately three quarters of the year. Figure S1 presents the timeseries of meteorological parameter, NOx concentration, and the of INP concentrations during the field campaign. During the campaign, the relative humidity (RH) ranged from 33% to 100%, with a mean of 92.4 ± 0.4% (average ± standard error of the mean (SEM)). Notably, seventy percent of the RH exceeded 90% throughout the campaign, indicating that the campaign was performed under humid weather conditions. The sampling site was predominantly affected by southerly and westerly winds, with wind speed (WS) ranging from 0.1 m s$^{-1}$ to 25.7 m s$^{-1}$. Changbai Mountain is a national natural reserve with no large industrial facilities nearby, and tourism is the important economic activity in the region. Due to the COVID-19 pandemic, strict lockdown measures were implemented from August 10, 2021, resulting in a substantial reduction in visitor numbers, as indicated by the marked decrease in NOx concentration (Figure S1). The surroundings of the observation site are covered by dense vegetation, such as shrubs and perennial herbs. Most of the time, the site is above the PBL and in the free troposphere, making it an ideal site for studying the regional background atmosphere of Northeast Asia.

**2.2 Sample collection**

Total suspended particulate (TSP) was collected on polycarbonate (PCTE) membrane filters (Sterlitech 1870, nominal porosity 0.45 μm) using a TH-150D medium flow sampler (Wuhan Tianhong Corporation, China, Figure 1c) at a flow rate of 50 L min$^{-1}$ for the INPs analysis. Samples were collected during the daytime (06:00 to 17:30) and nighttime (18:00 to 05:30 in the following day) in local time. A total of 24 PCTE filters were collected, including 22 aerosol samples and 2 blank filters. These samples were used for INP analysis with the detailed sampling information provided in Table S1. Meanwhile, fine particulate matter (PM$_{2.5}$) samples were collected on quartz microfiber filters (PALL Pallflex, 7204), which were heated at 560 ℃ for 4 h before sampling to remove any adsorbed organics, using a TH-150A medium flow sampler (Wuhan Tianhong Corporation, China) with a 2.5 µm impactor at a flow rate of 100 L min$^{-1}$. A total of 157 samples were collected on quartz filters every 3 h and used for chemical composition analysis. After sampling, all filter samples were kept frozen at ≤ −18 ℃ until analysis.

Real-time measurements of PM$_{2.5}$ and black carbon (BC) were recorded at 1 min intervals by using SHARP 5012 (Thermo Scientific, USA) and SHARP 5030 (Thermo Scientific, USA), respectively. Trace gases including CO, SO$_2$, NO$_x$, and O$_3$, were detected using Thermo Scientific 48i, 43i, 42i, and 49i, respectively. Ambient volatile organic compounds (VOCs) were collected by taking air samples using stainless-steel canisters at two specific time intervals (i.e., 11:00-13:00 and 20:00-22:00) on clean days, and the sampling frequency was increased to every 3h (5 samples per day) during air pollution episodes. Meteorological data, such as temperature, humidity, WS, wind direction, pressure, and precipitation, were monitored by the Tianchi weather station, a national meteorological station located approximately 20 m away from the sampling site. The parameters mentioned above were analyzed by determining the average value over the corresponding INPs sampling period.

**2.3 INPs analysis**

INP measurements in the immersion mode were conducted using the Guangzhou Institute of Geochemistry Ice Nucleation Apparatus (GIGINA) from −40 ℃ to 0 ℃. GIGINA is a cold-stage-based ice nucleation array that consists of an enclosed droplet chamber with a commercial cold stage inside (LTS120, Linkam, Epsom Downs, UK), an external refrigerated water circulator (VIVO RT4, Julabo, Seelbach, Germany), and a charge-coupled device (CCD) camera (DMK33G274, The Imaging Source, Bremen, Germany). Further details regarding GIGINA have been published by Chen et al. (2023).

Each polycarbonate filter was immersed in 5 mL MilliQ water (resistivity of 18.2 MΩ cm$^{-1}$ at 25 ℃) and sonicated for 30 min to wash off the particles (Chen et al., 2021). Note that an ice water bath was utilized during ultrasonic extraction to mitigate any potential alterations in protein properties and biogenic activities. In addition, the suspension underwent dilution at multiple levels: 30-fold, 60-fold, and 120-fold, in order to generate a spectrum that encompass freezing temperature below

−25°C, as illustrated in Figure S2. The INPs measurement process is briefly described as follows. First, a hydrophobic glass slide was placed on a cold stage and covered with silicone oil between them to achieve good thermal contact. Second, a round aluminum spacer with 90 round compartments was placed on the glass slide, and the particle suspension was sequentially pipetted into the center region of each compartment. Then, another glass slide was placed above the spacer to avoid the Wegener–Bergeron–Findeisen process (Jung et al., 2012). Afterward, the temperature of the droplets was cooled down to 0 °C at a cooling rate of 10 °C min⁻¹, after which the cooling of the droplets continued at a rate of 1 °C min⁻¹ until all the droplets were frozen. During the freezing experiment, high-purity nitrogen was continuously delivered onto the cold stage to prevent frost from forming on the surface of the glass slide. Meanwhile, real-time images of the droplets were photographed by the CCD camera and recorded by the LINK software every 6 s. After the experiment, the phase transition of each droplet was identified by analyzing the changes in image brightness, which distinguished between unfrozen (white) and frozen (dark) droplets.

The frozen fraction, $f_{ice}$, was calculated according to Eq. (1):

$$f_{ice}(T) = \frac{n_{ice}}{n_{tot}} \ ,\tag{1}$$

where $n_{ice}$ is the number of frozen droplets at a certain temperature $T$, and $n_{tot}$ is the total number of droplets (90 droplets). The cumulative number concentration of INPs ($N_{INP}$) per unit volume of sampled air were calculated following the method of Vali (1971, 2015):

$$N_{INP}(T) = -\frac{ln[1-f_{ice,\ sample}(T)] - ln[1-f_{ice,\ blank}(T)]}{V_{air}} (L^{-1}\ air),\tag{2}$$

where $V_{air}$ is the total volume of sampled air per droplet converted to standard conditions (0 °C and 1013 hPa). The $f_{ice,\ sample}$ and $f_{ice,\ blank}$ are the measured frozen fractions for the filter samples and the field blanks, respectively. The calculation for $V_{air}$ entails multiplying the droplet volume (1 µl) by the sample volume and then dividing by the volume of wash water. In our study, the $N_{INP}$ values were significantly larger in filter samples than in the field blanks. In the following analysis, the concentrations of the two blank filters were subtracted from the daytime and nighttime samples at each freezing temperature, respectively. The rarity of INPs in the atmosphere leads to their low concentration in the suspension. Because the suspension used in the measurement contained a limited number of droplets, we need to consider the resulting uncertainty caused by statistical errors. Therefore, we calculated the confidence intervals of the apparatus for $f_{ice}$ according to the method of Gong et al. (2022) and Agresti and Coull (1998) to address uncertainty associated with the droplet-freezing apparatus:

$$\left(f_{ice} + \frac{Z_{\alpha/2}^2}{2n_{tot}} \pm Z_{\alpha/2}\sqrt{[f_{ice}(1-f_{ice}) + Z_{\alpha/2}^2/(4n_{tot})]/n_{tot}}\right) / \left(1 + Z_{\alpha/2}^2/n_{tot}\right),\tag{3}$$

where $Z_{\alpha/2}$ is the standard score at a confidence level $\alpha/2$, which is 1.96 for a 95% confidence interval.

## 2.4 Chemical analysis

The $PM_{2.5}$ samples collected by quartz membranes were used to analyze the particle chemical composition. For each sample, an eighth of the filter was ultrasonically extracted using 15 mL MilliQ water for 30 min to make a suspension. The concentrations of inorganic water-soluble anions ($Cl^-$, $SO_4^{2-}$, and $NO_3^-$) and cations ($Na^+$, $NH_4^+$, $K^+$, $Mg^{2+}$, and $Ca^{2+}$) were identified using the ICS 1100 ion chromatograph (Thermo Scientific). In addition, the concentrations of organic carbon (OC) and elemental carbon (EC) were measured using the Sunset Laboratory Model-5 semi-continuous OC/EC field analyzer. The VOCs canister samples were analyzed using online gas chromatography–mass spectrometry (TT24xr, Makers, UK; GC–MS, Thermo Scientific, USA) in the laboratory. A total of 106 target VOCs, including 29 alkanes, 11 alkenes, one alkyne, 17 aromatics, 35 halogenated hydrocarbons and 13 oxygenated VOCs (OVOCs), were quantified.

## 2.5 The PBL data and air mass back trajectory model

The PBL data were downloaded from the fifth-generation ECMWF global atmospheric reanalysis product (ERA5 https://cds.climate.copernicus.eu), which provides hourly records on latitude–longitude grids at $0.25° \times 0.25°$ resolution. The 72-h air mass backward trajectories at the sampling site were calculated on an hourly basis during the sampling days. These calculations were performed using the Hybrid Single Particle Lagrangian Integrated Trajectory (HYSPLIT) model (http://ready.arl.noaa.gov/HYSPLIT.php), which is developed by the National Oceanic and Atmospheric Administration Air Resources Laboratory (NOAA ARL) (Stein et al., 2015). The simulations were based on meteorological data from the Global Data Assimilation System (GDAS) with a spatial resolution of $1° \times 1°$. A trajectory ending height of 967 m above ground level (a.g.l.) was selected because the terrain height of Changbai Mountain was approximately 1656 m in the GDAS data. Using the open-source software of MeteoInfo, concentration-weighted trajectory (CWT) analysis was conducted to explore the potential sources of INPs based on the air mass backward trajectories and $N_{INP}$. The CWT reflects the pollution levels of different trajectories by calculating the weight concentration of the airmass trajectory in potential source areas. The calculation was used Equation 4 according to the method of Hsu et al. (2003):

$$C_{ij} = \frac{1}{\sum_{k=1}^{M} \tau_{ijk}} \sum_{k=1}^{M} C_k \tau_{ijk}, \tag{4}$$

where $C_{ij}$ is the average weighted concentration in the $ij$ cell, $k$ is the index of the trajectory, $M$ is the total number of trajectories, $C_k$ is the concentration observed on arrival of trajectory $k$ in the $ij$ cell, and $\tau_{ijk}$ is the time spent in the $ij$ cell by trajectory. The number of endpoints that fall in the cell is defined as nij, and the weight function $W_{ij}$ is used to reduce the uncertainty in the cells with small values of $n_{ij}$:

$$WCWT_{ij} = C_{ij} \times W(n_{ij}), \tag{5}$$

Note that CWT analysis does not account for the impacts of wet deposition such as precipitation along the trajectories. Uncertainties may exist in the CWT analysis due to the relatively small dataset of INPs in this study.

# 3 Results and Discussion

## 3.1 INP concentrations

The freezing temperature at which 50% of the droplets are frozen ($T_{50}$) was calculated for each sample. The frozen fractions ($f_{ice}$) of all freezing curves containing the collected samples and MilliQ water are shown in Figure S2. For the blank filters, $T_{50}$ was averaged to $-24.9 \pm 1.1$ °C, which was slightly higher than that of MilliQ water, but significantly lower than that of the collected samples (for which $T_{50}$ was $-17.0 \pm 0.6$ °C). This result suggests the presence of minimal contaminants stemming from the filter membrane.

The $N_{INP}$ values as a function of temperature are presented in Figure 2, where the pink and blue circles represent the samples collected during the daytime and nighttime, respectively. The freezing of ambient samples was observed in the temperature range of $-5.5$ °C to $-29.0$ °C, with $N_{INP}$ spanning up to four orders of magnitude from $1.6 \times 10^{-3}$ L$^{-1}$ to 78.3 L$^{-1}$. For freezing temperatures above average $T_{50}$ ($-17.0$ °C), the temperature region is referred to as the high-temperature region (HTR), where $N_{INP}$ spans three orders of magnitude from $1.6 \times 10^{-3}$ L$^{-1}$ to 6.2 L$^{-1}$. Some of the $N_{INP}$ curves exhibited bumps in the HTR, which has been also observed at a coastal site (the Cape Verde Atmospheric Observatory, Africa) by Welti et al. (2018) in air samples and in the upper bound of the composite nucleus spectrum of cloud water and precipitation samples by Petters and Wright (2015). Welti et al. (2018) reported that the narrower the IN properties, the steeper slope can be observed in a temperature spectrum. In contrast, in the low-temperature region (LTR, freezing temperature below average $T_{50}$, $-17.0$ °C ~ $-29.0$ °C), $N_{INP}$ showed a relatively narrow variation from 0.1 L$^{-1}$ to 78.3 L$^{-1}$. Furthermore, there were no significant differences observed in $N_{INP}$ between daytime and nighttime (the significance level is 0.61). However, in some mountainous sites, such as Mt. Huang, where samples were collected twice daily at 08:00 and 14:00 (Jiang et al., 2015), and the Weissfluhjoch in the Swiss Alps, where samples were collected at 20-minute intervals (Wieder et al., 2022), $N_{INP}$ displayed a distinct diurnal cycle induced by the orographically lifted air masses containing high INP concentrations from low elevation upstream during the daytime. In this study, local sources such as vegetation and the lake may impact INP concentrations, with biogenic emissions potentially exhibiting variations between daytime and nighttime. However, we collected two samples per day, and the limited dataset size and low sampling frequency may have contributed to the absence of diurnal variations.

We compared our $N_{INP}$ measurements with previous results from diverse sites. For instance, in mountainous regions, the $N_{INP}$ value at the Weissfluhjoch varied from $10^{-4}$ L$^{-1}$ to $10^1$ L$^{-1}$ in the temperature range of $-4.0$ °C to $-24.0$ °C (Wieder et al., 2022). In our observations, the spectra range of $N_{INP}$ were narrowly located in the relatively high-concentration regions at overlapping freezing temperatures compared to the measurements of Wieder et al. (2022). Jiang et al. (2015) reported that the INP concentrations at the top of Mt. Huang spanning from 0.1 L$^{-1}$ to 11.9 L$^{-1}$ over a temperature range from $-15.0$ °C to $-23.0$ °C, which overlapped with our results. In the LTR, our results were comparable to the measurements conducted at the Storm Peak Laboratory in the northwestern Colorado Rocky Mountains by Hodshire et al. (2022). But in HTR, Conen et al. (2022) recorded results in Switzerland were 1-3 orders of magnitude higher than our study. Gong et al. (2022) measured

INPs at the mountain station at Cerro Mirador (622 m a.s.l., Chile), and reported $N_{INP}$ values lower than those in our study by around one order of magnitude at similar freezing temperatures from −26.0 °C to −3.0 °C. In heavily polluted urban sites, such as Beijing (Chen et al., 2018) and Tai'an (Jiang et al., 2020) in China, the INP concentrations were comparable to our measurements at overlapping freezing temperatures. Chen et al. (2018) reported that INP concentrations might not be influenced by urban air pollution because no correlation was found between the immersion-freezing INP concentrations and the $PM_{2.5}$ or BC concentration.

### 3.2 Contribution of biological particles, other organics, and inorganics to INPs

Generally, biological particles can induce ice nucleation in the immersion mode at relatively high temperatures above −15.0 °C (Murray et al., 2012). Proteinaceous components mainly induce biological ice nucleation, and wet heat treatment (i.e., heating the particle suspension to 95.0 °C for 30 min) is used to identify the protein-based biological ice nucleation activity (Beall et al., 2022; Chen et al., 2021). We measured the $N_{INP}$ values of the suspensions after heat treatment, which we refer to as heat-resistant $N_{INP}$ ($N_{INP-heat}$, as shown in Figure S3), and the difference between the original $N_{INP}$ and $N_{INP-heat}$ was considered to be mainly due to the proteinaceous biological $N_{INP}$ ($N_{INP-bio}$). However, some biological aerosols, such as pollen, cellulose, or other macromolecular organic particles, are insensitive to heat treatment at 95.0 °C (Daily et al., 2022). Therefore, we also measured the heat-stable organic INPs, which are defined as other organic INPs (other org-INPs), following the methods of Suski et al. (2018) and Testa et al. (2021). We added 30% $H_2O_2$ (guaranteed reagent) to the suspension to obtain a final concentration of 10%, and then heated it at 95 °C for 20 min under UVB fluorescent bulbs. To prevent freezing point depression, we neutralized the remaining $H_2O_2$ in the suspension with catalase. The $N_{INP}$ value following treatment by this procedure was denoted $H_2O_2$-resistant $N_{INP}$ ($N_{INP-H2O2}$), which is the concentration of inorganic INPs ($N_{INP-inorg}$). The difference between heat-resistant $N_{INP}$ and $H_2O_2$-resistant $N_{INP}$ was considered to be equivalent to the concentration of other organic INPs ($N_{INP-other org}$).

Figure 3 illustrates the concentrations and fractions of the three types of INPs. The biological INPs (bio-INPs) showed ice nucleation activity at temperatures between −28.5 °C and −5.5 °C. After the ice nucleation activity of the bio-INPs was destroyed, $N_{INP-heat}$ decreased by around 1–2 orders of magnitude compared with the original $N_{INP}$, indicating a significant contribution of $N_{INP-bio}$, as shown in Figure S3. The initial freezing temperature of other org-INPs was −11.0 °C, which was approximately 5.5 °C lower than that of bio-INPs. Inorganic INPs exhibited ice nucleation activity at temperatures between −28.0 °C and −10.0 °C. Interestingly, the initial freezing temperatures of some inorganic INPs were slightly higher than those of other org-INPs, indicating that some inorganic aerosols could trigger freezing at relatively high temperatures.

The proportions of the three types of INPs as functions of temperature are presented in Figure 3(b). Here, the fractions of $N_{INP-bio}$ ($F_{INP-bio}$) account for 100% of the $N_{INP}$ value above −11 °C, and show a decreasing trend as the temperature decreases from −11.5 °C to −16.5 °C. This decreasing trend of $F_{INP-bio}$ is consistent with trends observed in other areas dominated by bio-INPs in similar temperature regions (Gong et al., 2022; Testa et al., 2021), suggesting that the importance

of bio-INPs decreases with decreasing temperature. Interestingly, when the temperature decreased from −16.5 °C to −21.5 °C, the median of $F_{INP\text{-}bio}$ increased from 0.8 to 0.9, indicating the high concentration of bio-INPs in the LTR. Previous observational studies have indicated that although most bio-INPs act as INPs at high freezing temperature, some heat-sensitive biological aerosols, such as fungal cloths, exhibit ice-nucleating activity at low temperatures (Iannone et al., 2011; Kanji et al., 2017). Modelling studies have also shown that bio-INPs can influence the ice phase of clouds and produce ice crystals when the cloud-top temperature is below −15 °C (Hummel et al., 2018). This phenomenon may also be related to the sensitivity of different species of biological aerosol to heating conditions. In wet heat treatment it is assumed that the ice-nucleating active protein in bio-INPs is completely destroyed and denatured, thus losing any ice formation potential. However, this method may lead to decrease in the freezing temperature of bacteria and fungi, but their ice-forming activity still cannot be ignored (Daily et al., 2022). As the temperature dropped to −25.0 °C, $F_{INP\text{-}bio}$ began to decrease significantly to 0.7. Overall, the median value of $F_{INP\text{-}bio}$ was more than 67% in the entire temperature range from −25.0 °C to −5.5 °C, with the value exceeding 90% above −13.0 °C, which was much higher than in some mountainous areas in southwestern South America (Gong et al., 2022) and urban areas in China (Chen et al., 2021). The fractions of $N_{INP\text{-}other\ org}$ ($F_{INP\text{-}other\ org}$) showed an opposite trend from that of $F_{INP\text{-}bio}$ at freezing temperatures between −25.0 °C and −11.0 °C. First, $F_{INP\text{-}other\ org}$ increased from 0.08 to 0.2 as the temperature decreased from −11.0 °C to −15.0 °C, and then sharply decreased from 0.2 to 0.05 as the temperature decreased further from −15.0 °C to −22.0 °C. When the temperature was lower than −22.0 °C, $F_{INP\text{-}other\ org}$ gradually increased to 0.3. The fractions of $N_{INP\text{-}inorg}$ ($F_{INP\text{-}inorg}$) remained below 0.22 throughout the entire temperature range, with an increasing trend observed below −22.0 °C. Overall, our results showed that protein-based biological aerosols contribute the most to $N_{INP}$ at Changbai Mountain.

### 3.3 Source analysis of different types of INPs

We investigated the relationship between different types of INPs and various environmental conditions, as well as the gases and particle compositions, as show in Figure 4 (details can be found in Table S2). In the HTR, our results showed a significant positive correlation ($r$ = 0.5-0.8, $p$ < 0.05) between $N_{INP}$ and WS at temperature ranging between −11.0 °C and −9.0 °C. High WS can enhance the uplift of soil dust and the long-distance transport of aerosols. Moreover, $N_{INP}$ and $Ca^{2+}$ showed a good positive correlation ($r$ = 0.6-0.9) between −11.0 °C and −8.0 °C, leading us to speculate that soil dust may play an important role in ice nucleation in this temperature range. The correlations between $N_{INP}$ and both $Ca^{2+}$ and WS were more pronounced during the daytime (Figure S4), potentially linked to the evolution of PBL and the presence of valley breeze (refer to details in section 3.4). Previous studies have shown that when soil dusts mix with biological components, their freezing temperatures can increase to as high as −6 °C, which is much higher than that of natural dust (below −20 °C) (Hill et al. 2016; O'sullivan et al., 2014). In the LTR, $N_{INP}$ demonstrated a significant positive correlation with ambient air temperature ($r$ = 0.5-0.6, $p$ < 0.05), but showed a significant negative correlation with RH ($r$ = 0.6-0.7, $p$ < 0.05). We further examined the $N_{INP}$ under various weather conditions, including sunny, foggy, and rainy days, which characterized by

significantly different RH levels. As shown in Figure S5, it was observed that on rainy days, the concentrations of INPs were slightly lower compared to the other two weather types in the LTR. This phenomenon may be attributed to the wet deposition of coarse particles (Olszowski, 2016; Kumar et al., 2020). When the temperature falls below -20.0 °C, $N_{INP}$ exhibits a significant positive correlation with $PM_{2.5}$ and BC, implying that inorganic components may serve as active INPs in lower freezing temperature.

We also investigated the potential sources of different types of INPs, as shown in Figure 4(b–d). Similar to the total INP concentrations, $N_{INP-bio}$ were more abundant during the high freezing temperature and low RH. A significant positive correlation was found between $N_{INP-bio}$ with WS ($r = 0.5$-$0.7$) and $Ca^{2+}$ ($r = 0.6$-$0.9$) within a temperature range of $-11$ °C to $-8$ °C. Additionally, a good but not significant positive correlation emerged between $N_{INP-bio}$ and isoprene ($r = 0.6$-$0.7$, $p > 0.05$) along with its oxidation products (isoprene $\times O_3$, $r = 0.7$, $p > 0.05$). O'sullivan et al. (2016) and Augustin-Bauditz et al. (2016) previously reported that biological materials may attach to or mix with dust particles, thus promoting the formation of INPs. However, no mineral dust events were observed during our sampling period, based on the low mass concentrations of $PM_{2.5}$ (the range from 1.5 µg m$^{-3}$ to 31.6 µg m$^{-3}$ with average of $9.3 \pm 0.4$ µg m$^{-3}$) and $Ca^{2+}$ (the range from 0.007 µg m$^{-3}$ to 3.6 µg m$^{-3}$ with average of $0.5 \pm 0.2$ µg m$^{-3}$). We speculate that the higher WS may have facilitated the exposure of the local soil dust and bioaerosol containing bio-INPs to the air. Alternatively, the long-distance transport of biological aerosol attached to soil dust surfaces may also contribute to bio-INPs, leading to the high $N_{INP}$ accompanied by high WS and $Ca^{2+}$.

The positive correlation was more obvious between $N_{INP-other\ org}$ and isoprene, especially at temperatures ranging from $-16.0$ °C to $-14.0$ °C ($r = 0.7$-$0.8$, $p < 0.05$). It is considered to be an important natural gaseous precursor to the formation of secondary organic aerosol (Carlton et al., 2009). Additionally, $N_{INP-other\ org}$ was positively correlated with the oxidation of isoprene bio-INPs ($r = 0.5$-$0.6$) at temperatures ranging from $-18.0$ °C to $-14.0$ °C. Although the oxidation products of isoprene are expected to be water-soluble and unable to induce immersion freezing, the observed positive correlation suggests a potential role of secondary organic compounds associate with vegetation or other biogenic sources in ice nucleation.

In the temperature range of $-23.0$ °C to $-17.0$ °C within the LTR, $N_{INP-inorg}$ exhibited a significant negative correlation with RH ($r = 0.5$-$0.7$, $p < 0.05$), indicating an enrichment of inorg-INPs under low RH conditions. $N_{INP-inorg}$ showed a significant positive correlation with BC and SNA (sulfate, nitrate andammonium) in the LTR. Previous studies have suggested that carbonaceous particles may not act as efficient INPs in the immersion mode or could decrease ice nucleation activity in polluted urban environments, which is attribute to the formation of organic coatings (Kanji et al., 2020; Schill et al., 2020; Nichman et al., 2019; Hammer et al., 2018). However, our observations revealed a positive correlation between BC and both $N_{INP}$ and $N_{INP-inorg}$ in the LTR. The discrepancy may be attributed to the different sources and aging degrees of BC, which remains unclear. Note that $PM_{2.5}$ chemical composition was used in this study, which may lead to uncertainties in the interpretation of the INPs in TSP.

## 3.4 Transport pathways of INPs

At the mountaintop site, the horizontal and vertical transport of air mass are important pathways for INPs under favorable conditions, such as valley breezes, variations in mixing layer height, and long-range transport processes (Chow et al., 2013; Wieder et al., 2022). Understanding the coupling between the PBL changes and the air mass transport process can help us comprehend the characteristics of the target aerosols. Therefore, we conducted further analysis to examine the relationship between the PBL height and $N_{INP}$, and combined it with CWT analysis to explore the effect of transport on $N_{INP}$

at the sampling site.

     At Changbai Mountain, changes in the PBL are also complicated by a variety of processes, such as orographic gravity waves, moist convection, and turbulent transport. Figure 5(a, b) shows the relationship between bio-INPs and the PBL height during the daytime. We found a positive correlation between PBL height and $N_{INP\text{-}bio}$ in the freezing temperature ranging from −25.0 °C to −15.0 °C ($r = 0.4$-$0.8$), especially at temperatures ranging between −22.0 °C and −19.5 °C ($r > 0.7$,

$p < 0.05$). However, the correlation was no longer observed at the freezing temperature above −15 °C ($r < 0.5$, $p > 0.05$). Notably, this correlation increased in the HTR when we excluded two outliers with exceptionally high $N_{INP\text{-}bio}$ values (as shown in Figure 5a, $r$ increased to 0.77, $p < 0.05$). The two high values may be related to ocean and vegetation emissions, and they will be further discussed below. In brief, our findings suggest that an increase in the PBL height may cause a corresponding increase in $N_{INP\text{-}bio}$ in the clean mountaintop atmosphere. Moreover, based on the analysis of the air mass

backward trajectory, our analysis revealed a significant elevation in the height of the trajectory as it moved through the southern mountainous regions, indicative of an upslope valley wind effect (Figure S6d). According to Ketterer et al. (2014), such upslope valley winds have the potential to locally raise the altitude of the PBL, and could even trigger the vertical exchange of PBL air into the free troposphere. In our investigation, the observed positive correlation between the PBL and INPs suggests that valley breezes may facilitate the upward transport of INPs from the foothill to the top of Changbai

Mountain during the daytime.

     The CWT analysis revealed the transport pathways and potential source regions of bio-INPs, as shown in Figure 5(c, d). Ocean was identified as an important INPs source for long-range transport, as previous studies have reported that bubble bursting processes can release marine microorganisms (Burrows et al., 2013; Kwak et al., 2014; Vergara-Temprado et al., 2017). Different ice nucleating entities can trigger droplets to freeze at various temperatures in the marine environment. For

example, Wilson et al. (2015) found that the biogenic organic materials within the sea surface microlayer could induce droplet freezing in the immersion mode, with a broad freezing temperature range of −7.0 °C to −35.0 °C. Laboratory experiments have further revealed that aerosols generated by phytoplankton are particularly effective at triggering ice nucleation at temperatures below −15.0 °C, with a notable increase in INP concentration within the range of −15.0 °C to −23.0 °C, which was related to the unique dynamic processes of phytoplankton bloom and growth (Brooks and Thornton,

2018; Mccluskey et al., 2017; Thornton et al., 2023; Wilbourn et al., 2020). Our study detected the high concentrations of bio-INPs in the LTR originating from the Japan Sea (Figure 5d), implying that the air mass passing over the Japan Sea

surface might have carried marine bio-INPs, contributing to their presence at our sampling site. In contrast, in the HTR, bio-INPs are mainly originate from the southern part of the Korean Peninsula. Previous studies demonstrated that vegetation contains a substantial density of microorganisms ($10^6$-$10^7$ cm$^{-2}$) and serves as a recognized reservoir of highly efficient biological INPs (Moore et al., 2021; Lindow and Brandl, 2003). These bio-INPs typically induce freezing at relatively higher temperatures, ranging from −2 ℃ to −5 ℃ (Schneider et al., 2021; Maki et al., 1974). South Korea has a large vegetation coverage area, as shown in Figure 1(a), with a high potential to emit biological aerosols those of which may be able to reach our sampling site through long-distance transport.

The residence time of various biological particles in the atmosphere can range from less than a day to a few weeks, depending on their size and aerodynamic properties (Despres et al., 2012). The long-range transport of biological aerosols has been observed in previous studies. For example, abundant microbial components originating from the ocean or land have been found in the troposphere, even extending to the stratosphere and the Mesosphere (Burrows et al., 2009; Smith et al., 2013). High concentrations of microbial populations have also been identified in the background atmosphere during trans-Pacific intercontinental transport (Smith et al., 2013). On a global scale, microorganisms have been found to travel thousands of kilometers, with approximately 33%–68% originating in the ocean (Mayol et al., 2017). This suggests that the ocean's bubble bursting processes play a significant role in the generation of biological aerosols. In addition, bio-INPs can attach to dust particles for long-distance transmission, with an adhesion rate that can even exceed 99.9% (Creamean et al., 2013; Yahya et al., 2019). This process can enable biological aerosol to be transported over greater distances, significantly enhancing the ice nucleation activity of dust (O'sullivan et al., 2016; Augustin-Bauditz et al., 2016).

In addition, a positive correlation was found between the PBL height and other org-INPs during the daytime, with significant correlations observed between −18.5 ℃ and −16.5 ℃ ($r > 0.7$, $p < 0.05$), as shown in Figure S7. However, at the freezing temperatures greater than −15.0 ℃, no correlation was observed between the PBL and $N_{\text{INP-other org}}$, suggesting that local sources may be an important source for other org-INPs. For the inorg-INPs, a weak correlation with the PBL height was observed at temperatures greater than −23.0 ℃ and was not statistically significant ($p > 0.05$). However, at −24.5 ℃ and −24.0 ℃, the correlation is more significant ($r$ is 0.73 and 0.80, $p < 0.05$). The CWT simulation also indicated that high values of $N_{\text{INP-other org}}$ and $N_{\text{INP-inorg}}$ appeared in both local areas and adjacent Japan Sea regions (See Figure S8).

In summary, our findings suggest that valley breezes and the long-distance transport of air mass from the Japan Sea influence the abundance of INPs at Changbai Mountain. However, the impact of the PBL and valley breezes on the transport of inorg-INPs was found to be less significant than the contributions of bio-INPs and other org-INPs.

## 4 Conclusion

Measurements of INPs were carried out at the Changbai Mountain in northeastern Asia to explore the abundance and source of INPs in the immersion freezing mode. Our results showed that $N_{\text{INP}}$ spanned up to four orders of magnitude

between $1.6 \times 10^{-3}$ L$^{-1}$ and 78.3 L$^{-1}$ over the freezing temperature range from −5.5 °C to −29.0 °C, with these values corresponding to previously reported measurements for mountain sites.

The observed INPs predominantly comprised protein-based bio-INPs. The fractions of proteinaceous biological $N_{INP}$ ($F_{INP\text{-}bio}$) constituted 100% of $N_{INP}$ above −11 °C, gradually decreasing as the temperature decreased from −11.5 °C to −16.5 °C. Notably, a turning point occurred at −16.5 °C, where $F_{INP\text{-}bio}$ increased from 0.8 to 0.9 as the temperature decreased from −16.5 °C to −21.5 °C, indicating an enrichment of active bio-INPs in the low-temperature region (LTR, freezing temperature below $T_{50}$, −17.0 °C ~ −29.0 °C). When the temperature falls below -22.0 °C, $F_{INP\text{-}bio}$ exhibits a pronounced declining trend. We also found a significant positive correlation between biological INPs and both wind speed (WS) and $Ca^{2+}$, whereas there was only a weak positive correlation for biological INPs with isoprene and its oxidation products (isoprene $\times$ $O_3$). We speculate that the higher WS may facilitate the exposure of the local soil dust and bioaerosols containing bio-INPs to the atmosphere.

Our study also suggests that an increase in the planetary boundary layer (PBL) during the observation period may lead to a corresponding increase of diverse types of $N_{INP}$ in the clean mountaintop atmosphere. During the daytime, valley breezes facilitate the orographic lifting of INPs from the foothill to the top of mountain. However, for the high values of $N_{INP\text{-}bio}$, it may originate from long-distance transport from the Japan Sea and South Korea areas. Our findings suggested that the oceanic and vegetation biogenic aerosols from these areas make significant contributions to the INPs at the top of the Changbai Mountain.

Our measurements in the high-altitude atmosphere above Northeast Asia indicate the predominant role of bio-INPs. However, our study has limitation in terms of dataset size. Further observational and modelling studies employing high-resolution instruments are urgently needed to analyze the characteristics of INPs and their influence on ice crystal formation as well as the cloud properties in the high-altitude atmosphere.

*Author contributions*. Yue Sun analyzed data and wrote the paper. Likun Xue designed the research. Jiangshan Mu, Ye Shan, Mingxuan Liu, Yanbin Qi, Lingli Zhang and Yufei Wang conducted the field campaign. Lanxiadi Chen and Mingjin Tang provided guidance and assistance in the analysis of INPs samples. Yu Yang, Yanqiu Nie, Ping Liu, Can Cui and Ji Zhang helped with the interpretation of the results. Yujiao Zhu, Likun Xue, Xinfeng Wang and Wenxing Wang revised the original manuscript. All authors contributed toward improving the paper.

*Competing interests*. The authors declare that they have no conflict of interest.

*Data availability*. The datasets related to this work can be accessed via https:/ doi.org/10.17632/b9y6pfw39n.2 (Sun et al., 2023).

*Acknowledgements*. This work was funded by the National Natural Science Foundation of China (42075104, 41922051,42061160478). We are grateful to the staff of the Tianchi weather station for their logistical support and assistance during the field observations. We would also like to acknowledge the Global Data Assimilation System (GDAS) provided by the National Oceanic and Atmospheric Administration Air Resources Laboratory (NOAA ARL) for organizing and
445 publishing the data, and the open-source software of MeteoInfo developed by Yaqiang Wang's team for the concentration-weighted trajectory (CWT) analysis.

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

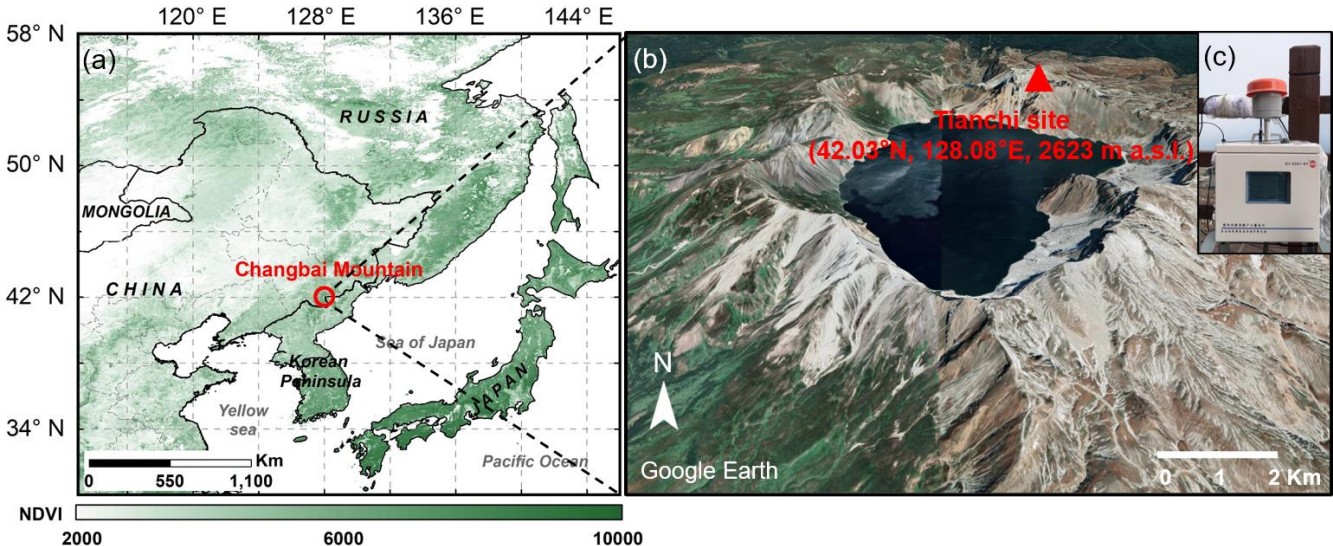

**Figure 1. Geographical maps showing the location of Changbai Mountain. (a) This map is color-coded according to the normalized difference vegetation index (NDVI) in 2015, which was downloaded from the Geospatial Data Cloud (https://www.gscloud.cn/search). (b) Map with the three-dimensional shape of the sampling site, which was obtained from Google Earth.(c)The INP sampler (The TH-150D medium flow sampler, Wuhan Tianhong Corporation, China).**

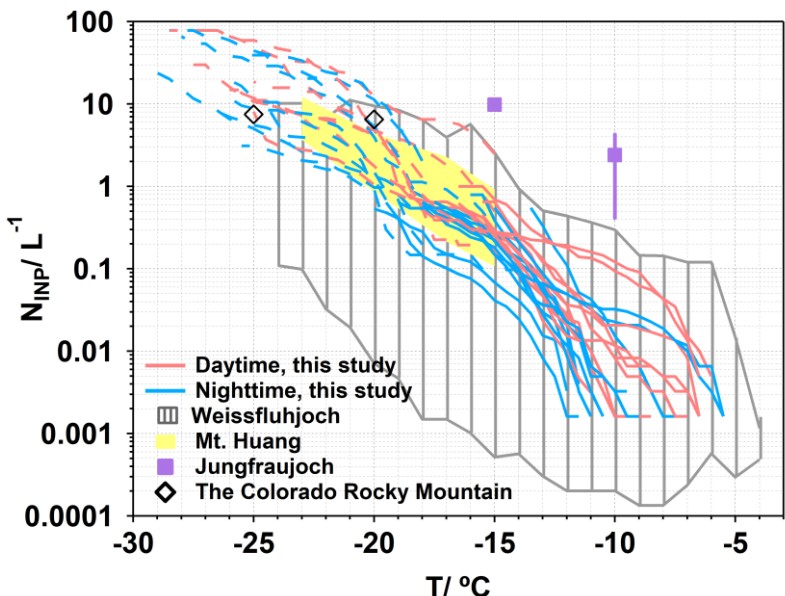

**Figure 2. The concentrations of INPs ($N_{INP}$) as functions of temperature. The dark gray shaded area represents the upper and lower limits of $N_{INP}$ over Weissfluhjoch (2693 m a.s.l.) (Wieder et al., 2022), the yellow shaded area represents the atmospheric $N_{INP}$ ranges at Mt. Huang (1840 m a.s.l) (Jiang et al., 2015), the purple square represents the median $N_{INP}$ at −15 °C and −10 °C in Jungfraujoch (3580 m a.s.l.) (Conen et al., 2022), and the black rhombus represents the median $N_{INP}$ at −25 °C and −20 °C at the Storm Peak Laboratory in the northwestern Colorado Rocky Mountains (3220 m a.s.l.) (Hodshire et al., 2022).**

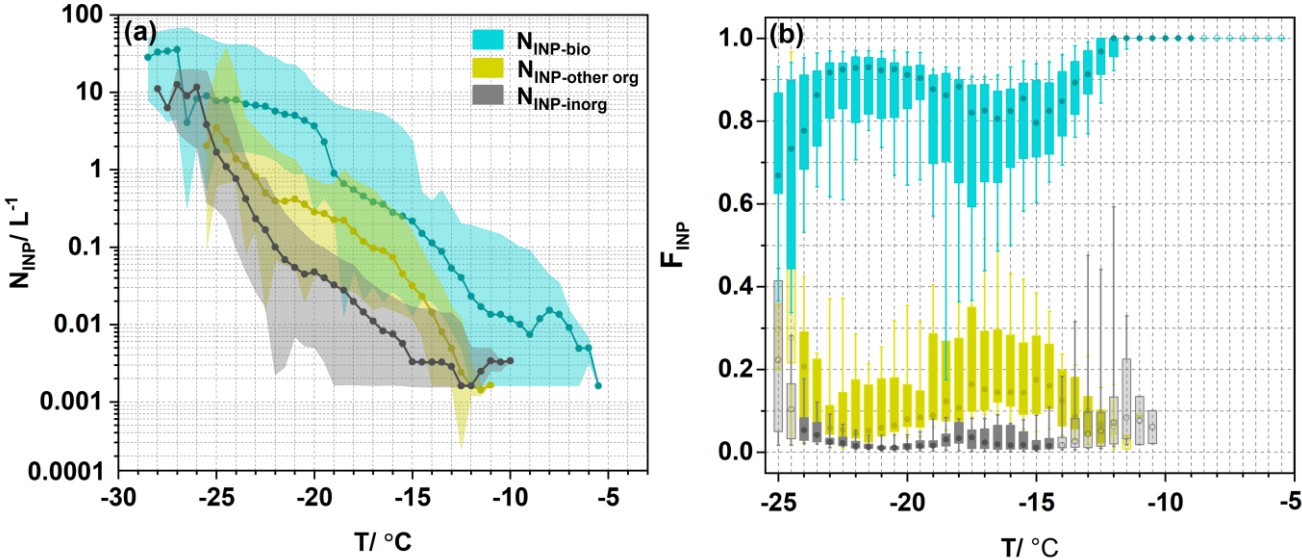

**Figure 3.** $N_{INP}$ **for different types of INPs and their fractions as functions of temperature. (a) The INPs spectra of biological INPs ($N_{INP\text{-}bio}$, blue dots), other organic INPs ($N_{INP\text{-}other\ org}$, yellow dots), and inorganic INPs ($N_{INP\text{-}inorg}$, gray dots). Each point represents the median value and the shadow area represents the maximum and minimum value. (b) Boxplot of fractions of bio-INPs ($F_{INP\text{-}bio}$, blue boxplot), other org-INPs ($F_{INP\text{-}other\ org}$, yellow boxplot), and inorganic INPs ($F_{INP\text{-}inorg}$, gray boxplot) as functions of**
780 **temperature. The upper and lower extents of the boxes represent the 75th and 25th percentiles, respectively, while the whiskers indicate the 10th and 90th values. The circle in each boxplot represents the median value. The light-colored boxes indicate that the number of data points is less than half (the sample number is less than 11) of all samples at each temperature.**

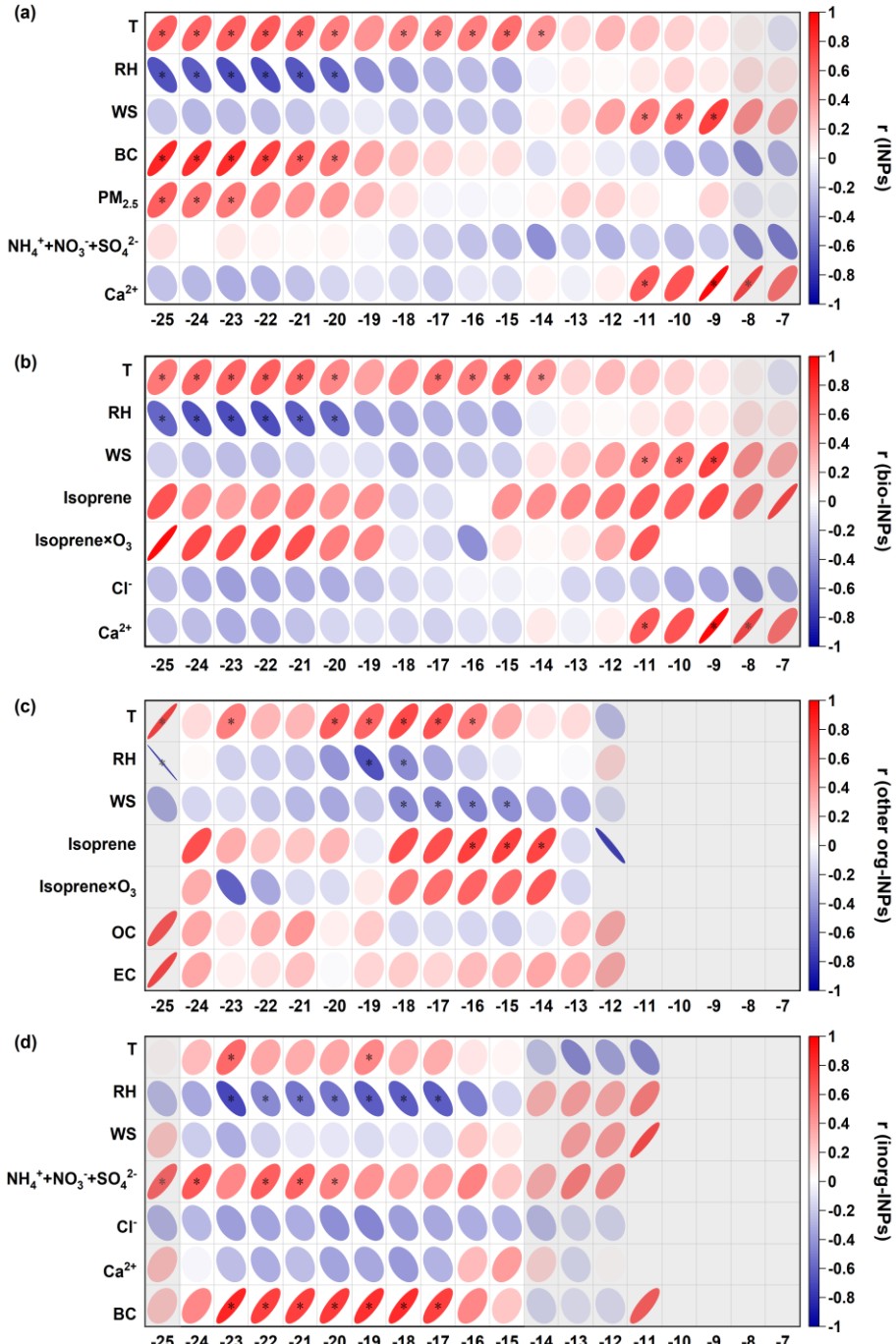

**Figure 4.** Correlation analysis between **(a)** $N_{INP}$, **(b)** $N_{INP-bio}$, **(c)** $N_{INP-other\ org}$, **(d)** $N_{INP-inorg}$, with meteorological parameters and chemical compositions as functions of temperature. The *r* denotes the Pearson correlation coefficients. The asterisk indicates $p < 0.05$, while the shades indicate that the number of data points is less than half of all samples at each temperature.

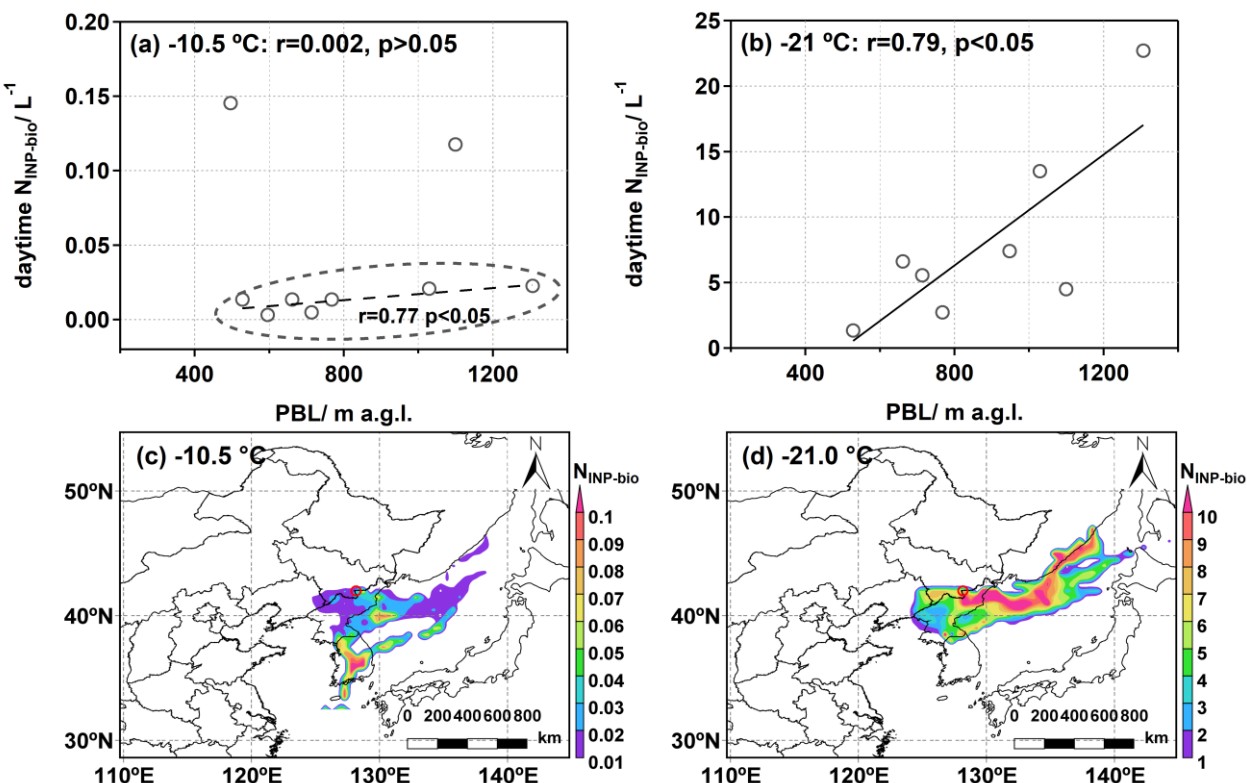

Figure 5. (a-b) Relationship between $N_{INP-bio}$ and average PBL height during the daytime (8:00–17:00 LT) at freezing temperature of −10.5 °C and −21.0 °C. The $r$ denotes the Pearson correlation coefficients. (c-d) The concentration-weighted trajectory (CWT) analysis for the distribution of $N_{INP-bio}$ at −10.5 °C and −21.0 °C during the measurement. The red circle represents the Tianchi site.