# Peer review of "Measurement report: Atmospheric Ice Nuclei at Changbai Mountain (2623 m a.s.l.) in Northeastern Asia"

_EGUsphere, 2023_

## Author Comment (AC1)

**Response to reviewer**

We gratefully thank the reviewer for the constructive comments and suggestions to improve the manuscript. Below are the detailed point-to-point responses to the reviewer's comments. For clarity, the reviewer's comments are listed below in *black italics*, while our responses and changes in the manuscript are shown in blue. The changes in the revised manuscript and supporting materials are also highlighted.

*Anonymous Referee #3*

*General comments*

*The manuscript by Sun et al. reports INP measurements at Changbai Summit in the summer of 2021. Using results from heat and H2O2 treatments, the authors distinguish between biological INPs, other organic INPs, and inorganic INPs and conclude that the majority of INPs in the temperature range studied were of biological origin. It is an interesting study that provides new information about biological INPs. They discuss correlations of biological INPs with meteorological data as well as with some chemical tracers, suggesting that biological INPs may originate from soil dust. In addition, possible sources and transport mechanisms of INPs are discussed based on knowledge of the height of the planetary boundary layer and the concentration-weighted backward trajectories of air masses. The authors have made an effort to explain their data, but given the small number of samples (22 in total, half of which were collected during the day and the other at night) and the lack of sample dilutions to extend the freezing spectra to lower temperatures, some sections require substantial revision in my opinion. I have major and minor comments that I hope will be helpful to the authors and can be addressed during the review process.*

**Response:** We are grateful to the reviewer for the valuable comments. During the observation period, a total of 22 samples were collected. Recognizing the potential for uncertainties due to the limited data size, we have addressed this concern in the revised manuscript. Furthermore, we have conducted dilution of the untreated and heat-treated samples and extended the freezing temperature below -25 ℃. We updated the $N_{INP}$ spectra and corresponding analyses in the revised manuscript. We have endeavored to respond to these valuable comments and revise our manuscript accordingly. Detailed point-to-point responses are shown below.

*Specific comments*

*Abstract*

*1. Line 12: "modifying" is maybe not the right word here.*

**Response:** We change the "modifying" to "modulating" in the revised manuscript.

*2. Lines 22-24: While this insight into atmospheric dynamics is welcome, I would note that this positive correlation between biological INPs with planetary boundary layer height is based on 7 data points only. 2 outliers were excluded and 2 missing data points were not mentioned. It is not stated which correlation analysis was done, something that persists throughout the study. Furthermore, it is unclear for the reviewer how these two cases of high concentrations of biological INPs differ from the other samples in terms of long-range transport. These results need thorough review, discussions about them need to be clarified, and the relevant conclusions probably need to be revised.*

**Response:** In the revised manuscript, we performed dilutions of the samples by the factors of 30, 60, and 120 times, ensuring that all samples reached a freezing temperature of at least -25 ℃. We re-calculated the Pearson correlation coefficients that include all samples, including the two high values. The correlations between PBL height and $N_{INP}$ are presented in Figure R1.

The two missing data corresponded to the blank filters, and we have provided clarification regarding this in the revised manuscript. Nevertheless, the data points ranged from 6 to 11 when calculating the Pearson correlation coefficients, resulting in uncertainties. The uncertainties have been addressed in the revised manuscript. Since the revised correlation analysis included the two cases with high concentrations of biological INPs, we have refrained from adding extensive discussions on these two cases.

[Figure]

Figure R1. The relationship between PBL height and $N_{INP-bio}$, $N_{INP-other org}$ as well as $N_{INP-inorg}$ during daytime (8:00-17:00, m above ground level) as a function of temperature. The r denote the Pearson correlation coefficients. The asterisk indicates p < 0.05. The shades indicate that the data points number were less than half of all samples at each temperature.

Introduction

*3. The reviewer thinks that some parts of the introduction should be worded more precisely.*

**Response:** Thanks for the comment. We have briefly introduced the three aerosol types (biological aerosol, non-proteinaceous biological particles, and non-organic aerosols) in the revised manuscript.

*4. Line 40 and line 52: Please use "etc" carefully. Try to list fully instead.*

**Response:** Thanks for the comment. We have made revisions in the revised manuscript.

*5. Line 40: There are also efficient proteinaceous cell-free INPs (Pummer et al., 2015) that are not embedded in cell membranes. Please, reformulate the sentence.*

**Response:** In revision, we have revised the statement as follows: "It is generally demonstrated that biological aerosols, which have proteins present in cell-free macromolecules or outer membranes of intact cell, are the most efficient INPs at temperatures above −15 ℃ (Phelps et al., 1986; Petters and Wright, 2015; Murray et al., 2012; Kunert et al., 2019; Huang et al., 2021)."

*6. Line 45: I am unsure what is meant by "and [in] agricultural soils". Do you mean a site, where presumably many INPs come from particles from agricultural soils? Please reformulate.*

**Response:** The term "and [in] agricultural soils" means the presence of INPs from regions with agricultural soils. In the revised manuscript, we have replaced "agricultural soils" with "agricultural regions" to clarity.

*7. Line 68: The tree sites that are referred to are not all located in the Swiss Alps, but are located across Switzerland.*

**Response:** Thanks for the comment. We have revised this statement as follows: "For example, in the Switzerland, simultaneous measurements taken at different-altitude stations revealed a reduction of approximately 50% per kilometer in the abundance of INPs in the vertical gradient (ranging from 489 m above sea level (a.s.l.) to 3580 m a.s.l. in the Swiss Alps) in the warm season (Conen et al., 2017)."

*8. Lines 70-71: Instead of "one order of magnitude", you could keep the same units as in the previous sentence. Maybe summarize in percent per kilometer.*

**Response:** In the revised manuscript, we replaced "one order of magnitude" with "exceed 60%".

*9. Line 72: The "atmosphere" can only be in singular.*

**Response:** Thanks for the comment. We have made the corrections in the revised manuscript.

*10. Lines 72-75: Please mention that Schrod et al. 2017 findings were obtained during specific events, i.e., a series of elevated Saharan dust plumes.*

**Response:** In the revision, we have revised as: "In contrast, Schrod et al. (2017) reported an increase in INPs abundance of approximately 10 times over the eastern Mediterranean (2.5 km a.s.l.) relative to ground level using unmanned aircraft systems, with this difference attributed to the long-distance transport of a series of elevated Saharan dust plumes at the height of a few kilometers."

*11. Lines 75-77: The citation Conen et al. 2022 is missing at the end of the sentence. I would suggest adding here the following specification: "under free-tropospheric conditions". Study could have been mentioned in L. 44-45 instead/as well.*

**Response:** Thanks for the comment. We have cited the reference and added the content in the revised manuscript.

*12. Line 87: "INPs measurements" --> "INP measurements"*

**Response:** Thanks for the comment. We have made the corrections in the revised manuscript.

*Methods*

*The reviewer found that the methods are well described, and that it was helpful that the authors provided the raw data for the reviewers.*

*13. Line 119: Please specify that your times are meant in local time or mention the correct time zone.*

**Response:** We used the local time and have added the description in the revised manuscript.

*14. Line 120: A total of 24 samples were collected for INP analysis. The results of only 22 samples were reported in the provided data file. Could the authors please explain why the results of two samples are missing? Could it maybe be that 22 samples and 2 background filters were collected?*

**Response:** Yes, the 24 samples include two background filters. In the revised manuscript, we have modified the statement to: "A total of 24 samples were collected on PCTE filters from August 10 to 24, 2021 in local time (LT), which includes 2 blank filters." In addition, detailed sampling information has been added in Table S1 in the Supporting Information, as presented in Table R1.

Table R1. Detail information including sampling date, duration and total sampling volume in this study.

| Sample Date | Start time | End time | Duration/ min | Total Volume/ L |
|---|---|---|---|---|
| 2021.8.10-Night | 2021/8/10 18:00 | 2021/8/11 5:30 | 294 | 34500 |
| 2021.8.11-Day | 2021/8/11 10:30 | 2021/8/11 17:30 | 168 | 21000 |
| 2021.8.11-Night | 2021/8/11 18:00 | 2021/8/11 20:26 | 242 | 28300 |
| | 2021/8/11 20:32 | 2021/8/12 3:32 | | |
| 2021.8.12-Day | 2021/8/12 6:00 | 2021/8/12 17:30 | 294 | 34500 |
| 2021.8.12-Night | 2021/8/12 18:00 | 2021/8/13 5:30 | 294 | 34500 |
| 2021.8.13-Day | 2021/8/13 6:00 | 2021/8/13 17:30 | 294 | 34500 |
| 2021.8.13-Night | 2021/8/13 18:00 | 2021/8/14 5:30 | 294 | 34500 |
| 2021.8.14-Day | 2021/8/14 6:00 | 2021/8/14 17:30 | 294 | 34500 |
| 2021.8.14-Night | 2021/8/14 18:00 | 2021/8/15 5:30 | 294 | 34500 |
| 2021.8.15-Day | 2021/8/15 6:00 | 2021/8/15 18:00 | 288 | 36000 |
| 2021.8.15-Night | 2021/8/15 18:30 | 2021/8/16 5:30 | 264 | 33000 |
| 2021.8.16-Day | 2021/8/16 6:00 | 2021/8/16 17:30 | 294 | 34500 |
| 2021.8.16-Night | 2021/8/16 18:00 | 2021/8/17 5:30 | 294 | 34500 |
| 2021.8.17-Day | 2021/8/17 6:00 | 2021/8/17 17:30 | 294 | 34500 |
| 2021.8.17-Night | 2021/8/17 18:00 | 2021/8/18 5:30 | 294 | 34500 |
| 2021.8.18-Day | 2021/8/18 10:00 | 2021/8/18 17:30 | 198 | 22500 |
| 2021.8.18-Night | 2021/8/18 18:00 | 2021/8/19 5:30 | 294 | 34500 |
| 2021.8.19-Day | 2021/8/19 6:00 | 2021/8/19 17:30 | 294 | 34500 |
| 2021.8.19-Night | 2021/8/19 18:00 | 2021/8/20 5:30 | 294 | 34500 |
| 2021.8.20-Day | 2021/8/20 6:00 | 2021/8/20 17:30 | 294 | 34500 |
| 2021.8.20-Night | 2021/8/20 18:00 | 2021/8/21 5:30 | 294 | 34500 |
| 2021.8.21-Day | 2021/8/21 6:00 | 2021/8/21 17:30 | 294 | 34500 |
| Blank-Night | 2021/8/23 17:29 | 2021/8/24 5:36 | 295 | - |
| Blank-Day | 2021/8/24 7:30 | 2021/8/24 18:50 | 284 | - |

*15. Line 130: It would be interesting to know how far the weather station is from the measurement site.*

**Response:** We added the distance description in the revised manuscript: "Meteorological data, such as temperature, humidity, WS, wind direction, pressure, and precipitation, were monitored by the Tianchi weather station, a national meteorological station located approximately 20 m from the sampling site."

*16. Lines 178-186: Please be more precise how you generated the air mass backwards trajectories and add information about starting time as well as number of trajectories per sample.*

**Response:** The air mass backward trajectories cover hourly data from 10 August to 21 August, resulting in a total of 264 trajectories generated. These have been added in the revision.

*17. Line 188: Which data product from the Climate Data Store did you use for the PBL data?*

**Response:** The PBL data was obtained from the fifth-generation ECMWF global atmospheric reanalysis (ERA5 reanalysis). We have added this information in the revised manuscript.

*Results and Discussion*

*The reviewer found that some parts of the results and discussion need thorough revision and much clearer statements.*

*18. Line 208 and lines 210-211: Wieder et al. (2022) contains INP concentrations from a mountain site called Weissfluhjoch (2693 m a.s.l.) and a valley site called Wolfgangpass (1631 m a.s.l.). The two sites in the Swiss Alps are only 4 km apart and are not necessarily representative of the entire Swiss Alps. Please be more specific: a) Swiss Alps and not any Alps, b) add the name of the site(s) you mean.*

**Response:** Thanks. We have revised "Alps" to "Weissfluhjoch" in the revision.

*19. Lines 208-211: Please provide the time spans for collection and averaging for these other studies. For example, Wieder et al. (2022) collected samples over short time spans (i.e., 20 minutes) and averaged them into 2-hour bins. They found a peak at 19 h UTC (i.e., 21 h CET), which is defined as nighttime based on the definition used in this manuscript, if I am not mistaken. A possible diurnal cycle at Changbai Mountain may have been averaged out due to the long sampling time of 11 hours. Please mention specifically for Changbai Mountain only the comparison of daytime and nighttime, since only two samples are taken per day. In my opinion, a conclusion over the entire diurnal cycle cannot be made here.*

**Response:** Thanks for the comment. The time spans have been added in the revision. In this

study, our objective is to compare differences in INPs concentration between daytime and nighttime, rather than delve into an extensive analysis of diurnal variations. In the revision, we have added the statement as follow: "Additionally, the limited data size and low sampling frequency may also result in a lack of diurnal variation in this study."

*20. Line 213: What is meant by "the temperature spectra showed a wider range"?*

**Response:** Thanks for the comment. We have removed the statement in the revision.

*21. Line 217: What is meant by "the NINP value was much larger at Mt. Huang than our results"? Can't the difference between both sites be explained by the fact that your results cover a warmer temperature range than the results from Mt. Huang? Please mention the freezing temperatures while comparing INP concentrations.*

**Response:** Based on our updated $N_{INP}$ values after the dilution procedure, we have revised the statement as follows: "Jiang et al. (2015) reported INP concentrations at the top of Mt. Huang, where the temperature spectra of $N_{INP}$ exhibited a much narrower range between -15.0 ℃ ~ -23.0 ℃ compared to those observed at the Weissfluhjoch. The $N_{INP}$ value at Mt. Huang ranged from 0.1 $L^{-1}$ to 11.9 $L^{-1}$, partially overlapping with our results in the temperature range from −15.0 ℃ to −23.0 ℃."

*22. Line 256: Shouldn't it be "bio-INP" instead of "INP"?*

**Response:** Thanks. We have made the corrections in the revised manuscript.

*23. Line 254-256: A mean increase in FINP-bio from 0.8 to 0.9 with decreasing temperatures between -16.5 ℃ and -20 °C doesn't make much sense and could be due to the way the data were processed here. It looks like some untreated and heat-treated samples already reached the upper detection limit at about -15 ℃ or -16 ℃. In order to draw conclusions about the proportion of bio-INPs, other org-INPs, and inorganic INPs below -15 ℃, I believe dilutions of the untreated and heat-treated samples would have been required. The number of data points of FINP-bio from maybe around -18 ℃ and lower is likely too low to draw any conclusions below that temperature threshold.*

**Response:** In the revised manuscript, we diluted the untreated and heat-treated samples by the factors of 30, 60, and 120 times, and extended the freezing temperature below -25 ℃ (details can be found in the response to comment 2). Based on the updated $N_{INP}$ data, we have made revisions to the boxplot, as shown in Figure R2. It can be found that as the temperature decreased from -16.5 ℃ to -21.5 ℃, the median value of $F_{INP-bio}$ increased from 0.8 to 0.9. We are confident that this increase accurately represents the presence of biogenic INPs that exhibit relatively high ice-nucleating activity in the LTR.

[Figure]

Figure R2. Boxplot of fractions of bio-INPs ($F_{\text{INP-bio}}$, blue boxplot), other org-INPs ($F_{\text{INP-other org}}$, yellow boxplot), and inorganic INPs ($F_{\text{INP-inorg}}$, gray boxplot) as functions of temperature. The upper and lower extents of the boxes represent the 75th and 25th percentiles, respectively, while the whiskers indicate the 10th and 90th values. The circle in each boxplot represents the median value. The light-colored boxes indicate that the number of data points is less than half (the sample number is less than 11) of all samples at each temperature.

*24. Lines 277-281: What about biological INPs originating from plants and oceans? Please discuss.*

**Response:** Because $N_{\text{INP}}$ was positively correlated with WS and $Ca^{2+}$, we emphasis the discussion of soil source. Section 3.4 discusses marine and plant sources.

*25. Lines 279-281: Maybe instead of citing the review, authors could cite the original studies e.g. Hill et al. 2016 or others.*

**Response:** Thanks. We have cited the references, Hill et al. 2016 and O'sullivan et al., 2014, in the revision.

*26. Line 301: Maybe instead of citing the review, authors could cite the original studies.*

**Response:** Thanks. We cited the reference, i.e., Cozic et al. (2008) and Levin et al. (2016), in the revision.

*27. Lines 315-321: Could the authors elaborate how the inclusion of these points would change the results and conclusion?*

**Response:** We have update the $N_{\text{INP}}$ data based on the diluted process, and the relationship between $N_{\text{INP-bio}}$ and PBL have revised as shown in Figure R1. In the HTR, there was no correlation between bio-INPs and PBL. However, upon excluding the two high INP cases at -10.5 °C, the remaining seven cases exhibited an increasing trend in bio-INPs as PBL height increased (r=0.77, p<0.05,), as shown in Figure R4a. The two high INP cases may be source

from potential strong but infrequent local sources, which have been added in the revision.

[Figure]

Figure R3. Relationship between $N_{INP\text{-}bio}$ and PBL height during the daytime (8:00–17:00 LT) at freezing temperature of −10.5 °C and −21.0 °C. The r denotes the Pearson correlation coefficients.

*28. Line 320: Please indicate here that the "freezing" temperature is meant.*

**Response:** Thanks. We have made the corrections in the revised manuscript.

*29. Lines 323-325: How does the CWT analysis on August 18 and 25 differ from the other days?*

**Response:** We have conducted a correlation analysis on the entire dataset and did not delve further into the two high values.

Conclusion

*The reviewer believes that some rewording needs to be done, especially in the last paragraph.*

*30. Lines 365-368: A decreasing trend in FINP-bio with decreasing temperature suggests that other organic or inorganic INPs become more important with decreasing temperatures. It does not suggest that the ice nucleation activity of bio-INPs decreases. In addition, due to the lack of data points in the low temperature regime, no conclusions can be drawn regarding FINP-bio for that temperature range, in my opinion. Dilutions would have been required, as mentioned above.*

**Response:** We agree and delete the statement about "suggesting that the ice nucleation activity of bio-INPs decreases with decreasing temperature." We have added data for the low freezing temperature range, confirming the conclusions regarding $F_{bio\text{-}INPs}$. Please see our response to comment 23.

*31. Line 379: Did you mean to write that the INP number concentration is lower than the ice crystal number concentration, especially in the warm temperature range?*

**Response:** Considering the last paragraph was not closely aligned with the context of this paper, we have removed it in the revision.

32. Line 381: Is there a word missing between "…is second only to…"?

**Response:** Considering the last paragraph was not closely aligned with the context of this paper, we have removed it in the revision.

*33. Line 383: Please delete "through various collisions with pre-existing ice". There are several multiplication processes, and they involve many more processes than only ice-ice collision. More information can be found for example in Korolev and Leisner, 2020.*

**Response:** Considering the last paragraph was not closely aligned with the context of this paper, we have removed it in the revision.

*34. Line 385: Bio-INPs do not impact secondary ice formation directly, please reformulate.*

**Response:** Considering the last paragraph was not closely aligned with the context of this paper, we have removed it in the revision.

*Figures*

*Fig.2(b)*

*35. Please indicate in the figure and legend the name of the mountain in the Swiss Alps from which the data originate similar to what you did with the Mt. Huang dataset.*

**Response:** We have replaced the name to "Weissfluhjoch".

*36. Should the y-axis perhaps be extended to lower values to see the lower limit of the error bar?*

**Response:** We have modified the y-axis to display all the data. But to ensure that the information in the figure can be readily grasped, we did not display the error bar in Figure 2b. The error bar of whole samples was shown in Figure S3, as illustrated in Figure R4.

[Figure]

Figure R4. The $N_{INP}$, $N_{INP-heat}$, and $N_{INP-H2O2}$ as function of temperature. The solid line and dotted line show the sample measurement result by immersed in 5 mL MilliQ water and diluted the sample 30-120 times, respectively. The original $N_{INP}$ is marked by black dots, $N_{INP-heat}$ is marked by purple dots, and $N_{INP-H2O2}$ is marked by pink dots, with 20% error bars indicating the 95% confidence intervals.

*Fig. 3(a)*

*37. Why are the error bars for NINP-H2O2 below -15 ℃ extending to values outside the graph?*

**Response:** We adjusted the y-axis range to display all error bars.

*38. Please use the same axes in Fig. 3a than in Fig. 3b and Fig. 2b.*

**Response**: We have revised the manuscript accordingly.

*Fig.3(c)*

*39. I appreciate that there is an indication of the number of data points included in the analysis. Since the upper detection limit of about 2 INPs per liter is reached around -15 ℃ for NINP-bio, dilutions would have been necessary to extend the plot from -15 to -20 ℃ as mentioned above.*

**Response**: In the revised manuscript, we extended the freezing temperature below -25 ℃ by diluting the samples.

*40. How can the median fraction of inorganic INPs above -12 ℃ be more than 0.0 if the*

*median fraction of bio-INPs is 1.0?*

**Response**: We first calculated the daily proportions and then calculated the average the proportion across all samples.

*Fig.4*

*41. What kind of correlation analysis did you do? Please specify.*

**Response**: We calculated the Pearson correlation coefficients, which has been added in the revised manuscript.

*Fig.5(a-c)*

*42. What does "r" stand for? Please also describe what the red lines and the purple circles represent.*

**Response**: The r denotes the Pearson correlation coefficients. We have described the manuscript accordingly. The purple circles represented the exclusion of two high-value data points, and the red line indicated their correlation. However, as mentioned previously, we have updated the data and no longer exclude high values. Instead, we have performed a correlation analysis on the entire dataset. Therefore, revisions have been made, as illustrated in Figure R3.

*43. Is the x-axis showing the mean PBL height?*

**Response**: Yes, the x-axis showed the mean PBL height. We have revised the figure caption to "average PBL height during the daytime".

*44. Maybe add "freezing" between "three" and "temperatures".*

**Response**: We have revised the manuscript accordingly.

*Fig.5 (d-f)*

*45. Trajectories of which samples were included in the CWT analysis? Are the trajectories of all the samples included or only those of the daytime samples? Were the trajectories of the two outliers excluded or included here? Please be more specific.*

**Response**: All samples were included in the CWT analysis. we have revised the figure caption to "The concentration-weighted trajectory (CWT) analysis for the distribution of $N_{INP-bio}$ at −10.5 °C and −21.0 °C during the measurement."

*46. Also describe in the legend what the star and triangle represent.*

**Response**: We have revised the manuscript accordingly.

*Fig. S1*

*47. Please describe in the legend precisely what is shown and what is meant by "R". Is the large "R" here similar to the small "r" in Fig. 5?*

**Response**: The "R" here was same as the "r" in Figure 5. We used "r" consistently throughout the revised manuscript.

*Fig. S2*

*48. Authors could add the star and the triangle in these maps, similar to Fig. 5 (d-f).*

**Response**: We have revised the manuscript accordingly.

**Reference:**

Conen, F., Yakutin, M. V., Yttri, K. E., and Hüglin, C.: Ice Nucleating Particle Concentrations Increase When Leaves Fall in Autumn, Atmosphere-Basel, 8, 202, https://doi.org/10.3390/atmos8100202, 2017.

Cozic, J., Mertes, S., Verheggen, B., Cziczo, D. J., Gallavardin, S. J., Walter, S., Baltensperger, U., and Weingartner, E.: Black carbon enrichment in atmospheric ice particle residuals observed in lower tropospheric mixed phase clouds, J Geophys Res-Atmos, 113, 11, https://doi.org/10.1029/2007jd009266, 2008.

Hill, T. C. J., DeMott, P. J., Tobo, Y., Fröhlich-Nowoisky, J., Moffett, B. F., Franc, G. D., and Kreidenweis, S. M.: Sources of organic ice nucleating particles in soils, Atmos. Chem. Phys., 16, 7195-7211, 10.5194/acp-16-7195-2016, 2016.

Huang, S., Hu, W., Chen, J., Wu, Z., Zhang, D., and Fu, P.: Overview of biological ice nucleating particles in the atmosphere, Environment International, 146, 106197, https://doi.org/10.1016/j.envint.2020.106197, 2021.

Jiang, H., Yin, Y., Su, H., Shan, Y. P., and Gao, R. J.: The characteristics of atmospheric ice nuclei measured at the top of Huangshan (the Yellow Mountains) in Southeast China using a newly built static vacuum water vapor diffusion chamber, Atmos Res, 153, 200-208, https://doi.org/10.1016/j.atmosres.2014.08.015, 2015.

Kunert, A. T., Pöhlker, M. L., Tang, K., Krevert, C. S., Wieder, C., Speth, K. R., Hanson, L. E., Morris, C. E., Schmale Iii, D. G., Pöschl, U., and Fröhlich-Nowoisky, J.: Macromolecular fungal ice nuclei in Fusarium: effects of physical and chemical processing, Biogeosciences, 16, 4647-4659, https://doi.org/10.5194/bg-16-4647-2019, 2019.

Levin, E. J. T., McMeeking, G. R., DeMott, P. J., McCluskey, C. S., Carrico, C. M., Nakao, S., Jayarathne, T., Stone, E. A., Stockwell, C. E., Yokelson, R. J., and Kreidenweis, S. M.: Ice-nucleating particle emissions from biomass combustion and the potential

importance of soot aerosol, J Geophys Res-Atmos, 121, 5888-5903, https://doi.org/10.1002/2016jd024879, 2016.

Murray, B. J., O'Sullivan, D., Atkinson, J. D., and Webb, M. E.: Ice nucleation by particles immersed in supercooled cloud droplets, Chem. Soc. Rev., 41, 6519-6554, https://doi.org/10.1039/c2cs35200a, 2012.

O'Sullivan, D., Murray, B. J., Malkin, T. L., Whale, T. F., Umo, N. S., Atkinson, J. D., Price, H. C., Baustian, K. J., Browse, J., and Webb, M. E.: Ice nucleation by fertile soil dusts: relative importance of mineral and biogenic components, Atmos. Chem. Phys., 14, 1853-1867, https://doi.org/10.5194/acp-14-1853-2014, 2014.

Petters, M. D. and Wright, T. P.: Revisiting ice nucleation from precipitation samples, Geophys Res Lett, 42, 8758-8766, https://doi.org/10.1002/2015GL065733, 2015.

Phelps, P., Giddings, T. H., Prochoda, M., and Fall, R.: Release of cell-free ice nuclei by Erwinia herbicola, Journal of Bacteriology, 167, 496-502, https://doi.org/10.1128/jb.167.2.496-502.1986, 1986.

Schrod, J., Weber, D., Drucke, J., Keleshis, C., Pikridas, M., Ebert, M., Cvetkovic, B., Nickovic, S., Marinou, E., Baars, H., Ansmann, A., Vrekoussis, M., Mihalopoulos, N., Sciare, J., Curtius, J., and Bingemer, H. G.: Ice nucleating particles over the Eastern Mediterranean measured by unmanned aircraft systems, Atmos Chem Phys, 17, 4817-4835, https://doi.org/10.5194/acp-17-4817-2017, 2017.

---

## Author Comment (AC2)

**Response to reviewer**

We gratefully thank the reviewer for the constructive comments and suggestions to improve the manuscript. Below are the detailed point-to-point responses to the reviewer's comments. For clarity, the reviewer's comments are listed below in *black italics*, while our responses and changes in the manuscript are shown in blue. The changes in the revised manuscript and supporting materials are also highlighted.

*Anonymous Referee #1*

*Review of Sun et al. 2023: Measurement report: Atmospheric Ice Nuclei at Changbai Mountain (2623 m a.s.l.) in Northeastern Asia*

*Sun et al. 2023 present INP measurement results from the Changbai mountain in summer 2021. Changes in INP concentration are investigated towards diurnal variability, composition, source, and transport mechanism. It is an interesting study given the location of the measurement site, methodology and efforts taken in the analysis. However, major adjustments are needed to streamline and support the claims of the manuscript.*

**Response:** We are grateful to the reviewer for the comments and have endeavored to respond to these and revise our manuscript accordingly.

*General Comments:*

*Major doubts concern the influence of the PBL height to Changbai mountain in Section 3.4. To the reviewer the analysis and interpretation are inconclusive. Especially it is not entirely traceable how the PBL height was derived and the involved error. Given the points raised below, the reviewer is not entirely sure if some claims need to be removed if not being support by further evidence from in-situ observations. In addition, the local wind system and dynamics should be better described or referenced.*

*Overall, it appears that frequently there are statements in the manuscript that should be supported by a reference. The authors may want to consider this aspect, while working through the manuscript. In addition, the figures of the manuscript could also be linked to the made statements more often. Furthermore, there appear some side information at some point, which not necessarily add to the flow of the manuscript. The authors may want to critically read through the paper, deciding which information is needed to reach the presented conclusions.*

*Some strong statements are made without reference or presenting data. For additional support of some of the made claims particle size measurements may be helpful. Have there not measurements been available for the study?*

*Changbai mountain is referred as 2623 m a.s.l. (Tianchi site) and 2740 m a.s.l. (highest point). To avoid confusion the two locations and heights should be consistently appear combined.*

*For deeper explanations, see specific comments below.*

**Response:** Thanks for the comment. In this study, the PBL data was obtained from the fifth-generation ECMWF global atmospheric reanalysis (ERA5 reanalysis). The reliability of ERA5 dataset has been substantiated in previous studies, such as Le et al. (2020), Tornow et al. (2021), and Slattberg et al. (2022). We made more specified statements regarding the source and data reliability of the PBL in the following response. Further details regarding the source and the robustness of the PBL data can be found in the subsequent responses.

We calculated the 72-hour backward trajectories, and found that prevailing air masses predominantly approached the sampling site from the east. However, the local winds exhibited a prevailing pattern from the west and south. The Changbai Mountains feature a topography characterized by higher elevations in the southeast and lower elevations in the northwest, and our observation site is located in the northwest direction. We inferred that air masses arriving from the east encounter obstruction by the Changbai Mountains, resulting in a lifting along the southern to western slopes. Combined with the trajectory heights of the air masses, it is evident that as the air mass approached the observation sites, their trajectories inclined upward along the southern or southwestern mountainsides. These findings suggested that the air masses underwent a noticeable lifting process prior to reaching the sampling site, potentially attributed to orographic lifting along the mountain slopes in the south and westward directions. We have added this information into the revised manuscript to provide a clearer understanding of the wind dynamics in this region.

We have added essential references to provide more comprehensive support for some of the statements. Furthermore, we have improved the integration of figure citations within the context. To enhance the overall flow of the article, we have removed the discussion of secondary ice formation from the conclusion section.

We apologize for the absence of parallel size measurements to distinguish particle chemical concentrations between particles larger than 2.5 μm and those smaller than 2.5 μm. This may result in some uncertainties in the investigation of the sources of INPs based on $PM_{2.5}$ chemical composition. These uncertainties have been addressed in the revised manuscript.

Our sampling site is situated at an elevation of 2623 m a.s.l. on Changbai Mountain. Notably, Lu et al. (2016) collected rainwater samples at the peak of Changbai mountain, which stands at an elevation of 2740 m a.s.l. We have clarified this in the revised manuscript.

Moreover, we diluted the untreated and heat-treated samples by the factors of 30, 60, and 120 times, and extended the freezing temperature below -25°C. We have updated the $N_{INP}$ spectra

and analysis in the whole revised manuscript.

Detailed point-to-point responses are shown below.

*Specific Comments:*

*Introduction*

*1. L29: 'As most precipitation in clouds initiates via the ice phase', this statement could be refined in regard of which cloud types and regions are affected and direct studies could be cited.*

**Response:** Thanks for the comment. We have revised this statement as follows: "Global precipitation is predominantly produced by clouds containing the ice phase, especially in continental regions and mid-latitude oceans, emphasizing the paramount significance of investigating ice formation within clouds (Mulmenstadt et al., 2015; Lau and Wu, 2003; Demott et al., 2010; Kanji et al., 2017)."

*2. L63: 'At present, it is unclear whether…', in the reviewer's opinion the current knowledge gap is not whether but to which extend and through which transport pathway INPs are brought to MPC relevant heights.*

**Response:** We agree and have therefore revised the statement as follows: "At present, there remain uncertainties how INPs can be extended and transported to the altitudes of mixed-phase cloud formation (approximately 3–7 km)."

*3. L68: One station in Conen et al. 2017 was located outside the Swiss Alps.*

**Response:** Thanks for the comment. We have revised the statement as follows: "For example, in the Switzerland, simultaneous measurements taken at different-altitude stations revealed a reduction of approximately 50% per kilometer in the abundance of INPs in the vertical gradient (ranging from 489 m above sea level (a.s.l.) to 3580 m a.s.l. in the Swiss Alps) in the warm season (Conen et al., 2017)."

*4. L71: Wieder et al. 2022 sampled frequently for short time spans throughout the day, whereas Conen et al. 2017 used filters sampling over longer timespans. Wieder et al. 2022 observed also a diurnal cycle with INP concentrations seeming to equilibrate over the course of a day. Further it may be important to point out to the reader that this was only observed in a wind direction where the topography promoted vertical transport of air masses from lower elevation.*

**Response:** Thanks for the comment. We have added "Note that variations in sampling methods and the influence of wind directions can also exert an impact on INP concentrations." in the revised manuscript.

*5. L75-77: "For example, at the Jungfraujoch station (3580 m a.s.l.) in the Swiss Alps, approximately 80% of INPs were biological aerosols at freezing temperatures above −15 °C" – add reference.*

**Response:** We apologize for missing this reference and cited it in the revised manuscript (Conen et al., 2022).

*6. L81-82: "… and establishing a parametric equation that depends on temperature and ice supersaturation for predicting the INPs concentration" – this information seems irrelevant for the current study.*

**Response:** We agree and have removed it in the revised manuscript.

*7. L83: Here, Changbai mountain is attributed with a height of 2740 m a.s.l., whereas in L87 a height of 2623 m a.s.l. is given. If understood correct, the latter height refers to the INP measurement station of the current study. To avoid confusion the distinction between the peak and the measurement station should be made (also throughout the manuscript).*

**Response:** Yes, the 2740 m a.s.l. in Line 83 was the peak height of Changbai mountain, and the 2623 m a.s.l. in Line 87 was the height of our sampling site. To avoid confusion, we have revised as "Changbai Mountain (at the peak of 2740 m a.s.l.)".

Methods

*8. L102: Shouldn't the elevations around the mountain decrease to all directions?*

**Response:** The elevations gradient around the mountain generally decreases to all directions. However, we aimed to describe the comprehensive pattern of height changes across the entire mountain, which is higher elevation in the southeast compared to the northwest directions. Furthermore, we have included the appropriate reference citation in the revised manuscript (Wang et al., 2014).

*9. L104: Is 150m correct? Looking at maps it seems like more.*

**Response:** Thanks. We carefully checked the distance from our observation site to Tianchi Lake, and revised the distance to 410 m in the manuscript.

*10. L106-112: Adding a timeseries plot (appendix or supplement) of the general meteorological parameters throughout the campaign including exemplary INP concentration would be helpful to understand the sampling conditions.*

**Response:** Figure R1 has been added to the Supporting Information and mentioned in section 2.1 as follows: "Figure S1 presents the timeseries of meteorological parameter (i.e. wind,

temperature, and RH), NO$_x$ concentration, and the concentration of INPs at -12°C, -17°C and -21°C, as measured during the field campaign."

In addition, detailed sampling information has been added in Table S1 in the Supporting Information, as presented in Table R1.

[Figure]

Figure R1. Time series of meteorological parameter (i.e., wind, temperature, and RH), the height of PBL and Tianchi site (above ground level), NO$_x$ and the cumulative number concentration of INPs ($N_{INP}$) at -12 °C, -17 °C and -21 °C measured during the campaign.

Table R1. Detail information including sampling date, duration and total sampling volume in this study.

| Sample Date | Start time | End time | Duration/ min | Total Volume/ L |
|---|---|---|---|---|
| 2021.8.10-Night | 2021/8/10 18:00 | 2021/8/11 5:30 | 294 | 34500 |
| 2021.8.11-Day | 2021/8/11 10:30 | 2021/8/11 17:30 | 168 | 21000 |
| 2021.8.11-Night | 2021/8/11 18:00
2021/8/11 20:32 | 2021/8/11 20:26
2021/8/12 3:32 | 242 | 28300 |
| 2021.8.12-Day | 2021/8/12 6:00 | 2021/8/12 17:30 | 294 | 34500 |
| 2021.8.12-Night | 2021/8/12 18:00 | 2021/8/13 5:30 | 294 | 34500 |
| 2021.8.13-Day | 2021/8/13 6:00 | 2021/8/13 17:30 | 294 | 34500 |
| 2021.8.13-Night | 2021/8/13 18:00 | 2021/8/14 5:30 | 294 | 34500 |
| 2021.8.14-Day | 2021/8/14 6:00 | 2021/8/14 17:30 | 294 | 34500 |
| 2021.8.14-Night | 2021/8/14 18:00 | 2021/8/15 5:30 | 294 | 34500 |
| 2021.8.15-Day | 2021/8/15 6:00 | 2021/8/15 18:00 | 288 | 36000 |
| 2021.8.15-Night | 2021/8/15 18:30 | 2021/8/16 5:30 | 264 | 33000 |
| 2021.8.16-Day | 2021/8/16 6:00 | 2021/8/16 17:30 | 294 | 34500 |
| 2021.8.16-Night | 2021/8/16 18:00 | 2021/8/17 5:30 | 294 | 34500 |
| 2021.8.17-Day | 2021/8/17 6:00 | 2021/8/17 17:30 | 294 | 34500 |
| 2021.8.17-Night | 2021/8/17 18:00 | 2021/8/18 5:30 | 294 | 34500 |
| 2021.8.18-Day | 2021/8/18 10:00 | 2021/8/18 17:30 | 198 | 22500 |
| 2021.8.18-Night | 2021/8/18 18:00 | 2021/8/19 5:30 | 294 | 34500 |
| 2021.8.19-Day | 2021/8/19 6:00 | 2021/8/19 17:30 | 294 | 34500 |
| 2021.8.19-Night | 2021/8/19 18:00 | 2021/8/20 5:30 | 294 | 34500 |
| 2021.8.20-Day | 2021/8/20 6:00 | 2021/8/20 17:30 | 294 | 34500 |
| 2021.8.20-Night | 2021/8/20 18:00 | 2021/8/21 5:30 | 294 | 34500 |
| 2021.8.21-Day | 2021/8/21 6:00 | 2021/8/21 17:30 | 294 | 34500 |
| Blank-Night | 2021/8/23 17:29 | 2021/8/24 5:36 | 295 | - |
| Blank-Day | 2021/8/24 7:30 | 2021/8/24 18:50 | 284 | - |

*11. L112: What about touristic activities?*

**Response:** We have added the touristic activities in the revised manuscript: "Changbai Mountain is a national nature reserve with no large industrial facilities nearby, and tourism is the important economic activity in the region. Due to the emergence of novel coronavirus (COVID-19) cases, strict lockdown measures have been implemented from August 10, 2021, resulting in a substantial reduction in visitor numbers, as indicated by the marked decrease in NOx concentration (Figure S1)."

*12. L114: How was differentiated whether the sampling site was in the free troposphere or influenced by the PBL? Please elaborate and add data or reference.*

**Response:** In Figure R1 (and also in Figure S1), we have included PBL data represented by the red line, while the blue dashed line corresponds to the height of the sampling site. Throughout the observation period, the sampling site experiences alternations between the free troposphere and the boundary layer due to changes in the PBL.

*13. L116: A dedicated reader may be interested in a picture of the setup (also possibly in the appendix or supplement).*

**Response:** We added the picture of the TH-150D medium flow sampler in Figure 1 in the revised manuscript, as show in Figure R2b.

[Figure]

Figure R2. Geographical maps showing the location of Changbai Mountain. (a) This map is color-coded according to the normalized difference vegetation index (NDVI) in 2015, which was downloaded from the Geospatial Data Cloud (https://www.gscloud.cn/search). (b) This map shows the three-dimensional shape of the sampling site, which was obtained from Google Earth. (c) The ice nuclei sampler (The TH-150D medium flow sampler, Wuhan Tianhong Corporation, China).

*14. L117: Was there any pretreatment of the PCTE filters?*

**Response:** We did not pretreat the PCTE filters prior to sampling.

*15. L117-124: For clarity, maybe specify what each filter sample type was used for, i.e. PCTE for INP analysis, PM2.5 for chemical composition.*

**Response:** Thanks. We have added specific descriptions in the revised manuscript.

*16.L120: There were 24 samples taken covering roughly 30% of the entire measurement campaign. The authors may want to comment on the underlying measurement strategy and providing more concrete information under which meteorological conditions samples were taken.*

**Response:** Thanks for the comments. In this study, a total of 24 samples were collected on PCTE filters from August 10 to 24, 2021, which included 2 blank filters. Notably, our observation period coincided with a significant reduction in human activities following the strict lockdown measures enforced after August 10 (refer to response to comment 10). As demonstrated in Figure R1 (and also in Figure S1), the concentration of NOx decreased markedly from 3.0±2.1 ppb during July 24 to August 9 to 0.9±0.3 ppb between August 10 and August 24. This effectively minimizes the influence of human activities on the collection of INPs samples.

*17. L120: The chemical analysis was conducted on the samples collected using a PM2.5 inlet. Can the authors estimate the number of particles larger than 2.5 μm which contributed to the INP analysis samples, but would not be covered in the chemical analysis? Has there been any parallel size measurements supporting that claim?*

**Response:** We apologized that we did not collect the INPs at different stages and we had no parallel size measurement to differentiate the particle chemical concentrations between particles larger than 2.5 μm and those smaller than 2.5 μm. This may result in some uncertainties in the investigation of the sources of INPs based on $PM_{2.5}$ chemical composition. These uncertainties have been addressed in the revised manuscript.

*18. L130: For consistency, specify the type of weather station.*

**Response:** The Tianchi weather station is an institution under the Changbai Mountain Meteorological Bureau and is affiliated with the National Meteorological Station. We added the information in the revised manuscript: "Meteorological data, such as temperature, humidity, WS, wind direction, pressure, and precipitation, were monitored by the Tianchi weather station, a national meteorological station located approximately 20 m from the sampling site."

*19. L153: above -> at*

**Response:** Thanks, we had revised the manuscript accordingly.

*20. L158: What was the value of Vair?*

**Response:** $V_{air}$ ranged from $2.1 \times 10^4$ to $3.6 \times 10^4$ L during the sampling period. Details of the total sampling volume was shown in Table R1.

*21. L159: If the INP concentrations were given in standard liters, the reviewer would advise to indicate this by using e.g., sL-1, or stdL-1 in Equation 3 and corresponding INP data plots.*

**Response:** In the explanation of the formula for NINP calculation, we described the computations conducted in standard liters. Consistent with the approach used in Chen et al. (2018; 2021), we intend to retain this explanation in the main text while excluding it from the INP data plots.

*22. L178: Which PBL data product was used? The reviewer could not find a unique record. Furthermore, what is the uncertainty and sensitivity of the PBL data product? Is Changbai mountain centered in a grid box that data was taken from or were different adjacent grid boxes averaged? Given the complex terrain of the mountainous region, how reliable does a 25km x 20km grid box represent the PBL height? Generally, the reviewer is a bit skeptical of the representativeness for the presented application. The authors should elaborate why this data is applicable. In addition, are there any direct meteorological (especially wind) observations along the mountain slope that would support the later claim of vertical transport due to orographic lifting?*

**Responds:** The PBL data was obtained from the fifth-generation ECMWF global atmospheric reanalysis (ERA5 reanalysis). This dataset has been widely used in numberous studies, such as Le et al. (2020), Tornow et al. (2021), and Slattberg et al. (2022). Guo et al. (2021) conducted a comparative analysis of ERA5 reanalysis products against other widely used products, i.e., MERRA-2, JRA-55, and NCEP-2. The results showed that the ERA5 exhibited the smallest bias. Therefore, we have confidence in the reliability of the PBL data sourced from ERA5 for our analysis.

In the input meteorological dataset, specifically the Global Data Analysis System (GDAS) data, Changbai Mountain's terrain height is recorded at 1656 m. In our simulation, we utilize a trajectory ending height of 967 m (above ground level), to achieve a sampling station elevation of 2623 m.

As show in Figure R3b, the 72-hour backward trajectories showed that prevailing air masses predominantly approached the sampling site from the east. However, local winds exhibited a prevailing pattern from the west and south. As elucidated in the manuscript, the Changbai Mountains exhibit a topography characterized by a southeastern high and a northwestern low. The observation site is situated in the northwest of Tianchi Lake. Air masses arriving from the east encounter obstruction by the Changbai Mountains, resulting in a lifting along the southern to western slopes. Figure R3c showed the trajectory heights of the air masses during

the daytime. It is evident that as they approached the observation sites, their trajectories inclined upward along the southern or southwestern mountainsides (Figure R3c). This suggests that the air masses underwent a noticeable lifting process prior to reaching the sampling site, potentially attributed to orographic lifting along the mountain slopes in the south and westward directions.

Air masses arriving from the east encounter obstruction by the Changbai Mountains, resulting in upward displacement along the southern to western slopes. Figure R3c illustrates the trajectory heights of these air masses during daytime. It is evident that as they approached the observation site, their trajectories exhibited an upward trend along the southern or southwestern mountainsides (Figure R3c). This suggests that the air masses underwent notably lifting process prior to reaching the sampling site, likely attributed to orographic lifting along the mountain slopes in the south and west directions.

[Figure]

Figure R3. Wind rose illustrating one-minute wind speed and directions measured at the sampling site (a). Air mass trajectories over the entire campaign duration (b) and the average daytime air mass trajectory heights (c).

*23. L184: The authors may want to briefly describe the principle of CWT.*

**Responds: W**e have added the description of CWT in Section 2.5 as follows:
"The CWT assigns the average weighted concentration by trajectories were divided into grids. The calculation was used Equation 5 according to the method of Hsu et al. (2003):

$$C_{ij} = \frac{1}{\sum_{k=1}^{M} \tau_{ijk}} \sum_{k=1}^{M} C_k \tau_{ijk}, \tag{5}$$

where $C_{ij}$ is the average weighted concentration in the $ij$ cell, $k$ is the index of the trajectory, $M$ is the total number of trajectories, $C_k$ is the concentration observed on arrival of trajectory $k$ in the $ij$ cell, and $\tau_{ijk}$ is the time spent in the $ij$ cell by trajectory. The weight function $W_{ij}$ was also applied to the CWT analysis to reduce the uncertainty in the cells with small values of $n_{ij}$:

$$WCWT_{ij} = C_{ij} \times W(n_{ij}), \tag{6}"$$

*Results and Discussion*

*24. L190: "characterize situation of droplet freezing" sounds odd.*

**Responds:** In the revised manuscript, we have revised this sentence to make it more readable: "A metric was applied to compare the droplet freezing results, i.e., the freezing temperature at which 50% of the droplets are frozen ($T_{50}$)."

*25. L192: As there are only two MilliQ water backgrounds displayed, rephrase to "were - 30°C and -28.5°C".*

**Response:** We conducted measurements on more than two MilliQ water samples, as displayed by the purple lines in Figure R4.

[Figure]

Figure R4. Frozen fractions ($f_{ice}$) as functions of temperature. The $f_{ice}$ of collected samples measured by GIGINA is shown by the black curves, and presented together with blank filters (orange curves) and MilliQ water (purple curves) as background signals.

*26. L193-196: This seems contradicting. If the contaminants could be ignored, one would not need to correct for it.*

**Response:** We agree and therefore revised the statement as follows: "For the two blank filters, $T_{50}$ was averaged to −24.2 ± 2.12 °C, which was slightly higher than that of MilliQ water, but much lower than that of the collected samples (for which $T_{50}$ was −17.0 ± 4.1 °C), indicating the presence of minimal contaminants stemming from the filter membrane.

*27. L195-196: Were averaged concentrations at each temperature step of the two blank filter samples subtracted as correction – please specify. As two samples are not many, the authors may want to comment on the overall obtained repeatability of blank filter measurements obtained which are not presented in this study.*

**Response:** We collected blank filters during both daytime and nighttime, as listed in Table R1. The sampling duration lasted for 11 hours, and we believe that the two blank samples could adequately represents the background values of the filters. During the calculation process, we applied corrections by subtracting the values obtained from the two blank filter samples at each freezing temperature.

*28. L199: -26.0°C -> -20°C*

**Response:** Thank you for the comment. In the revised manuscript, we have updated the temperature value from "-26.0°C" to "-29°C." This modification was made because we diluted the samples to obtain the $N_{INP}$ spectra at lower freezing temperature.

*29. L200: What it the values of T50? -13°C from above? The authors should repeat this value here, or introduce another variable to avoid the nomenclature confusion of T50 representing the result of one sample (L191) or the average of all samples (L194).*

**Response:** Thank you for the comment. The $T_{50}$ was -17°C based on the full freezing temperature spectra from $-29.0$ °C to $-5.5$ °C. We have made the corrections in the revised manuscript.

*30. L201: It is not necessarily the diversity of different INPs but could also just relate to different emission strengths.*

**Response:** Thanks for the comment. We have removed the statement in the revision.

*31. L202: What would that local source be?*

**Response:** We apologize for the vague wording. In the revision, it has been modified to read as follows: "Some of the $N_{INP}$ curves exhibited bumps in the HTR, which has been previously reported at a coastal site (the Cape Verde Atmospheric Observatory, Africa) in air samples by Welti et al. (2018), as well as in the upper bound of the composite nucleus spectrum of cloud water and precipitation samples by Petters and Wright (2015). Welti et al. (2018) reported that when the IN properties are narrower, the steeper slope can be observed in a temperature spectrum".

*32. L205-206: This statement cannot be made, as it seems that there have no dilutions of samples been made. This sets the upper limit in detectable NINP = 1/Vair i.e., all droplets frozen.*

**Response:** In the revised manuscript, we extended the freezing temperature below -25°C by diluting the samples. Firstly, we re-measured INP concentrations for the original samples, and found that the concentration of $N_{INP}$, $N_{INP-heat}$, and $N_{INP-H2O2}$ were basically consistent between the latest measurements and previously recorded concentrations (Figure R5). Subsequently, we diluted the suspension liquid by the factors of 30, 60, and 120 times, ensuring that all samples reached a freezing temperature of at least -25°C. Consequently, we have updated the freezing temperature spectra of $N_{INP}$ in the revised manuscript.

Based on the updated data, we are confident in this statement. Here, we have modified the statement in the revised manuscript: "In contrast, in the low-temperature region (LTR, freezing temperature below $T_{50}$, -17.0 °C ~ -29.0 °C), $N_{INP}$ showed a relatively narrow

variation than LTR, from 0.1 L$^{-1}$ to 78.3 L$^{-1}$.”

[Figure]

Figure R5. Comparison of frozen fractions ($f_{ice}$) as functions of temperature between the latest measurements and previously recorded concentrations. The blue and red lines represent the experiment conducted on January 2022 (previous experiment) and September 2023 (latest experiment), respectively.

*33. L206: To the reviewer, the temperature dependence results especially from the fact that NINP is cumulative. A direct connection to the complexity of sources cannot be made.*

**Response:** We agree and have removed the statement in the revision.

*34. L207: What test was used to check for differences between daytime and nighttime samples?*

**Response:** We conducted an independent t-test, yielding a significance level is 0.61 which is higher than 0.05. Therefore, we concluded that there are no significant differences in N$_{INP}$ between the daytime and nighttime samples.

*35. L207-211: It remains questionable if the sampling intervals used in the study (maximally two per day, 20 samples total spread out unknown over a month) allow for the detection of a diurnal cycle. It is conceivable that if, e.g., the minimum in INP concentration occurs at 6:00 and the maximum at 18:00, no difference would be observed on the filters. The described scenario could be likely if the transport is facilitated by convection. In addition, are there any potential local sources located on Changbai mountain, e.g., from the lake, which could cover up a potential diurnal cycle? The authors should elaborate on these aspects.*

**Response:** Our objective is to compare differences in INPs concentration between daytime and nighttime, rather than delve into an extensive analysis of diurnal variations. Factors such as changes in valley breezes, the evolution of the PBL, photochemical reactions, and other variables may exert distinct effects on INPs concentration during these two time periods. Additionally, local sources such as vegetation and the lake could also influence INPs concentration, with biogenic emissions potentially differing between daytime and nighttime.

We agree that our dataset was limited in size, resulting in uncertainties when comparing INPs concentrations between daytime and nighttime. These uncertainties have been addressed in the revised manuscript.

*36. L212-226: In the following discussion, references to the data figure could be beneficial. Furthermore, when discussing the obtained results to previous studies, more precision is needed, what is the difference between "narrowly" (L214) and "much narrower" (L216)? More quantitative expressions are needed. In addition, the present study motivated the need for measurements on high altitude sites. It is not entirely clear, how the sites of Cerro Mirador and Beijing relate to this. Maybe an extra motivation for this comparison would be needed. Studies from high altitude stations like Storm Peak (US), Jungfraujoch (Switzerland), or Altzomoni (Mexico) might be interesting additions, which could also complement Figure 2b, but the reviewer leaves this up to the authors.*

**Response:** In the revised manuscript, we have included temperature range data to provide a more precise explanation of the terms "narrow" and "much narrower."

In the revised Figure 2b, we have excluded the $N_{INP}$ data for Cerro Mirador and Beijing, and added the $N_{INP}$ data for Jungfraujoch and Colorado Rocky Mountains. The corresponding discussions have been added as follows: "In the LTR, our results were comparable with the measurements conducted at the Storm Peak Laboratory in the northwestern Colorado Rocky Mountains (Hodshire et al., 2022). However, in the HTR, $N_{INP}$ measured in Switzerland were approximately 1-3 orders of magnitude higher than in our study. The high concentration of $N_{INP}$ was primarily attributed to the aerosolized epiphytic microorganisms, which contributed most of the INPs to primary ice formation in Switzerland (Conen et al., 2022)."

*37. L213: Precise to Swiss Alps.*

**Response:** Thanks for the comment. We have made the corrections in the revised manuscript.

*38. L214: Shouldn't it be high-temperature and low-concentration region and vice versa?*

**Response:** Thanks for the comment. We have revised the statement as follows: "In our observations, the spectra range of $N_{INP}$ were narrowly located in the relatively high-concentration regions."

*39. Section 3.2: The reviewer perceived the usage of the abbreviations bio-INPs, other org-INPs, and inorg-INPs at times a bit odd. If the authors want to keep them, they may want to consider introducing them already in the introduction.*

**Response:** Thanks for the comment. In the introduction, we provided a brief overview of the categories of bio-INPs, other org-INPs, and inorg-INPs to facilitate a better understanding for readers in following sections.

40. L236: Specify type of H2O2

**Response:** Thanks for the comment. We have added this information in the revised

manuscript.

*41. L254-256: Was this increase significant and isn't it rather a bias due to the measurement limitations (no dilutions)?*

**Response:** We have updated Figure 3b based on the dilution procedure, and have significantly expanded the dataset at lower temperatures below -25°C. Figure R6 illustrated the revised boxplot depicting fractions of biological INPs, other organic INPs, and inorganic INPs. It can be found that as the temperature decreased from -16.5°C to -21.5°C, the median value of $F_{INP-bio}$ increased from 0.8 to 0.9. We are confident that this increase reflects the presence of biological INPs exhibiting relatively high ice-nucleating activity in the LTR.

[Figure]

Figure R6. Boxplot of fractions of bio-INPs ($F_{INP-bio}$, blue boxplot), other org-INPs ($F_{INP-other\ org}$, yellow boxplot), and inorganic INPs ($F_{INP-inorg}$, gray boxplot) as functions of temperature. The upper and lower extents of the boxes represent the 75th and 25th percentiles, respectively, while the whiskers indicate the 10th and 90th values. The circle in each boxplot represents the median value. The light-colored boxes indicate that the number of data points is less than half (the sample number is less than 11) of all samples at each temperature.

*42. L264: Add the actual number of samples that make up 50% (also in the caption of Fig. 3).*

**Response:** Thanks for the comment. We have added the actual number of samples in the revised manuscript.

*43. L269-270: Again, how significant was this increase and is the decrease to low temperatures owed to a measurement bias?*

**Response:** Please see our response to Comment 41.

*44. Section 3.3: The data used in this section is unavailable. As the raw data for creating Figure 4 is already quite digested in Figure 4, the authors may want to add a table with the data written out per sample in the appendix. In addition, the found correlations are described as "good" or "weak". As the definitions for these terms may vary, the value and type of used*

*correlation should be named throughout the manuscript.*

**Response:** Thanks for the comment. We have added a table to show the Pearson correlation coefficients in the supplementary materials, as shown in Table R3. In the table caption, we provided a description of the correlations as follows: "When r is below 0.5, the correlation is considered weak; when r exceeds 0.5, the correlation is considered good." In addition, we have added the r and p values in the revised manuscript.

Table R3. The Pearson correlation coefficients (r) between (a) $N_{INP}$, (b) $N_{INP\text{-}bio}$, (c) $N_{INP\text{-}other\ org}$, (d) $N_{INP\text{-}inorg}$ and meteorological parameters, chemical compositions, as functions of temperature. Coefficients reported in bold are statistically significant at $p < 0.05$, while the shades indicate that the number of data points is less than half (the sample number is less than 11) of all samples at each temperature. When r is below 0.5, the correlation is considered weak; when r exceeds 0.5, the correlation is considered good.

| | -25 | -24 | -23 | -22 | -21 | -20 | -19 | -18 | -17 | -16 | -15 | -14 | -13 | -12 | -11 | -10 | -9 | -8 | -7 |
|---|---|---|---|---|---|---|---|---|---|---|---|---|---|---|---|---|---|---|---|
| **(a)** | | | | | | | | | | | | | | | | | | | |
| T | **0.63** | **0.61** | **0.63** | **0.64** | **0.60** | **0.51** | 0.43 | **0.47** | **0.49** | **0.52** | **0.58** | **0.43** | 0.16 | 0.28 | 0.25 | 0.19 | 0.12 | 0.08 | -0.14 |
| RH | **-0.66** | **-0.63** | **-0.69** | **-0.69** | **-0.65** | **-0.6** | -0.44 | -0.37 | -0.28 | -0.26 | -0.31 | -0.04 | 0.07 | 0.03 | 0.08 | 0.16 | 0.09 | 0.17 | 0.14 |
| WS | -0.22 | -0.28 | -0.24 | -0.25 | -0.2 | -0.12 | -0.08 | -0.2 | -0.23 | -0.22 | -0.23 | 0.04 | 0.20 | 0.36 | **0.52** | **0.57** | **0.74** | 0.57 | 0.44 |
| BC | **0.84** | **0.8** | **0.83** | **0.76** | **0.63** | **0.53** | 0.34 | 0.23 | 0.16 | 0.1 | 0.13 | -0.11 | 0.07 | -0.06 | -0.14 | -0.32 | -0.28 | -0.54 | -0.37 |
| PM$_{2.5}$ | **0.63** | **0.56** | **0.54** | 0.47 | 0.43 | 0.42 | 0.27 | 0.11 | -0.02 | -0.04 | -0.01 | 0.04 | 0.19 | 0.17 | 0.07 | 0 | 0.17 | -0.1 | -0.04 |
| NH$_4^+$+NO$_3^-$ +SO$_4^{2-}$ | 0.13 | 0.01 | 0.09 | 0.05 | 0.02 | 0.05 | -0.01 | -0.14 | -0.18 | -0.22 | -0.26 | -0.43 | -0.20 | -0.29 | -0.18 | -0.24 | -0.19 | -0.57 | -0.64 |
| Ca$^{2+}$ | -0.22 | -0.27 | -0.31 | -0.30 | -0.24 | -0.14 | -0.09 | -0.13 | -0.19 | -0.09 | -0.14 | 0.05 | -0.05 | 0.08 | **0.64** | 0.67 | **0.94** | **0.93** | 0.71 |
| **(b)** | | | | | | | | | | | | | | | | | | | |
| T | **0.53** | **0.59** | **0.62** | **0.64** | **0.59** | **0.48** | 0.37 | 0.48 | **0.54** | **0.5** | **0.57** | **0.44** | 0.16 | 0.27 | 0.25 | 0.19 | 0.12 | 0.08 | -0.14 |
| RH | **-0.58** | **-0.68** | **-0.68** | **-0.69** | **-0.64** | **-0.57** | -0.37 | -0.34 | -0.28 | -0.27 | -0.31 | -0.05 | 0.06 | 0.04 | 0.08 | 0.16 | 0.09 | 0.17 | 0.14 |
| WS | -0.17 | -0.23 | -0.24 | -0.24 | -0.19 | -0.1 | -0.1 | -0.28 | -0.25 | -0.21 | -0.2 | 0.11 | 0.2 | 0.37 | **0.52** | **0.57** | **0.74** | 0.57 | 0.44 |
| Isoprene | 0.66 | 0.45 | 0.37 | 0.45 | 0.5 | 0.42 | 0.44 | -0.14 | -0.13 | 0.01 | 0.43 | 0.44 | 0.49 | 0.53 | 0.63 | 0.61 | 0.7 | 0.66 | 0.94 |
| Isoprene | 0.95 | 0.71 | 0.7 | 0.71 | 0.69 | 0.51 | 0.47 | - | - | - | 0.13 | 0.03 | 0.09 | 0.33 | 0.65 | - | - | - | - |

| | | | | | | | | | | | | | | | | | | | |
|---|---|---|---|---|---|---|---|---|---|---|---|---|---|---|---|---|---|---|---|
| ×O₃ | | | | | | | | | 0.09 | 0.16 | 0.44 | | | | | | | | |
| Cl⁻ | -0.24 | -0.32 | -0.38 | -0.35 | -0.31 | -0.31 | -0.22 | -0.15 | -0.12 | -0.03 | -0.05 | -0.01 | -0.15 | -0.19 | 0.22 | 0.31 | 0.32 | -0.51 | -0.43 |
| Ca²⁺ | -0.23 | -0.25 | -0.32 | -0.31 | -0.24 | -0.15 | -0.12 | -0.15 | -0.18 | -0.11 | -0.13 | 0.09 | -0.04 | 0.08 | **0.64** | 0.67 | **0.94** | **0.93** | 0.71 |
| (c) | | | | | | | | | | | | | | | | | | | |
| T | **0.96** | 0.15 | **0.51** | 0.29 | 0.28 | **0.62** | **0.61** | **0.7** | **0.68** | **0.51** | 0.32 | 0.1 | 0.16 | -0.3 | - | - | - | - | - |
| RH | - | 0.02 | -0.17 | -0.18 | -0.23 | -0.42 | **-0.67** | **-0.44** | -0.33 | -0.18 | -0.04 | 0.01 | -0.01 | 0.22 | - | - | - | - | - |
| WS | -0.4 | -0.15 | -0.12 | -0.21 | -0.27 | -0.33 | -0.21 | **-0.45** | **-0.46** | **-0.48** | **-0.42** | -0.32 | -0.3 | -0.17 | - | - | - | - | - |
| Isoprene | - | 0.68 | 0.33 | 0.24 | 0.24 | 0.28 | -0.07 | 0.69 | 0.7 | **0.76** | **0.76** | **0.73** | -0.14 | -0.96 | - | - | - | - | - |
| Isoprene ×O₃ | - | 0.32 | -0.6 | -0.33 | -0.13 | -0.13 | 0.09 | 0.54 | 0.58 | 0.61 | 0.59 | 0.64 | -0.15 | - | - | - | - | - | - |
| OC | 0.85 | 0.35 | 0.12 | 0.32 | 0.42 | 0.07 | 0.2 | -0.15 | -0.12 | -0.16 | -0.18 | -0.06 | 0.28 | 0.43 | - | - | - | - | - |
| EC | 0.93 | 0.36 | 0.07 | 0.13 | 0.25 | 0 | 0.18 | 0.22 | 0.16 | 0.27 | 0.28 | 0.35 | 0.31 | 0.43 | - | - | - | - | - |
| (d) | | | | | | | | | | | | | | | | | | | |
| T | 0.06 | 0.26 | **0.59** | 0.34 | 0.34 | 0.36 | **0.47** | 0.32 | 0.34 | 0.1 | 0.04 | -0.28 | -0.59 | -0.45 | -0.56 | - | - | - | - |
| RH | -0.33 | -0.32 | **-0.7** | **-0.46** | **-0.52** | **-0.53** | **-0.62** | **-0.63** | **-0.64** | -0.5 | -0.16 | 0.38 | 0.48 | 0.42 | 0.65 | - | - | - | - |
| WS | 0.28 | -0.19 | -0.31 | -0.16 | -0.08 | -0.1 | -0.13 | -0.09 | -0.14 | 0.24 | 0.08 | 0.01 | 0.47 | 0.49 | 0.91 | - | - | - | - |
| NH₄⁺+NO₃⁻ +SO₄²⁻ | **0.78** | **0.65** | 0.46 | **0.63** | **0.62** | **0.49** | 0.43 | 0.34 | 0.36 | 0.49 | 0.23 | 0.41 | 0.64 | 0.56 | - | - | - | - | - |
| Cl⁻ | -0.36 | -0.27 | -0.37 | -0.35 | -0.3 | -0.43 | -0.46 | -0.38 | -0.32 | -0.31 | -0.3 | -0.32 | -0.19 | -0.18 | - | - | - | - | - |
| Ca²⁺ | 0.34 | -0.03 | -0.26 | -0.31 | -0.24 | -0.35 | -0.32 | -0.39 | -0.29 | 0.27 | 0.39 | 0.21 | -0.18 | 0.03 | - | - | - | - | - |

| BC | 0.28 | 0.46 | **0.85** | **0.74** | **0.74** | **0.78** | **0.8** | **0.83** | **0.74** | 0.48 | 0.24 | -0.2 | -0.14 | -0.16 | 0.82 | - | - | - | - |

*45. L278-279: Even though the authors selected a mild formulation for their interpretation, this statement should be toned down a little bit more for having only found a good correlation with one element. Furthermore, could there be other sources as well?*

**Response:** We have revised the statement as follows: "Moreover, $N_{INP}$ and $Ca^{2+}$ showed a good positive correlation ($r$ = 0.6-0.9) in the HTR within the range of -11.0 ℃ to -9.0 ℃, leading us to speculate that soil dust may play an important role in ice nucleation in this temperature range."

In addition to $Ca^{2+}$, we did not conduct the analysis of other ions originating from soil dust.

*46. L281: Kanji et al. 2017 is a summary, maybe use direct study for this reference.*

**Response:** Thanks. We have cited the references, i.e., Hill et al. (2016) and O'sullivan et al., (2014), in the revision.

*47. L288-289: Is there reference for this methodology? What was the used threshold in concentration to come to this conclusion?*

**Response:** The dust event is defined as the day when the peak $PM_{10}$ concentration exceeds 150 µg m$^{-3}$ and the $PM_{2.5}/PM_{10}$ ratio falls below 0.4, in accordance with previous studies (Wu et al., 2020; Liu et al., 2006). During our sampling period, $PM_{2.5}$ concentrations ranged from 1.5 µg m$^{-3}$ to 31.6 µg m$^{-3}$, with an average of 9.3±6.0 µg m$^{-3}$. Notably, these values remained significantly lower than the established threshold.

*48. L306-309: Despite being textbook knowledge, the authors may want to give a reference to read up in these topics. Furthermore, a short description of the essential processes for Changbai mountain could be given in the introduction.*

**Response:** We have cited the references, i.e., Chow et al. (2013) and Wieder et al. (2022) in the revised manuscript.

In the method section, we have added a description of the meteorological conditions as follows: "Changbai Mountain is situated within the westerly wind belt and experiences a typical temperate continental mountain climate influenced by the monsoon, characterized by long cold winters and short temperate summers. The prevailing winds in this region are the westerly and northwesterly winds in the spring, autumn, and winter seasons, and the southeasterly and southwesterly winds in the summer season (Zhao et al., 2015)."

*49. L313: Specify "moderate-to-good".*

**Response:** We have added the r values in the revision.

*50. L315: "exceptionally high NINP-bio values" add e.g., 'as discussed below' for*

*readability.*

**Response:** Thanks for the comment. We have made revisions in the revised manuscript.

*51. L315-316: Even though being only a suggestion, in the reviewer's opinion this statement cannot be made given too many assumptions and misinterpretation. First, the two high INP cases are excluded and given the argument should coincide with height. Is that an indication for a potential strong but infrequent local source? This might well be pure coincidence, but given the dates being a week apart, was there some periodic event near the measurement site? Second, while the correlation (which type of coefficient?) for some temperature is comparably large, there does not seem to be a significant increase in INP concentration. In addition, if the transport of bio-INP was the underlaying process, one could expect that the correlation should be expressed at a broad range of INP concentrations at a wide range of temperature in the HTR – has this been observed and could the correlation at all temperature been shown, e.g., in a table? Lastly, the PBL height never extends to the mountain top. For transport there is further evidence needed like wind speed and direction along the slope to support this claim. Ultimately, the analysis also bases only on 6-9 datapoints and the uncertainty in the PBL product remains undiscussed.*

**Response:** In the revision, we extended the freezing temperatures below -25°C by diluting the samples, resulting a larger dataset for $N_{INP}$, especially in the low-temperature region (LTR). Therefore, we re-calculated the correlation that include all samples, without excluding the two high values. The correlation between PBL height and $N_{INP}$ is illustrated in Figure R7. It is evident that the Pearson correlation coefficient between bio-INPs and PBL in the LTR has notably increased compared to the initial calculation. However, in the HTR, there was no correlation between bio-INPs and PBL. For example, Figure R8 (also Figure 5ab in the revision) showed the relationship between bio-INPs and PBL at temperatures at -10.5°C and -21°C. During daytime sampling, bio-INPs and PBL exhibited a good positive correlation at -21°C, but showed no correlation at -10.5°C. However, upon excluding the two high INP cases at-10.5°C, the remaining seven cases exhibited an increasing trend in bio-INPs as PBL height increased (r=0.77, p<0.05). The two high INP cases may be source from potential strong but infrequent local sources, which have been added in the revision.

The backward trajectory in Figure R3c showed that the air mass underwent a noticeable lifting process prior to reaching the sampling site, which could be associated with orographic lifting along the mountain slopes in the south and westward directions. We agree the presence of uncertainties in both the simulation of air mass backward trajectory and the determination of PBL height. These uncertainties have been added in the revised manuscript.

[Figure]

Figure R7. The relationship between PBL height and $N_{INP\text{-}bio}$, $N_{INP\text{-}other\,org}$ as well as $N_{INP\text{-}inorg}$ during daytime (8:00-17:00, m above ground level) as a function of temperature. The r denote the Pearson correlation coefficients. The asterisk indicates $p < 0.05$. The shades indicate that the data points number were less than half of all samples at each temperature.

[Figure]

Figure R8. Relationship between $N_{INP\text{-}bio}$ and PBL height during the daytime (8:00–17:00 LT) at freezing temperature of −10.5 °C and −21.0 °C. The r denotes the Pearson correlation coefficients.

*52. L316-319: For this claim, data should be presented.*

**Response:** We have provided data in the supplementary materials as Figure S6, as shown in Figure R3c.

*53. L328: Add reference for phytoplankton blooms occurrence.*

**Response:** We have conducted a correlation analysis on the entire dataset and did not delve further into the origin of the two high values.

*54. L348: Were there still enough datapoints available below*

**Response:** After the dilution experiment, there were enough datapoints below -19.5℃.

*55. L357-358: Following Figure S1, it appears to the reviewer, that the correlations for other*

*org-INPs seem similarly or even more consistent throughout the temperature range than for the presented bio-INPs data.*

**Response:** After the dilution experiment, we updated the $N_{INP}$ values and revised Figure S1 (as shown in Figure R7. Generally, bio-INPs and PBL showed good correlations (r=0.4-0.8) in the LTR within the range of -25.0 ℃ to -15.0 ℃, while other org-INPs and PBL showed good correlations (r=0.5-0.8) within the range of -23.0℃ to -20.0 ℃ and -19.0 ℃ to -15.5 ℃. Detail description can be found in the revised manuscript.

Conclusion

*56. L364-371: Maybe repeat introduced variables such as FINP-bio, LTR, WS.*

**Response:** We have added an explanation to the abbreviation in the revised manuscript.

*57. L377-378: "With larger contributions observed from local and oceanic sources" - was this shown in the results section?*

**Response:** The conclusion has been included in the revised Section 3.4.

58. L379-389: The reviewer is unsure whether diving in to the topic of secondary ice formation in the last paragraph without any prior mention of the topic is within the scope of this publication.

**Response:** The last paragraph has been removed in the revision.

*59. L384: "confirm" seems to be the wrong word, the statement should be weakened.*

**Response:** We have changed the word to "indicate".

Figures

*60. Figure 1: Is it essential to indicate Beijing? If so, it should be named in the caption. To stay consistent between (b) and (a) it could be beneficial to indicate Changbai mountain (check spelling in figure) in red in (a).*

**Response:** We removed the labeling for Beijing and adjusted the text color for the stations in (a) and (b), as shown in Figure R2.

*61. Figure 2a: What is the error in T and f?*

**Response:** The error of $f_{ice}$ was 0.002-1.0. To ensure the readability of the information in the figure, we did not display the error bars in Figure 2a.

*62. Figure 2b: Specify the data of Wieder et al. 2022 to Weissfluhjoch instead of Alps. The*

*data of Wieder et al. 2022 is not fully visible. Beijing data is hard to read. For better comparability, maybe an average or median of each data set could be added.*

**Response:** We have revised the site names and adjusted the concentration display range on the spectra. The data from Beijing has been removed, and data from the high-altitude stations have been added into Figure 2b, as shown in Figure R9.

[Figure]

Figure R9. The concentrations of INPs ($N_{INP}$) as functions of temperature. The dark gray shaded area represents the upper and lower limits of $N_{INP}$ over the Weissfluhjoch (2693 m a.s.l.) (Wieder et al., 2022). The yellow shaded area represents the atmospheric $N_{INP}$ ranges at Mt. Huang (1840 m a.s.l) (Jiang et al., 2015). The purple square represents the median $N_{INP}$ at −15 °C and −10 °C in the Jungfraujoch (3580 m a.s.l.) (Conen et al., 2022). And the black rhombus represents the median $N_{INP}$ at −25 °C and −20 °C at the Storm Peak Laboratory in the northwestern Colorado Rocky Mountains (3220 m a.s.l.) (Hodshire et al., 2022).

*63. Figure 3a and 3b: Using thin lines instead of dots per spectrum may enhance the readability.*

**Response:** We made the changes in the revised manuscript.

*64. Figure 3a: Error bars extend beyond axis limits. As NINP, NINP-heat, and NINP-H2O2 are only used to calculate NINP-bio, NINP-other org, and NINP-inorg Figure 3a could be moved to appendix to focus on the essential plots 3b and 3c.*

**Response:** We modified the $N_{INP}$ concentration display range on the spectra, and moved the figure to Supplementary Information.

*65. Figure 3b: Are all the calculated differences in concentration significant? Could (exemplary) error bars be added?*

**Response:** The presence of error bars can sometimes obscure the information in the figure. Consequently, we have presented the error bars in Figure R10, and added it in the supporting information.

[Figure]

Figure R10. The $N_{INP}$, $N_{INP\text{-heat}}$, and $N_{INP\text{-}H2O2}$ as function of temperature. The solid line and dotted line show the sample measurement result by immersed in 5 mL MilliQ water and diluted the sample 30-120 times, respectively. The original $N_{INP}$ is marked by black dots, $N_{INP\text{-heat}}$ is marked by purple dots, and $N_{INP\text{-}H2O2}$ is marked by pink dots, with 20% error bars indicating the 95% confidence intervals.

*66. Figure 4a: Figure missing.*

**Response:** Thanks, we added this figure in revised manuscript, as shown in Figure R11.

[Figure]

Figure R11. Correlation analysis between (a) $N_{INP}$, (b) $N_{INP\text{-}bio}$, (c) $N_{INP\text{-}other\ org}$, (d) $N_{INP\text{-}inorg}$, and meteorological parameters, chemical compositions, as functions of temperature. The r denotes the Pearson correlation coefficients. The asterisk indicates $p < 0.05$, while the shades indicate that the number of data points is less than half of all samples at each temperature.

*67. Figure 4: As all other figures temperature increases to the right, the x axis of Figure 4 could be flipped for consistency. Which correlation coefficient was used? Specify in caption.*

**Response:** We adjusted the x axis in Figure 4, as illustrated in Figure R11.

*68. Figure 5: What is the red star? Is there a specific reason for the y-axis not being logarithmic in contrast to the other plots of INP concentrations?*

**Response:** The red star represents the city of Beijing, which has been removed from the figure. The logarithmic axe is not used because INP concentration is shown at specific freezing temperature, where the range of values is not broad.

**Reference:**

Chen, J., Wu, Z., Chen, J., Reicher, N., Fang, X., Rudich, Y., and Hu, M.: Size-resolved atmospheric ice-nucleating particles during East Asian dust events, Atmos Chem Phys, 21, 3491-3506, https://doi.org/10.5194/acp-21-3491-2021, 2021.

Chen, J., Wu, Z., Augustin-Bauditz, S., Grawe, S., Hartmann, M., Pei, X., Liu, Z., Ji, D., and Wex, H.: Ice-nucleating particle concentrations unaffected by urban air pollution in Beijing, China, Atmos Chem Phys, 18, 3523-3539, https://doi.org/10.5194/acp-18-3523-2018, 2018.

Chow, F. K., Wekker, S. F. D., and Snyder, B. J.: Mountain Weather Research and Mountain Weather Research and Forecasting: Recent Progress and Current Challenges, Springer Atmospheric Sciences, https://link.springer.com/book/10.1007/ 978-94-007-4098-3 (last access: 21 February 2022), 2013.

Conen, F., Einbock, A., Mignani, C., and Hüglin, C.: Measurement report: Ice-nucleating particles active $\geq -15$ °C in free tropospheric air over western Europe, Atmos. Chem. Phys., 22, 3433-3444, 10.5194/acp-22-3433-2022, 2022.

Conen, F., Yakutin, M. V., Yttri, K. E., and Hüglin, C.: Ice Nucleating Particle Concentrations Increase When Leaves Fall in Autumn, Atmosphere-Basel, 8, 202, https://doi.org/10.3390/atmos8100202, 2017.

DeMott, P. J., Prenni, A. J., Liu, X., Kreidenweis, S. M., Petters, M. D., Twohy, C. H., Richardson, M. S., Eidhammer, T., and Rogers, D. C.: Predicting global atmospheric ice nuclei distributions and their impacts on climate, P Natl Acad Sci USA, 107, 11217-11222, https://doi.org/10.1073/pnas.0910818107, 2010.

Guo, J., Zhang, J., Yang, K., Liao, H., Zhang, S., Huang, K., Lv, Y., Shao, J., Yu, T., Tong, B., Li, J., Su, T., Yim, S. H. L., Stoffelen, A., Zhai, P., and Xu, X.: Investigation of near-global daytime boundary layer height using high-resolution radiosondes: first results and comparison with ERA5, MERRA-2, JRA-55, and NCEP-2 reanalyses, Atmos. Chem. Phys., 21, 17079-17097, https://doi.org/10.5194/acp-21-17079-2021, 2021.

Hill, T. C. J., DeMott, P. J., Tobo, Y., Fröhlich-Nowoisky, J., Moffett, B. F., Franc, G. D., and Kreidenweis, S. M.: Sources of organic ice nucleating particles in soils, Atmos. Chem. Phys., 16, https://doi.org/7195-7211, 10.5194/acp-16-7195-2016, 2016.

Hodshire, A. L., Levin, E. J. T., Hallar, A. G., Rapp, C. N., Gilchrist, D. R., McCubbin, I., and McMeeking, G. R.: Technical Note: A High-Resolution Autonomous Record of Ice Nuclei Concentrations for Fall and Winter at Storm Peak Laboratory, Atmos. Chem. Phys. Discuss., 2022, 1-15, https://doi.org/10.5194/acp-2022-29, 2022.

Hsu, Y.-K., Holsen, T. M., and Hopke, P. K.: Comparison of hybrid receptor models to locate PCB sources in Chicago, Atmospheric Environment, 37, 545-562, https://doi.org/10.1016/S1352-2310(02)00886-5, 2003.

Jiang, H., Yin, Y., Su, H., Shan, Y. P., and Gao, R. J.: The characteristics of atmospheric ice nuclei measured at the top of Huangshan (the Yellow Mountains) in Southeast China using a newly built static vacuum water vapor diffusion chamber, Atmos Res, 153, 200-208, https://doi.org/10.1016/j.atmosres.2014.08.015, 2015.

Kanji, Z. A., Ladino, L. A., Wex, H., Boose, Y., Burkert-Kohn, M., Cziczo, D. J., and Krämer, M.: Overview of Ice Nucleating Particles, Meteorological Monographs, 58, 1.1-1.33, https://doi.org/10.1175/amsmonographs-d-16-0006.1, 2017.

Lau, K. M. and Wu, H. T.: Warm rain processes over tropical oceans and climate implications, Geophys Res Lett, 30, 5, https://doi.org/10.1029/2003gl018567, 2003.

Le, T. H., Wang, Y., Liu, L., Yang, J. N., Yung, Y. L., Li, G. H., and Seinfeld, J. H.: Unexpected air pollution with marked emission reductions during the COVID-19 outbreak in China, Science, 369, 702-711, https://doi.org/10.1126/science.abb7431, 2020.

Mulmenstadt, J., Sourdeval, O., Delanoe, J., and Quaas, J.: Frequency of occurrence of rain from liquid-, mixed-, and ice-phase clouds derived from A-Train satellite retrievals, Geophys Res Lett, 42, 6502-6509, https://doi.org/10.1002/2015gl064604, 2015.

O'Sullivan, D., Murray, B. J., Malkin, T. L., Whale, T. F., Umo, N. S., Atkinson, J. D., Price, H. C., Baustian, K. J., Browse, J., and Webb, M. E.: Ice nucleation by fertile soil dusts: relative importance of mineral and biogenic components, Atmos. Chem. Phys., 14, 1853-1867, https://doi.org/10.5194/acp-14-1853-2014, 2014.

Petters, M. D. and Wright, T. P.: Revisiting ice nucleation from precipitation samples, Geophys Res Lett, 42, 8758-8766, https://doi.org/10.1002/2015GL065733, 2015.

Slattberg, N., Lai, H. W., Chen, X. L., Ma, Y. M., and Chen, D. L.: Spatial and temporal patterns of planetary boundary layer height during 1979-2018 over the Tibetan Plateau using ERA5, International Journal of Climatology, 42, 3360-3377, https://doi.org/10.1002/joc.7420, 2022.

Tornow, F., Ackerman, A. S., and Fridlind, A. M.: Preconditioning of overcast-to-broken cloud transitions by riming in marine cold air outbreaks, Atmos Chem Phys, 21, 12049-12067, https://doi.org/10.5194/acp-21-12049-2021, 2021.

Wang, Z. W., Gallet, J. C., Pedersen, C. A., Zhang, X. S., Ström, J., and Ci, Z. J.: Elemental carbon in snow at Changbai Mountain, northeastern China: concentrations, scavenging ratios, and dry deposition velocities, Atmos. Chem. Phys., 14, 629-640, https://doi.org/10.5194/acp-14-629-2014, 2014.

Welti, A., Müller, K., Fleming, Z. L., and Stratmann, F.: Concentration and variability of ice nuclei in the subtropical maritime boundary layer, Atmos. Chem. Phys., 18, 5307-5320, https://doi.org/10.5194/acp-18-5307-2018, 2018.

Wieder, J., Mignani, C., Schär, M., Roth, L., Sprenger, M., Henneberger, J., Lohmann, U., Brunner, C., and Kanji, Z. A.: Unveiling atmospheric transport and mixing mechanisms of ice-nucleating particles over the Alps, Atmos Chem Phys, 22, 3111-3130, https://doi.org/10.5194/acp-22-3111-2022, 2022.

Wu, C., Zhang, S., Wang, G. H., Lv, S. J., Li, D. P., Liu, L., Li, J. J., Liu, S. J., Du, W., Meng, J. J., Qiao, L. P., Zhou, M., Huang, C., and Wang, H. L.: Efficient heterogeneous

formation of ammonium nitrate on the saline mineral particle surface in the atmosphere of East Asia during dust storm periods, Environ Sci Technol, 54, 15622-15630, https://doi.org/10.1021/acs.est.0c04544, 2020.

Zhao, X., Kim, S.-K., Zhu, W., Kannan, N., and Li, D.: Long-range atmospheric transport and the distribution of polycyclic aromatic hydrocarbons in Changbai Mountain, Chemosphere, 119, 289-294, https://doi.org/10.1016/j.chemosphere.2014.06.005, 2015.

---

## Referee Report (RR1)

Review of "Measurement report: Atmospheric Ice Nuclei at Changbai Mountain (2623 m a.s.l.) in Northeastern Asia" by Sun et al.

The paper reports on the outcome of a one-month field campaign conducted at the Tianchi site on Mt. Changbai during the summer of 2021. Filter samples collected during 10 days at the end of the campaign are used to measure the concentration of immersion freezing INPs and additional heat treatment and H2O2 degradation of the sample are used to infer the contribution of biological INP to the INP concentration. In addition, the INP concentration data is correlated to axillary meteorological data, bulk aerosol composition and trace gas concentration, and back trajectories are used to find potential source regions of INP active in a certain temperature regime. The authors find that biological INP contribute the majority to the INP concentration and stress the importance of soil dust as a regional source as well as long range transport of biological marine INP and biological INP from vegetation.

While the conducted measurements and analysis are state of the art, the interpretation is mostly speculative and not explained clear enough. Instead of deducing conclusions from signals in the current data the reader is pointed to literature sources to back up interpretations. An additional shortcoming is the very limited number of samples collected, making it more of a preliminary study hinting at several interesting aspects on the sources of INPs in Northeast Asia that deserve deeper investigation.

The authors have given insightful replies to some of the comments in the previous round of review but have not managed to implement all necessary clarifications and improvements into the manuscript. On some occasions the changes the authors mention in the reply have not been transferred to the revised manuscript, for example reply to question 5 of Referee #3.

Because the manuscript has not been substantially improved in reply to the previous round of review and the many additional comments below, I recommend major revision before the manuscript can be considered for publication. I also recommend a thorough language check by a native English speaker to avoid misunderstandings of the scientific content due to poor phrasing.

Major comments:

The surrounding of Tianchi station next to Tianchi Lake, inside a crater could have a major impact on the measured $N_{INP}$. Provide an explain in the manuscript how the contribution of long-distant transport can be clearly distinguished from local or regional sources of INP.

The influence of fog and precipitation at the sampling site on the measured $N_{INP}$ should be analysed in more detail. In addition, the influence of precipitation along trajectories should be considered in the CWT analysis. For INPs to form ice in mixed-phase clouds they need to act as cloud condensation nuclei as well. Cloud formation and precipitation along trajectories should therefore wash them out preventing their long-range transport.

As pointed out in the previous round of reviews, the appropriateness of the PBL height from ERA5 for the Tianchi station site is questionable and needs further confirmation. The authors acknowledged in their response, that the GDAS topography in the region is off by over 900m. Also, complex effects from the mountain and its crater on the formation of the boundary layer cannot be neglected. Are additional measurements for example from balloon soundings available to provide evidence that the PBL height from ERA5 are representative for the location?

Specific comments:

1) Line 21: explain why correlation to windspeed, $Ca^{2+}$ and isoprene suggests bio aerosol is attached to soil dust and act as INP.

2) Line 22ff: Provide an explanation why PBL height could be positively correlated to $N_{INP}$. The opposite could be expected because of dilution of aerosol concentration with increasing PBL height. Explain based on what data it is found that valley breeze influences $N_{INP}$.

3) Line 33ff: inappropriate citations. Koop et al., 2000, Murray et al., 2010, Cziczo et al., 2013 report on homogeneous ice nucleation and ice nucleation under cirrus cloud conditions not immersion freezing. For example, Murray et al., 2012 would be a better reference.

4) Line 37ff: By definition, mixed-phase clouds contain droplets and therefore only exist at water saturated conditions. Water saturation is not a prerequisite for ice formation. Reformulate.

5) Line 43ff: Not corrected as mentioned in response to Referee#3 question 5. Implement correction.

6) Line 45: Can lichen be considered biological aerosol? As I understand they are a symbiotic organism attached to surfaces.

7) Line 48: Double check if it is correct that pollen is non-proteinaceous. I find they contain about 30% protein.

8) Line 51: add a citation for the activity temperature of mineral dust and sea spray.

9) Line 55: inappropriate citations: references are for deposition ice nucleation, replace with references relevant for immersion freezing.

10) Line 59-62: Provide references for "numerous studies", "spatial distribution heterogeneity", and the altitude of mixed-phase clouds.

11) Line 69f: INP sources are not in the atmosphere but on the surface. Reformulate.

12) Line 74ff: JFJ station is not a good example for having high vegetation coverage. It is surrounded by bare rock and ice. Reformulate.

13) Line 83f: Clarify what is meant by "impact of bio INP on cloud droplets and their contribution to formation of precipitation".

14) Line 91: Explain what boundaries are meant by "transboundary transport of air mass".

15) Line 98: provide a reference for pollution transport to the Arctic from this region.

16) Line 112: Extend the discussion on weather conditions. What is meant by humid weather? Was it raining, or was there fog? Fog and rain can have an impact on the sampling as well as on the transport of INP. Potential evidence for this impact can be seen from the anticorrelation of $N_{INP}$ in the LTR with RH in Fig.4.

17) Line 116f: Elaborate on the potential impact of the surrounding (lake, dense vegetation) on $N_{INP}$ and other variables, for example isoprene concentrations.

18) Line 121ff: Provide characteristics on the sampler inlet cut-off. Is it a total inlet? As RH was high during the campaign and often at 100%, was there fog or precipitation? Does the sampler collect cloud droplets as well?

19) Line 123: Table S1 lists 25 sample intervals. Was the filter not changes for the two intervals during nighttime on the 11.8.? If not, maybe the 6min gap in sampling can be neglected and doesn't need to be listed. Otherwise specify what happened.

20) Line 124, Tab.S1: The sample duration in Tab.S1 do not agree with the time difference between start to end time. Where does the duration come from? Does the sampler turn off if the pressure-drop over the filter is too high? The sample volume has been calculated using the set flow rate and start to end time. Does the sampler not provide a more precise measurement of the sample volume? If possible, use measured and not calculated sample volume to calculate $N_{INP}$.

21) Line 126: provide model number for the $PM_{2.5}$ sampler.

22) Explain how auxiliary data was averaged for the correlation analysis with $N_{INP}$.

23) Line 139: is the enclosed droplet chamber part of the LTS120 cold stage?

24) Line 147ff: Clarify if sample droplets rest inside the oil or if the oil is between the cold stage and the glass slide? Are droplets in contact with the aluminium spacer? Revise the step-by-step description in the manuscript.

25) Line 148: "filled" might be the wrong word to describe the procedure. Do you mean covered?

26) Line 160 and Eq.2: As Equation 2 is irrelevant for the rest of the manuscript I suggest deleting it. If you want to keep it in, define K(T) in the text and explain what is meant by "cumulative concentration of each droplet above K(T)".
Add how Vair is calculated from the sample volume, volume of washing water and droplet volume. Add the equation how the background signal is subtracted.

27) Line 183-184: As pointed out by Referee#1 comment 22. in the previous round of reviews, the authors should elaborate in the text why the ERA5 PBL height data is applicable specific to their analysis.

28) Figure 2a: add dilution data to this figure to show that measurements were not affected by the water background instead of having an additional Figure S2.

29) Line 209: As pointed out above, add equation to Sec.2.3. how the background was subtracted and refer to the equation here.

30) Line 241f: You could cite Kanji et al.,2020, here. Correct sentence structure.
There are significant correlations between BC and $N_{INP}$ below -20°C shown in Fig.4 not supporting this statement. The discrepancy to literature is worth adding a discussion of these data here or line 300.

31) Line 271f: The fraction of bio-INP is affected by the concentration of these INP and much less by their moderate or high ice activity. Reformulate.

32) Line 287f: Specify where these previous studies were conducted and provide references.

33) Line 295f, 309f: Please justify speculations by explaining the deductive chain of logical steps that lead to them. It is not obvious to me here. Is $Ca^{2+}$ a proxy for soil dust? Are there sources of soil dust downwind of the sampling site? Considering the weather situation, wouldn't soil be wet and therefore not prone to wind erosion? Isn't soil covered by vegetation in this season that prevents erosion? Clarify in the text.
Is it realistic that particle concentrations below 0.1 $L^{-1}$ ($N_{INP}$ in the -8°C to -11°C range) influence the measured $Ca^{2°}$ concentration?

34) Line 298f: Provide an explanation for the correlation of $N_{INP}$ to ambient temperature and RH.

35) Line 313f: Explain why a correlation with WS and ambient temperature indicates local sources.

36) Line 316f, Line 396: Oxidation products of isoprene should be water soluble and not contribute to immersion freezing. Correlation does not imply direct causation.

37) Line 318: It could be expected that SOA activate as CCN when RH>100%. Did the sampler also collect cloud droplets? An anticorrelation to RH could provide a hint if the INP are also CCN. For INP to generate ice in mixed-phase clouds CCN activation is a prerequisite. I encourage a reanalysis of the data considering the weather situation at the time of sampling.

38) Line 323: The $N_{INP}$ was measured in this study not the IN activity. Reformulate.

39) Line 338f: Provide a reasonable explanation how PBL height can be positively correlated to $N_{INP}$. The opposite could be expected due to dilution of air with increasing PBL height. Total particle concentration is usually anticorrelated to PBL height.

40) Line 341f: This is not supported by the data. If valley breeze transported INPs to the station during daytime, why is there not a difference between night and day $N_{INP}$?

41) Line 350f: Clarify how $N_{INP}$ is related to phytoplankton bloom and growth. The Japanese Sea experiences a phytoplankton bloom twice a year, one in spring and one in fall (Wang et al., 2022). Clarify the relevance for the current observations.

42) Line 352-355, Figure 5: It is surprising that two distinct origins for HTR and LTR bio INPs are found, as it could be assumed that for cumulative $N_{INP}$ the occurrence of high $N_{INP}$ at -10.5°C is correlated to high $N_{INP}$ at -20°C. Is this not the case? Could there be an artifact from the measuring range? For example, how were samples with a frozen fraction =1 at temperatures above -20°C included in the

CWT analysis in the LTR? Did you use the data of their dilutions? The $N_{INP}$ colour scale indicates that dilutions were not included.

43) Line 370ff: Explain how it can be concluded that long-range bio-INP were less prominent in your measurements? It seems contradictory to say there was no qualitative or quantitative analysis of bio INP and then state that they were less prominent from long-range transport.

44) Line 376: Clarify the interpretation between PBL height and local sources or long-range transport. Are you inferring that long-range transport contribution to $N_{INP}$ occurs exclusively by night and as soon as the boundary layer forms exclusively local sources contribute to $N_{INP}$? This seems a critical assumption for your analysis and should be discussed in more detail and supported by stronger evidence.

45) Line 381ff: There was no diurnal cycle of $N_{INP}$ observed in this study. Explain how conclusions can still be drawn about a diurnal cycle.

46) Line 385f: The methods applied in this study do not allow to explore properties of INPs. Only their abundance was measured. Reformulate.

47) Line 393: An increase in bio $N_{INP}$ indicates that the concentration of bio-INP increased, not the activity. The type of measurements does not provide information if, for example, 1% or 100% of a certain INP type was active at a certain temperature. Reformulate.

48) Line 398ff: Please resolve the apparent contradiction between local soil dust sources and CWT pointing to long range transport by clarifying how long-distance sources can be disentangled from local sources. In addition, would the CWT analysis reveal different patterns for the nighttime $N_{INP}$ measurements, or at -10°C and -20°C? Would the correlation analysis, for example with $Ca^{2+}$ differ if done separately for daytime and nighttime samples?

49) Figure S5: replace with Fig. R3 from the Author's Response and include the reply to question 22 from Referee #1 explaining how airmasses approach the measurement site and why the trajectory endpoint was chosen at 967m a.g.l.

50) Figure S6: Specify if only daytime or all samples were used to create this figure.

Technical corrections:

1) misspelled citations: DeMott, O'Sullivan, McCluskey, in several instances.
2) there is no "the" needed before Switzerland.
   Also, in Figure 2.b) remove "The" in front of Weissfluhjoch and Jungfraujoch.
3) Line 110: Maybe use standard error of the mean instead of standard deviation to avoid range of RH (92.4%+11.8%=104,2%) to extend beyond the range of measurement.
   Also, line 309: 0.5ugm$^{-3}$ – 1.0ugm$^{-3}$ = - 0.5ugm$^{-3}$ is unphysical.
4) Line 166-168: It is unclear what is explained here. Reformulate.
5) Line 173: Do you mean: …, which is 1.96 for a 95% confidence interval? Reformulate.
6) Line 191ff: Incomprehensible sentence. Reformulate.
7) Line 198: define $n_{ij}$ in the text.
8) Line 213: replace "three" with "up to 5" to be consistent with what is stated in the conclusion.
9) Line 233-235: Incomprehensible sentence. Reformulate.
10) Line 237: Reformulate. …magnitude at temperatures from -26.0°C to -3.0°C.
11) Line 288: replace INPs with $N_{INP}$
12) Line 321: define SNA.
13) Line 329: replace INPs with $N_{INP}$.
14) Line 336: Define HLR. Do you mean HTR?
15) Line 347: "under immersion mode" do you mean "in the immersion mode"?
16) Line 352: "occurred in" do you mean "coming from"?
17) Line 363: define "the middle layer".

18) When reporting $N_{INP}$ ranges and consecutive the temperature range, I suggest giving the temperature range from high to low temperature if the $N_{INP}$ range is given from low to high, so that the first number in the $N_{INP}$ range corresponds to the first number in the temperature range.

19) Specify whenever you mean "ambient temperature" to avoid confusion to experimental temperature of IN experiments. In particular: line 299, 313.

References:

Kanji, Z. A., Welti, A., Corbin, J. C., & Mensah, A. A. (2020). Black carbon particles do not matter for immersion mode ice nucleation. Geophysical Research Letters, 46, e2019GL086764. https://doi.org/10.1029/2019GL086764

Wang D, Fang G, Jiang S, Xu Q, Wang G, Wei Z, Wang Y and Xu T (2022). Satellite-detected phytoplankton blooms in the Japan/East Sea during the past two decades: Magnitude and timing. Front. Mar. Sci. 9:1065066. doi: 10.3389/fmars.2022.1065066

---

## Author Response (AR2)

*Public justification (visible to the public if the article is accepted and published):*

*I thank the authors for addressing most of the comments and suggestions provided by Reviewers #1 and #3; however, the two additional reviewers are concerned about the readability of the manuscript. Additionally, Reviewer #4 has several Major, Specific, and Technical comments and I also added a list of Minor/Technical comments. Please improve the grammar of entire manuscript and properly address each comment listed below.*

**Response:** We gratefully thank the reviewer for the constructive comments and suggestions to improve the manuscript. Below are the detailed point-to-point responses to the reviewer's comments. For clarity, the reviewer's comments are listed below in black italics, while our responses and changes in the manuscript are shown in blue. The changes in the revised manuscript and supporting materials are also highlighted.

*Reviewer #2:*

*Please carefully go through the manuscript to meet the standard of ACP.*

**Response:** Thanks for the comments. We carefully go through the manuscript and make the necessary revisions to ensure that it meets the standards of ACP.

*Reviewer #4:*

*Review of "Measurement report: Atmospheric Ice Nuclei at Changbai Mountain (2623 m a.s.l.) in Northeastern Asia" by Sun et al.*

*The paper reports on the outcome of a one-month field campaign conducted at the Tianchi site on Mt. Changbai during the summer of 2021. Filter samples collected during 10 days at the end of the campaign are used to measure the concentration of immersion freezing INPs and additional heat treatment and H2O2 degradation of the sample are used to infer the contribution of biological INP to the INP concentration. In addition, the INP concentration data is correlated to axillary meteorological data, bulk aerosol composition and trace gas concentration, and back trajectories are used to find potential source regions of INP active in a certain temperature regime. The authors find that biological INP contribute the majority to the INP concentration and stress the importance of soil dust as a regional source as well as long range transport of biological marine INP and biological INP from vegetation.*

*While the conducted measurements and analysis are state of the art, the interpretation is mostly speculative and not explained clear enough. Instead of deducing conclusions from signals in the current data the reader is pointed to literature sources to back up interpretations. An additional shortcoming is the very limited number of samples collected, making it more of a preliminary study hinting at several interesting aspects on the sources of INPs in Northeast Asia that deserve deeper investigation. The authors have given insightful replies to some of the comments in the previous round of review but have not managed to implement all necessary clarifications and improvements into the manuscript. On some occasions the changes the authors mention in the reply have not been transferred to the revised manuscript, for example reply to question 5 of Referee #3.*

*Because the manuscript has not been substantially improved in reply to the previous round of review and the many additional comments below, I recommend major revision before the*

*manuscript can be considered for publication. I also recommend a thorough language check by a native English speaker to avoid misunderstandings of the scientific content due to poor phrasing.*

**Response:** We are grateful to the reviewer for the valuable comments. We have endeavored to respond to these comments and revise our manuscript accordingly.

*Major comments:*
*The surrounding of Tianchi station next to Tianchi Lake, inside a crater could have a major impact on the measured NINP. Provide an explain in the manuscript how the contribution of long-distant transport can be clearly distinguished from local or regional sources of INP.*

**Response:** Thanks for the comment. Changbai Mountain is a dormant volcano, unaffected by local volcanic eruptions. Human activities exhibit minimal impact, with no nearby industrial facilities, and a significant reduction in tourism activities during our sampling period. Therefore, our sampling site is an ideal site for studying the regional background atmosphere of Northeast Asia, as extensively elaborated in the section 2.1. In the regional background atmosphere, the impact of long-range transport on aerosols and INPs were determined using HYSPLIT model.

*The influence of fog and precipitation at the sampling site on the measured NINP should be analysed in more detail. In addition, the influence of precipitation along trajectories should be considered in the CWT analysis. For INPs to form ice in mixed-phase clouds they need to act as cloud condensation nuclei as well. Cloud formation and precipitation along trajectories should therefore wash them out preventing their long-range transport.*

**Response:** We distinguished three different meteorological conditions, i.e., sunny days, foggy days, and rainy days, and conducted a comparative analysis of INPs spectra (see Figure R1). Within the HTR, no significant difference was observed among the three weather conditions. In the LTR, the INP concentrations on rainy days were slightly lower than those on sunny and foggy days. These have been added in the revision.

[Figure]

Figure R1. $N_{INP}$ for sunny, foggy, and rainy day as functions of temperature. Each point represents the median value and the shadow area represents the maximum and minimum value.

The CWT analysis does not account for the impacts of wet deposition, including cloud droplets and precipitation along the trajectories. We acknowledge this limitation in this study, and added this in section 2.5.

For non-precipitation clouds, particulate matter with a dry diameter in the range of 80 to 300 nm is deemed to exhibit a more efficient activation as cloud condensation nuclei (CCN) (Ma et al., 2016). For particles with diameters exceeding 1 μm, they are less likely to undergo activation in a cloud due to their large activation radius (> 10 μm) (Grabowski et al., 2022). In contrast, particulate matter with the potential to serve as ice nuclei typically exhibits larger diameters, generally surpassing 500 nm, owing to the presence of a broader spectrum of nucleation-active sites (Demott et al., 2010). This distinction in particle radius underscores the variability in their ability to active as either CCN or IN.

*As pointed out in the previous round of reviews, the appropriateness of the PBL height from ERA5 for the Tianchi station site is questionable and needs further confirmation. The authors acknowledged in their response, that the GDAS topography in the region is off by over 900m. Also, complex effects from the mountain and its crater on the formation of the boundary layer cannot be neglected. Are additional measurements for example from balloon soundings available to provide evidence that the PBL height from ERA5 are representative for the location?*

**Response:** We apologize that the balloon-derived PBL data is unavailable around our sampling site. Instead, we conducted a comparison of PBL heights between ERA5 and GDAS products, respectively. As shown in Figure R2, the two datasets showed high consistency. This further demonstrates the robustness and reliability of PBL heights in both products. Hence, employing ERA5 PBL data in our analysis is deemed reasonable.

[Figure]

Figure R2. The relationship of PBL data obtained from ERA5 and GDAS, from Aug. 10 to 21, 2021, at our sampling site.

***Specific comments:***

*1) Line 21: explain why correlation to windspeed, $Ca^{2+}$ and isoprene suggests bio aerosol is attached to soil dust and act as INP.*

**Response:** Thanks for the comment. In this study, the origin of $Ca^{2+}$ was considered to be mainly of natural sources due to calcareous nature of the soil. The presence of isoprene indicates a biological source. Previous studies have reported that biological materials may attach to or mix with dust particles and promote INPs formation. Therefore, we speculate that the higher windspeed may have facilitated the exposure of the local soil dust and bioaerosol containing bio-INPs to the air. We have modified the statements in the abstract and section 3.3.

*2) Line 22ff: Provide an explanation why PBL height could be positively correlated to NINP. The opposite could be expected because of dilution of aerosol concentration with increasing PBL height. Explain based on what data it is found that valley breeze influences NINP.*

**Response:** Thanks for the comment. Our sampling site is situated atop Changbai Mountain, at an elevation of 2623 m above sea level. The impact of dilution within the boundary layer is negligible at this location, as it consistently above the PBL for the majority of the time. According to Ketterer et al. (2014), the upslope valley wind could increase the altitude of the PBL height locally, and if strong enough, trigger the vertical exchange of PBL air into the free troposphere. In our investigation, the positive correlation between the PBL and INPs may be elucidated by the presence of a valley breeze.

*3) Line 33ff: inappropriate citations. Koop et al., 2000, Murray et al., 2010, Cziczo et al., 2013 report on homogeneous ice nucleation and ice nucleation under cirrus cloud conditions*

*not immersion freezing. For example, Murray et al., 2012 would be a better reference.*

**Response:** Thanks. We have replaced the references accordingly in the revision.

*4) Line 37ff: By definition, mixed-phase clouds contain droplets and therefore only exist at water saturated conditions. Water saturation is not a prerequisite for ice formation. Reformulate.*

**Response:** Thanks for the comment. We have revised the statement as follows: "Recent studies have concluded that water saturation is a prerequisite for ice formation in low- and mid-level clouds, and that contact and immersion freezing are the most primary pathways for ice formation (Sassen and Khvorostyanov, 2008; Phillips et al., 2007; Murray et al., 2012)."

*5) Line 43ff: Not corrected as mentioned in response to Referee#3 question 5. Implement correction.*

**Response:** Thanks for the comment. We have revised the statement as follows: "Biological aerosols from microbial and proteinaceous origin, containing ice nucleation active protein, demonstrate significant efficiency as INPs at temperatures above −15 °C (Phelps et al., 1986; Petters and Wright, 2015; Murray et al., 2012; Kunert et al., 2019; Huang et al., 2021)."

*6) Line 45: Can lichen be considered biological aerosol? As I understand they are a symbiotic organism attached to surfaces.*

**Response:** Lichens are recognized as extraordinary biological ice nucleators, and the ice nucleation activity is primarily resided in the mycobiont, which is the proteinaceous biological INPs. We have revised the statement and added reference as follows: "For example, biological components in lichens can induce freezing above −10 °C (Kieft, 1988; Moffett et al., 2015), and certain bacterial organisms such as Pseudomonas syringae can facilitate droplet freezing even at extremely high temperatures (above −2 °C) (Maki et al., 1974)."

*7) Line 48: Double check if it is correct that pollen is non-proteinaceous. I find they contain about 30% protein.*

**Response:** Although pollen contains proteins, its non-proteinaceous compounds, such as polysaccharides, that induce ice formation. We rewrote this sentence to make it more precisely: "The other biological aerosols with non-proteinaceous compounds act as ice nucleation catalysts, such as pollen, cellulose, and other macromolecular organic particles, can also induce ice formation through heat-resistant polysaccharides on their surfaces, but at lower temperatures than proteinaceous biological particles (Knopf et al., 2010; Pummer et al., 2012)."

*8) Line 51: add a citation for the activity temperature of mineral dust and sea spray.*

**Response:** Thanks. We have cited the references, i.e., Alpert et al. (2022); Atkinson et al. (2013); Ladino et al. (2019), in the revision.

*9) Line 55: inappropriate citations: references are for deposition ice nucleation, replace with references relevant for immersion freezing.*

**Response:** Thanks. We have revised the citations in the revision.

*10) Line 59-62: Provide references for "numerous studies", "spatial distribution heterogeneity", and the altitude of mixed-phase clouds.*

**Response:** Thanks. We have cited the references in the revision.

*11) Line 69f: INP sources are not in the atmosphere but on the surface. Reformulate.*

**Response:** To avoid ambiguity, we have revised as "This decline in INPs could exceed 60% per kilometer during the cold season (from 1631 m a.s.l. to 2693 m a.s.l.) (Wieder et al., 2022), which was attributed to the scarcity of effective INPs sources in high-altitudes".

*12) Line 74ff: JFJ station is not a good example for having high vegetation coverage. It is surrounded by bare rock and ice. Reformulate.*

**Response:** In the revised manuscript, we have revised this sentence to make it more accurate: "In some mountainous areas, biogenic aerosols can act as the most abundant type of INPs."

*13) Line 83f: Clarify what is meant by "impact of bio INP on cloud droplets and their contribution to formation of precipitation".*

**Response:** We have revised this sentence to make it more accurate: "Because the number of rainwater samples was limited, further research is necessary to explore the impact of biological INPs on ice formation and their contribution to the formation of precipitation."

*14) Line 91: Explain what boundaries are meant by "transboundary transport of air mass".*

**Response:** The term "boundaries" refers specifically to national boundaries. We have added this information as follows: "Given the high-altitude of Changbai Mountain, it is an ideal location to capture the characteristics of the regional atmospheric background and transboundary transport of air masses from nearby countries."

*15) Line 98: provide a reference for pollution transport to the Arctic from this region.*

**Response:** Thanks. We have cited the references in the revision.

*16) Line 112: Extend the discussion on weather conditions. What is meant by humid weather?*

*Was it raining, or was there fog? Fog and rain can have an impact on the sampling as well as on the transport of INP. Potential evidence for this impact can be seen from the anticorrelation of NINP in the LTR with RH in Fig.4.*

**Response:** We distinguished three different meteorological conditions, i.e., sunny days, foggy days, and rainy days, and conducted a comparative analysis of INPs spectra (see Figure R1). Within the HTR, no significant difference was observed among the three weather conditions. In the LTR, the INP concentrations on rainy days were slightly lower than those on sunny and foggy days. These have been added in the revision.

*17) Line 116f: Elaborate on the potential impact of the surrounding (lake, dense vegetation) on NINP and other variables, for example isoprene concentrations.*

**Response:** We have added the influence of the surrounding in section 3.1, read as follows: "In this study, local sources such as vegetation and the lake may impact INPs concentration, with biogenic emissions potentially exhibiting variations between daytime and nighttime."

*18) Line 121ff: Provide characteristics on the sampler inlet cut-off. Is it a total inlet? As RH was high during the campaign and often at 100%, was there fog or precipitation? Does the sampler collect cloud droplets as well?*

**Response:** We use a TSP sampler with a 100 μm cut-point. The sampling head design effectively prevents the collection of rainwater during rainy days. But fog and cloud droplets remain unavoidable. It is a common challenge encountered with all types of samplers.

*19) Line 123: Table S1 lists 25 sample intervals. Was the filter not changes for the two intervals during nighttime on the 11.8.? If not, maybe the 6min gap in sampling can be neglected and doesn't need to be listed. Otherwise specify what happened.*

**Response:** On August 11, a momentary power outage led to a temporary interruption in instrument measurements lasting approximately 6 minutes. This information has been added in Table S1.

*20) Line 124, Tab.S1: The sample duration in Tab.S1 do not agree with the time difference between start to end time. Where does the duration come from? Does the sampler turn off if the pressure-drop over the filter is too high? The sample volume has been calculated using the set flow rate and start to end time. Does the sampler not provide a more precise measurement of the sample volume? If possible, use measured and not calculated sample volume to calculate NINP.*

**Response:** Thank you for the comment. The variation between the instrument-recorded volume and the calculated volume is within 2%. We adopted the recorded volume in Table S1 accordingly. Throughout the sampling period, there were no instances of the sampler shutting down due to excessive pressure.

*21) Line 126: provide model number for the PM2.5 sampler.*

**Response:** Thank you for the comment. We added the instrument model in the revision.

*22) Explain how auxiliary data was averaged for the correlation analysis with NINP.*

**Response:** The calculate method has been added in the revision as follows: "The parameters mentioned above were analyzed by determining the average value over the corresponding INPs sampling period."

*23) Line 139: is the enclosed droplet chamber part of the LTS120 cold stage?*

**Response:** Yes, the cold stage is located inside the droplet chamber. We revised the statement as follows: "GIGINA is a cold-stage-based ice nucleation array that consists of an enclosed droplet chamber with a commercial cold stage inside (LTS120, Linkam, Epsom Downs, UK), an external refrigerated water circulator (VIVO RT4, Julabo, Seelbach, Germany), and a charge-coupled device (CCD) camera (DMK33G274, The Imaging Source, Bremen, Germany)."

*24) Line 147ff: Clarify if sample droplets rest inside the oil or if the oil is between the cold stage and the glass slide? Are droplets in contact with the aluminium spacer? Revise the step-by-step description in the manuscript.*

**Response:** The silicone oil was between the hydrophobic glass slide and cold stage, and the droplets cannot contact with the aluminium spacer. To avoid confusion, we have revised as "First, a hydrophobic glass slide was placed on a cold stage and covered with silicone oil between them to achieve good thermal contact. Second, a round aluminum spacer with 90 round compartments was placed on the glass slide, and the particle suspension was sequentially pipetted into the center region of each compartment."

*25) Line 148: "filled" might be the wrong word to describe the procedure. Do you mean covered?*

**Response:** Yes, we have replaced this word by "covered" in the revision.

*26) Line 160 and Eq.2: As Equation 2 is irrelevant for the rest of the manuscript I suggest deleting it. If you want to keep it in, define K(T) in the text and explain what is meant by "cumulative concentration of each droplet above K(T)". Add how Vair is calculated from the sample volume, volume of washing water and droplet volume. Add the equation how the background signal is subtracted.*

**Response:** We agree and delete the Equation of K(T). The equation for calculating $N_{INP}$ has been revised to include the background signal and is modified as follows: "The cumulative

number concentration of INPs ($N_{INP}$) per unit volume of sampled air were calculated following the method of Vali (1971, 2015):

$$N_{INP}(T) = -\frac{ln[1-f_{ice,\ sample}(T)]-ln[1-f_{ice,\ blank}(T)]}{V_{air}}\ (L^{-1}\ air), \qquad (2)$$

where $V_{air}$ is the total volume of sampled air per droplet converted to standard conditions (0 °C and 1013 hPa). The $f_{ice,\ sample}$ and $f_{ice,\ blank}$ are the measured frozen fractions for the filter samples and the field blanks, respectively. The calculation for Vair entails multiplying the droplet volume (1 µl) by the sample volume and then dividing by the volume of wash water.

*27) Line 183-184: As pointed out by Referee#1 comment 22. in the previous round of reviews, the authors should elaborate in the text why the ERA5 PBL height data is applicable specific to their analysis.*

Response: The PBL data obtained from ERA5 has been widely used in numerous studies, such as Le et al. (2020), Tornow et al. (2021), and Slattberg et al. (2022). In this analysis, we compared PBL heights obtained from ERA5 and the Global Data Assimilation System (GDAS) meteorological fields, and find a high consistency between the two datasets, as illustrated in Figure R2. This further demonstrates the robustness and reliability of PBL data of the two products. Therefore, it is reasonable for us to use ERA5 PBL data.

*28) Figure 2a: add dilution data to this figure to show that measurements were not affected by the water background instead of having an additional Figure S2.*

**Response:** We incorporated Figure 2a into the Figure S2 in the Supporting Information, as illustrated in Figure R3.

[Figure]

Figure R3. Frozen fractions ($f_{ice}$) of collected samples measured by GIGINA as functions of temperature is shown by the black curves, and presented together with blank filters (orange curves) and MilliQ water (purple curves) as background signals. The dashed line presented the diluted sample suspension.

*29) Line 209: As pointed out above, add equation to Sec.2.3. how the background was*

*subtracted and refer to the equation here.*

**Response:** Please see our response to Comment 26.

*30) Line 241f: You could cite Kanji et al.,2020, here. Correct sentence structure. There are significant correlations between BC and NINP below -20°C shown in Fig.4 not supporting this statement. The discrepancy to literature is worth adding a discussion of these data here or line 300.*

**Response:** Thanks. We have cited this reference in the revised manuscript.

We have added a brief discussion in section 3.3. "Previous studies have suggested that carbonaceous particles may not act as efficient INPs in the immersion mode or could decrease ice nucleation activity in polluted urban environments, which is attribute to the formation of organic coatings (Kanji et al., 2020; Schill et al., 2020; Nichman et al., 2019; Hammer et al., 2018). However, our observations revealed a positive correlation between BC and both $N_{INP}$ and $N_{INP-inorg}$ in the LTR. The discrepancy may be attributed to the different sources and aging degrees of BC, which remains unclear."

*31) Line 271f: The fraction of bio-INP is affected by the concentration of these INP and much less by their moderate or high ice activity. Reformulate.*

**Response:** We agree and revised as follow: "Interestingly, when the temperature decreased from -16.5 °C to -21.5 °C, the median of $F_{INP-bio}$ increased from 0.8 to 0.9, indicating the high concentration of bio-INPs in the LTR."

*32) Line 287f: Specify where these previous studies were conducted and provide references.*

**Response:** We have removed this sentence in the revision.

*33) Line 295f, 309f: Please justify speculations by explaining the deductive chain of logical steps that lead to them. It is not obvious to me here. Is Ca2+ a proxy for soil dust? Are there sources of soil dust downwind of the sampling site? Considering the weather situation, wouldn't soil be wet and therefore not prone to wind erosion? Isn't soil covered by vegetation in this season that prevents erosion? Clarify in the text. Is it realistic that particle concentrations below 0.1 L-1 (NINP in the -8°C to -11°C range) influence the measured Ca2+ concentration?*

**Response:** The vegetation types across Changbai Mountain exhibits obvious variations from the foothills to its summit. The transition encompasses tall temperate broad-leaved forests at the mountain's foothills, evolving into Changbai Pinus bungeana (coniferous forests), and further to the low creeping birch (shrub), alpine meadows, mossy lichens, and exposed rocks (Xu et al., 2004). Vegetated soil acts as a mitigating factor against erosion. However, the prevalence of low-lying vegetation around Tianchi Lake may facilitate soil erosion during

periods of high wind speed (up to 25.7 m/s in our observation). Numerous studies show that the origin of $Ca^{2+}$ is primarily attributed to natural sources due to the calcareous nature of the soil (Al-Momani, 2003; Al-Khashman, 2005). The $Ca^{2+}$ was extracted from $PM_{2.5}$ samples and the concentration was analyzed using an ion chromatograph, which is unaffected by INP concentrations.

*34) Line 298f: Provide an explanation for the correlation of NINP to ambient temperature and RH.*

**Response:** Please see our response to Comment 16.

*35) Line 313f: Explain why a correlation with WS and ambient temperature indicates local sources.*

**Response:** We apologize for the ambiguity and have removed the statement in the revision.

*36) Line 316f, Line 396: Oxidation products of isoprene should be water soluble and not contribute to immersion freezing. Correlation does not imply direct causation.*

**Response:** Thanks. We have revised the statement as "Although the oxidation products of isoprene are expected to be water-soluble and unable to induce immersion freezing, the observed positive correlation suggests a potential role of secondary organic compounds associate with vegetation or other biogenic sources in ice nucleation."

*37) Line 318: It could be expected that SOA activate as CCN when RH>100%. Did the sampler also collect cloud droplets? An anticorrelation to RH could provide a hint if the INP are also CCN. For INP to generate ice in mixed-phase clouds CCN activation is a prerequisite. I encourage a reanalysis of the data considering the weather situation at the time of sampling.*

**Response:** Particulate matter with a dry diameter in the range of 80 to 300 nm is deemed to exhibit a more efficient activation as CCN (Ma et al., 2016). For particles with diameters exceeding 1 μm, they are less likely to undergo activation in a cloud due to their large activation radius (> 10 μm) (Grabowski et al., 2022). In contrast, particulate matter with the potential to serve as ice nuclei typically exhibits larger diameters, generally surpassing 500 nm, owing to the presence of a broader spectrum of nucleation-active sites (Demott et al., 2010). This distinction in particle radius underscores the variability in their ability to active as either CCN or IN. In addition, most ice nuclei have appreciable fractions of soluble materials, so the same aerosol particles may serve both as IN and CCN. But the immersion freezing mode is the dominant freezing mechanism for insoluble particles in mixed-phase clouds. Therefore, we think the INP cannot act as CCN in the immersion freezing mode.

*38) Line 323: The NINP was measured in this study not the IN activity. Reformulate.*

**Response:** We have revised the statement as "Note that $PM_{2.5}$ chemical composition was used in this study, which may lead to uncertainties in the interpretation of the INPs in TSP."

*39) Line 338f: Provide a reasonable explanation how PBL height can be positively correlated to NINP. The opposite could be expected due to dilution of air with increasing PBL height. Total particle concentration is usually anticorrelated to PBL height.*

**Response:** Our sampling site is situated atop Changbai Mountain, at an elevation of 2623 m above sea level. The impact of dilution within the boundary layer is negligible at this location, as it consistently above the PBL for the majority of the time. According to Ketterer et al. (2014), the upslope valley wind could increase the altitude of the PBL height locally, and if strong enough, trigger the vertical exchange of PBL air into the free troposphere. In our investigation, the positive correlation between the PBL and INPs may be elucidated by the presence of a valley breeze.

*40) Line 341f: This is not supported by the data. If valley breeze transported INPs to the station during daytime, why is there not a difference between night and day NINP?*

**Response:** Upon examining the diurnal variation of $N_{INP}$ on individual days, notable differences between the daytime and nighttime can be observed on certain days, such as August 18 and 21, 2021, as shown in Figure R4. However, there was no statistical difference when considering the entire sampling period. The absence of diurnal variations may be attributed to the limited dataset size and low sampling frequency, as explained in section 3.1.

[Figure]

Figure R4. Time series of $N_{INP}$ at -12 °C measured during the campaign.

*41) Line 350f: Clarify how NINP is related to phytoplankton bloom and growth. The Japanese Sea experiences a phytoplankton bloom twice a year, one in spring and one in fall (Wang et al., 2022). Clarify the relevance for the current observations.*

**Response:** We identified elevated concentrations of bio-INPs originating from the Japan Sea and inferred that marine biogenic organic materials contribute to INPs. It is worth noting that large amounts of organic matter are present in the sea surface microlayer, regardless of whether there are phytoplankton blooms or not.

*42) Line 352-355, Figure 5: It is surprising that two distinct origins for HTR and LTR bio INPs are found, as it could be assumed that for cumulative NINP the occurrence of high NINP*

*at -10.5°C is correlated to high NINP at -20°C. Is this not the case? Could there be an artifact from the measuring range? For example, how were samples with a frozen fraction =1 at temperatures above -20°C included in the CWT analysis in the LTR? Did you use the data of their dilutions? The NINP colour scale indicates that dilutions were not included.*

**Response:** We clarify that Figure 5d includes the dilution data. The CWT analysis illustrates weight concentration of the target species. In order to reduce the uncertainty caused by the smaller number of trajectories through a grid, the CWT value is multiplied by the weight function as well. Specifically, Korea is an important source of biological INP in the HTR; however, in the LTR, its contribution is reduced relative to other sources. Consequently, its relative contribution is too small to be displayed, considering the calculation of the weight concentration in the CWT.

*43) Line 370ff: Explain how it can be concluded that long-range bio-INP were less prominent in your measurements? It seems contradictory to say there was no qualitative or quantitative analysis of bio INP and then state that they were less prominent from long-range transport.*

**Response:** Thanks for the comment. To avoid confusion, this sentence has been removed in the revision.

*44) Line 376: Clarify the interpretation between PBL height and local sources or long-range transport. Are you inferring that long-range transport contribution to NINP occurs exclusively by night and as soon as the boundary layer forms exclusively local sources contribute to NINP? This seems a critical assumption for your analysis and should be discussed in more detail and supported by stronger evidence.*

**Response:** We apologize for the ambiguity regarding this aspect. The influence of PBL on INPs can be elucidated by the presence of valley breezes, as detailed in our response to comment 2. This phenomenon was recognized as the mountain–valley breeze circulations in the vicinity of Changbai mountain. The CWT calculation encompassed the entire day, including both daytime and nighttime. The results from CWT analysis revealed the influence of long-range transport.

*45) Line 381ff: There was no diurnal cycle of NINP observed in this study. Explain how conclusions can still be drawn about a diurnal cycle.*

**Response:** Thanks for the comment. We have revised this statement as follows: "In summary, our findings suggest that valley breezes and the long-distance transport of air mass from the Japan Sea influence the abundance of INPs at Changbai Mountain."

*46) Line 385f: The methods applied in this study do not allow to explore properties of INPs. Only their abundance was measured. Reformulate.*

**Response:** Thanks for the comment. We have revised this statement as follows:

"Measurements of INPs were carried out at the Changbai Mountain in northeastern Asia to explore the abundance and source of INPs in the immersion freezing mode."

*47) Line 393: An increase in bio NINP indicates that the concentration of bio-INP increased, not the activity. The type of measurements does not provide information if, for example, 1% or 100% of a certain INP type was active at a certain temperature. Reformulate.*

**Response:** Thanks for the comment. We have revised this statement as follows: "Notably, a turning point occurred at −16.5 °C, where FINP-bio increased from 0.8 to 0.9 as the temperature decreased from −16.5 °C to −21.5 °C, indicating an enrichment of active bio-INPs in the low-temperature region (LTR, freezing temperature below $T_{50}$, -17.0 °C ~ -29.0 °C)."

*48) Line 398ff: Please resolve the apparent contradiction between local soil dust sources and CWT pointing to long range transport by clarifying how long-distance sources can be disentangled from local sources. In addition, would the CWT analysis reveal different patterns for the nighttime NINP measurements, or at -10°C and -20°C? Would the correlation analysis, for example with Ca2+ differ if done separately for daytime and nighttime samples?*

**Response:** Thanks for the comment. Local soil dust sources and long-range transport both contribute to the INPs concentrations. However, we cannot completely distinguish between the two aspects based on the field investigation measurements; instead, we analyze them separately. We don't think the two aspects as contradictory.

We examined the relationship between $N_{INP}$ and both $Ca^{2+}$ and WS separately for daytime and nighttime samples, and found the more pronounced correlations during the daytime (Figure R5). This further substantiates our findings, demonstrating the positive correlations between the PBL and INPs during the daytime, which is associated with the influence of valley breeze. These have been added in the revision.

[Figure]

Figure R5. Correlation analysis between $N_{INP}$ with wind speed (WS) and $Ca^{2+}$ as functions of temperature. The r denotes the Pearson correlation coefficients. The asterisk indicates p < 0.05, while the shades indicate that the number of data points is less than half of samples at each temperature.

*49) Figure S5: replace with Fig. R3 from the Author's Response and include the reply to*

*question 22 from Referee #1 explaining how airmasses approach the measurement site and why the trajectory endpoint was chosen at 967m a.g.l.*

**Response:** Thanks. We have replaced the Figure S5 by Figure R3 and provided additional explanations about the influence of the valley breezes in the revised manuscript. The trajectory ending height of 967 m a.g.l. was selected because the terrain height of Changbai Mountain was approximately 1656 m in the GDAS data. These have been added in the revision.

*50) Figure S6: Specify if only daytime or all samples were used to create this figure.*

**Response:** All samples were included in the CWT analysis. We have revised the figure caption to "The concentration-weighted trajectory (CWT) analysis for the distribution of $N_{INP}$-other org (a, b) and $N_{INP\text{-}inorg}$ (c, d) at -15 and -21 °C during the measurement."

***Technical corrections:***

*1) misspelled citations: DeMott, O'Sullivan, McCluskey, in several instances.*

**Response:** Thanks. We checked and removed inappropriate citations in the revised manuscript.

*2) there is no "the" needed before Switzerland. Also, in Figure 2.b) remove "The" in front of Weissfluhjoch and Jungfraujoch.*

**Response:** Thanks. We revised them in the revised manuscript.

*3) Line 110: Maybe use standard error of the mean instead of standard deviation to avoid range of RH (92.4%+11.8%=104,2%) to extend beyond the range of measurement. Also, line 309: 0.5ugm-3 – 1.0ugm-3 = - 0.5ugm-3 is unphysical.*

**Response:** Thanks. In the revision, we have used the standard error of the mean. The average RH stands at 92.4 ± 0.4%, and the average concentration of $Ca^{2+}$ is 0.5 ± 0.2 µg m$^{-3}$. Further adjustments to additional values have been made and elucidated in the revision.

*4) Line 166-168: It is unclear what is explained here. Reformulate.*

**Response:** We have revised this statement as follows: "The rarity of INPs in the atmosphere leads to their low concentration in the suspension. Because the suspension used in the measurement contained a limited number of droplets, we need to consider the resulting uncertainty caused by statistical errors. Therefore, we calculated the confidence intervals of the apparatus for $f_{ice}$ according to the method of Gong et al. (2022) and Agresti and Coull (1998) to address uncertainty associated with the droplet-freezing apparatus".

*5) Line 173: Do you mean: ..., which is 1.96 for a 95% confidence interval? Reformulate.*

**Response:** Yes. We have revised the sentence.

*6) Line 191ff: Incomprehensible sentence. Reformulate.*

**Response:** Thanks. We have revised the sentence.

*7) Line 198: define nij in the text.*

**Response:** We have added the define of $n_{ij}$ in the revised manuscript.

*8) Line 213: replace "three" with "up to 5" to be consistent with what is stated in the conclusion.*

**Response:** Thanks. We have revised the sentence.

*9) Line 233-235: Incomprehensible sentence. Reformulate.*

**Response:** Thanks. We have revised the sentence.

*10) Line 237: Reformulate. ...magnitude at temperatures from -26.0°C to -3.0°C.*

**Response:** We revised the statement as follows: "Gong et al. (2022) measured INPs at the mountain station at Cerro Mirador (622 m a.s.l., Chile), and reported $N_{INP}$ values lower than those in our study by around one order of magnitude at similar freezing temperatures from -26.0 °C to -3.0 °C."

*11) Line 288: replace INPs with NINP*

**Response:** Thanks, we revised the statement as "$N_{INP}$" in the revised manuscript.

*12) Line 321: define SNA.*

**Response:** We define the SNA as "sulfate, nitrate, and ammonium" in the revised manuscript.

*13) Line 329: replace INPs with NINP.*

**Response:** Thanks, we revised the statement as "$N_{INP}$" in the revised manuscript.

*14) Line 336: Define HLR. Do you mean HTR?*

**Response:** Yes, we corrected it to "HTR" in the revised manuscript.

*15) Line 347: "under immersion mode" do you mean "in the immersion mode"?*

**Response:** Yes, we revised the statement as "in the immersion mode" in the revised manuscript.

*16) Line 352: "occurred in" do you mean "coming from"?*

**Response:** Yes, we revised the statement as "originating from" in the revised manuscript.

*17) Line 363: define "the middle layer".*

**Response:** Thanks, we corrected the "middle layer" to "Mesosphere" in the revised manuscript.

*18) When reporting NINP ranges and consecutive the temperature range, I suggest giving the temperature range from high to low temperature if the NINP range is given from low to high, so that the first number in the NINP range corresponds to the first number in the temperature range.*

**Response:** Thanks, we have revised in the revised manuscript.

*19) Specify whenever you mean "ambient temperature" to avoid confusion to experimental temperature of IN experiments. In particular: line 299, 313.*

**Response:** Thanks, we revised the statement as "ambient temperature" in the revised manuscript.

*L70: Replace "high-altitude atmosphere" with "high-altitudes"*

**Response:** Thanks, we have revised it accordingly.

*L71: Delete "In contrast"*

**Response:** Thanks, we have removed it in the revised manuscript.

*L81: Replace "mountains" with "high-altitude sites"*

**Response:** Thanks, we have revised the manuscript accordingly.

*L81: Replace "m a.s.l.), Wuling" with "m a.s.l.), and Wuling"*

**Response:** Thanks, we have revised it accordingly.

*L82: I suggest replacing "Dinghu Mountain (1000 m a.s.l.), and found that the initiated freezing temperature" with "Dinghu Mountain (1000 m a.s.l.). The authors found that the onset freezing temperature was..."*

**Response:** Thanks, we have revised it accordingly.

*L84: "cloud droplets"? Do the authors mean "ice formation"?*

**Response:** Thanks, we revised the statement as "ice formation" in the revised manuscript.

*L89: Add a reference after "yields"*

**Response:** Thanks. We have cited a reference in the revision.

*L89 and along the text: "high altitude". Either use "high altitude" or "high-altitude"*

**Response:** Thanks, we revised the statement as "high-altitude" in the revised manuscript.

*L91: Replace "air mass" with "air masses"*

**Response:** Thanks, we have revised it accordingly.

*L93: Replace "pathways of INPs" with "pathways on the INP population"*

**Response:** Thanks, we have revised it accordingly.

*L93: Replace "understanding of INPs sources" with "understanding of their sources"*

**Response:** Thanks, we have revised the manuscript accordingly.

*L94: I think "clouds" should be "mixed-phase clouds"*

**Response:** Thanks, we revised the statement as "mixed-phase clouds" in the revised manuscript.

*L98: Add a reference after "Arctic"*

**Response:** Thanks. We have cited the references in the revision.

*L110: Replace "field measurements" with "field campaign"*

**Response:** Thanks, we have revised it accordingly.

*L113: Should "national nature reserve" be "national natural reserve"?*

**Response:** Thanks, we revised the statement as "national natural reserve" in the revised manuscript.

*L114-115: Replace "Due to the emergence of novel coronavirus (COVID-19) cases, strict" with something like "Due to the COVID-19 pandemic, strict"*

**Response:** Thanks, we have revised the manuscript accordingly.

*L115: Replace "measures have been implemented" with "measures were implemented"*

**Response:** Thanks, we have revised it accordingly.

*L124: Replace "analysis and detailed" with "analysis with the detailed"*

**Response:** Thanks, we have revised it accordingly.

*L124: Delete "is" in "information is provided"*

**Response:** Thanks, we have removed it in the revised manuscript.

*L133: Replace "frequency increased to every 3 h during" with "frequency was increased from 2h to 3h"*

**Response:** Thanks, we have revised it as: "sampling frequency was increased to every 3 h (5 samples per day)".

*L135: "national meteorological station". Please add the model and manufacturer.*

**Response:** The national meteorological station doesn't have model and manufacturer.

*L135: Replace "from" with "away from"*

**Response:** Thanks, we have revised it accordingly.

*L140: Replace "Germany), a charge-coupled" with "Germany), and a charge-coupled"*

**Response:** Thanks, we have revised it accordingly.

*L140-141: "The Imaging Source, 140 Bremen, Germany), a ring LED light, and a computer system" is unclear.*

**Response:** This segment does not influence the comprehension of the apparatus's structure and the execution of its performance. To enhance clarity, we have removed "a ring LED light, and a computer system" in the revised manuscript.

*L144: Replace "wash off particles" with "wash off the particles"*

**Response:** Thanks, we have revised it accordingly.

*L146: Replace "spectra" with "a spectrum"*

**Response:** Thanks, we have revised it accordingly.

*L147: "Figure S3". This figure cannot be called before Figure S2. Please reorganize the order of the figures to call them sequentially*

**Response:** We changed Figure S3 to Figure S2 in the revised manuscript.

*L149: Replace "and the particle suspension" with something like "and the suspension containing the particles"*

**Response:** Thanks, we have revised it accordingly.

*L160-161: "the unit volume of sampled air". This is unclear.*

**Response:** Thanks, we have revised the statement as "The cumulative concentration of each droplet above K (T), and the cumulative number concentration of INPs ($N_{INP}$) per the unit volume of sampled air were calculated following the method of Vali (1971, 2015):" in the revised manuscript.

*L185: Replace "during our sampling" with "during the sampling"*

**Response:** Thanks, we have revised it accordingly.

*L203: Delete "A metric was applied to evaluate the freezing of droplets, i.e.,"*

**Response:** Thanks, we have revised it accordingly.

*L204: Replace "frozen (T50)" with "frozen (T50) was calculated for each sample"*

**Response:** Thanks, we have revised it accordingly.

*L207: "(for which T50 was -17.0 ± 4.1 ℃)." Please double check this. From figure 2 it seems that the average T50 is round -14C with T50 values as warm as -10C*

**Response:** We diluted all samples and extended the freezing temperature below -25°C. The $T_{50}$ was recalculated as $17.0 \pm 0.6$ °C.

*L212: Replace "The freezing of" with "The onset freezing of"*

**Response:** Thanks, we have revised it accordingly.

*L214: Should "freezing temperatures above T50" be "freezing temperatures above the average T50"?*

**Response:** Yes, we have added "average" before $T_{50}$ in the revised manuscript.

*L225-226: "and the limited dataset size and low sampling frequency may have contributed to the absence of diurnal variations." Is this related to the present study?*

**Response:** Reviewer #1 and #3 suggested that the lack of diurnal variation might be related to sampling intervals and a small number of samples. Therefore, we have provided an explanation of the uncertainty here.

*L228: Units are missing.*

**Response:** Thanks, we added the units in the revision.

*L237: Should "magnitude during the measured" be "magnitude at similar"*

**Response:** Thanks, we have revised it accordingly.

*L240: Replace "nuclei" with "INP"*

**Response:** Thanks, we have revised it accordingly.

*L272: Replace "ice nuclei" with "INP"*

**Response:** Thanks, we have revised it accordingly.

*L288: Replace "atmospheres" with "environments"*

**Response:** Thanks, we have revised it accordingly.

*L293: Replace "with temperature range from -11.0 ℃ to -9.0 ℃." With "at temperatures ranging between -11.0C and -9.0C"*

**Response:** Thanks, we have revised it accordingly.

*L296: Replace "within the range of" with "between"*

**Response:** Thanks, we have revised it accordingly.

*L299: Replace "with temperature" with "with ambient air temperature"*

**Response:** Thanks, we have revised it accordingly.

*L303: Replace "And a significant" with "A significant"*

**Response:** Thanks, we have revised it accordingly.

*L303: Replace "was showed" with "was found"*

**Response:** Thanks, we have revised it accordingly.

*L303-307: Improve the grammar in this paragraph.*

**Response:** Thanks, we have corrected the grammar in this paragraph.

*L308: Replace "with average" with "with an average"*

**Response:** Thanks, we have revised it accordingly.

*L309: "metal ions with Ca2+". This is unclear*

**Response:** Thanks, we have deleted the "metal ions with" in the revised manuscript.

*L313: Replace "did observe that" with "observed"*

**Response:** Thanks, we have revised it accordingly.

*L316: Add a reference after "aerosols"*

**Response:** Thanks, we added a reference, i.e. Carlton et al., 2009 in the revised manuscript.

*L322: Replace "role in INPs formation" with "role as INPs"*

**Response:** Thanks, we have revised it accordingly.

*L322-323: "2016). Note that PM2.5 chemical composition was used in this study, which may lead to uncertainties in the interpretation of the bulk IN activities". This is unclear*

**Response:** We have revised the statement as follows: "Note that PM$_{2.5}$ chemical composition was used in this study, which may lead to uncertainties in the interpretation of the INPs in TSP."

*L334: Replace "significant spanning temperatures from" with "at temperatures ranging between xx and xx."*

**Response:** Thanks, we have revised it accordingly.

*L335: Replace "observed when the" with "observed at the"*

**Response:** Thanks, we have revised it accordingly.

*L338: Replace "discussed in the following paragraph" with "discussed below"*

**Response:** Thanks, we have revised it accordingly.

*L341: Figure S5. This figure cannot be called before S4.*

**Response:** We changed the order of Figure S4 and S5 in the revised manuscript.

*L357: "as -2~-5 ℃". Please fix this.*

**Response:** Thanks, we revised it as "higher temperatures, ranging from −2 °C to −5 °C" in the revised manuscript.

*L358-359: Replace "with biological aerosols produced there able to reach our sampling site through long-distance transport" with "."*

**Response:** Thanks, we have revised it accordingly.

*L365: Should "In global transmission" be "On a global scale"?*

**Response:** Thanks, we revised it as "On a global scale" in the revised manuscript.

*L366: Add a reference after "ocean"*

**Response:** Thanks, we added a reference, i.e. Mayol et al. (2017), in the revised manuscript.

*L369: What do the authors mean with "aerosol transmission"*

**Response:** We revised the statement as follows: "This process can enable biological aerosol to be transported over greater distances, significantly enhancing the ice nucleation activity of dust."

*L369-370: Replace "with the ice nucleation activity of dust significantly enhanced" with "enhancing the ice nucleation activity of dust significantly"*

**Response:** Thanks, we have revised it accordingly.

*L371-272: "activity. Although long-range transported bio-INPs were less prominent in our study". What do the authors mean and how is this supported?*

**Response:** Thanks, we have removed the statement.

*L375: Replace "However, when" with "However, at"*

**Response:** Thanks, we have revised it accordingly.

*L378: Replace "But" with "However"*

**Response:** Thanks, we have revised it accordingly.

*L382: Replace "influence the diurnal cycles of INPs" with "influence the diurnal INPs"*

**Response:** Thanks, we have revised it accordingly.

*L390-391: Please improve the grammar.*

**Response:** Thanks, we have corrected the grammar.

*Figure S1. Why there is not INP data between 2021/7/24 and 2021/8/11?*

**Response:** Thanks for the comments. As shown in Figure S1, the concentration of NOx decreased markedly from 3.0±2.1 ppb during July 24 to August 9 to 0.9±0.3 ppb between August 10 and August 24. This effectively minimizes the influence of human activities on the collection of INPs samples. Therefore, the collection of samples was carried out from August 11th, 2021.

*Figure 1: Replace "(b) This map shows the three-dimensional shape of the sampling site, which was obtained from Google Earth. (c) The ice nuclei sampler (The TH-150D medium flow sampler, Wuhan Tianhong Corporation, China)." With "(b) Map with the three-dimensional shape of the sampling site, which was obtained from Google Earth. (c) The INP sampler (The TH-150D medium flow sampler, Wuhan Tianhong Corporation, China)."*

**Response:** Thanks, we have revised it as "(b) Map with the three-dimensional shape of the sampling site, which was obtained from Google Earth. (c) The INP sampler (The TH-150D medium flow sampler, Wuhan Tianhong Corporation, China)." in the revised manuscript.

*Figure 2: Replace "(b) NINP was measured during the daytime and nighttime. The dark gray shaded area represents the upper and lower limits of NINP over the Weissfluhjoch (2693 m a.s.l.) (Wieder et al., 2022). The yellow shaded area represents the atmospheric NINP ranges at Mt. Huang (1840 m a.s.l) (Jiang et al., 2015). The purple square represents the median NINP at -15 ℃ and -10 ℃ in the Jungfraujoch (3580 m a.s.l.) (Conen et al., 2022). And the black" with "(b) NINP measured during the daytime and nighttime. The dark gray shaded area represents the upper and lower limits of NINP over the Weissfluhjoch (2693 m a.s.l.) (Wieder et al., 2022), the yellow shaded area represents the atmospheric NINP ranges at Mt. Huang (1840 m a.s.l) (Jiang et al., 2015), the purple square represents the median NINP at -15 ℃ and -10 ℃ in the Jungfraujoch (3580 m a.s.l.) (Conen et al., 2022), and the black"*

**Response:** Thanks, we have revised it as "(b) $N_{INP}$ measured during the daytime and nighttime. The dark gray shaded area represents the upper and lower limits of $N_{INP}$ over the Weissfluhjoch (2693 m a.s.l.) (Wieder et al., 2022), the yellow shaded area represents the atmospheric $N_{INP}$ ranges at Mt. Huang (1840 m a.s.l) (Jiang et al., 2015), the purple square represents the median $N_{INP}$ at −15 °C and −10 °C in the Jungfraujoch (3580 m a.s.l.) (Conen et al., 2022), and the black" in the revised manuscript.

*Figure 3: Replace "dots). The point plot represents the median value. The shadow area represents" with "dots). Each point represents the median value and the shadow area represents"*

**Response:** Thanks, we have revised it accordingly.

*Figure 4: Replace "and meteorological parameters, chemical compositions, as functions of temperature." With "with meteorological parameters and chemical compositions as functions of temperature."*

**Response:** Thanks, we have revised it accordingly.

[revised manuscript text omitted]